# Global Carbon Budget 2020

Pierre Friedlingstein[1,2], Michael O'Sullivan[1], Matthew W. Jones[3], Robbie M. Andrew[4], Judith Hauck[5], Are Olsen[6,7], Glen P. Peters[4], Wouter Peters[8,9], Julia Pongratz[10,11], Stephen Sitch[12], Corinne Le Quéré[3], Josep G. Canadell[13], Philippe Ciais[14], Robert B. Jackson[15], Simone Alin[16], Luiz E.O.C. Aragão[17,12], Almut Arneth[18], Vivek Arora[19], Nicholas R. Bates[20,21], Meike Becker[6,7], Alice Benoit-Cattin[22], Henry

[1] College of Engineering, Mathematics and Physical Sciences, University of Exeter, Exeter EX4 4QF, UK

[2] Laboratoire de Météorologie Dynamique, Institut Pierre-Simon Laplace, CNRS-ENS-UPMC-X, Département de Géosciences, Ecole Normale Supérieure, 24 rue Lhomond, 75005 Paris, France

[3] Tyndall Centre for Climate Change Research, School of Environmental Sciences, University of East Anglia, Norwich Research Park, Norwich NR4 7TJ, UK

[4] CICERO Center for International Climate Research, Oslo 0349, Norway

[5] Alfred-Wegener-Institut Helmholtz-Zentrum für Polar- und Meeresforschung, Postfach 120161, 27515 Bremerhaven, Germany

[6] Geophysical Institute, University of Bergen, Bergen, Norway

[7] Bjerknes Centre for Climate Research, Bergen, Norway

[8] Wageningen University, Environmental Sciences Group, P.O. Box 47, 6700AA, Wageningen, The Netherlands

[9] University of Groningen, Centre for Isotope Research, Groningen, The Netherlands

[10] Ludwig-Maximilians-Universität Munich, Luisenstr. 37, 80333 München, Germany

[11] Max Planck Institute for Meteorology, Hamburg, Germany

[12] College of Life and Environmental Sciences, University of Exeter, Exeter EX4 4RJ, UK

[13] CSIRO Oceans and Atmosphere, Canberra, ACT 2101, Australia

[14] Laboratoire des Sciences du Climat et de l'Environnement, LSCE/IPSL, CEA-CNRS-UVSQ, Université Paris-Saclay, F-91198 Gif-sur-Yvette, France

[15] Department of Earth System Science, Woods Institute for the Environment, and Precourt Institute for Energy, Stanford University, Stanford, CA 94305–2210, United States of America

[16] National Oceanic & Atmospheric Administration, Pacific Marine Environmental Laboratory (NOAA/PMEL), 7600 Sand Point Way NE, Seattle, WA 98115, USA

[17] Remote Sensing Division, National Institute for Space Research, São José dos Campos, Brazil

[18] Karlsruhe Institute of Technology, Institute of Meteorology and Climate Research/Atmospheric Environmental Research, 82467 Garmisch-Partenkirchen, Germany

[19] Canadian Centre for Climate Modelling and Analysis, Climate Research Division, Environment and Climate Change Canada, Victoria, BC, Canada

[20] Bermuda Institute of Ocean Sciences (BIOS), 17 Biological Lane, St. Georges, GE01, Bermuda

[21] Department of Ocean and Earth Science, University of Southampton, European Way, Southampton, SO14 3ZH, UK

[22] Marine and Freshwater Research Institute, Fornubudir 5, 220 Hafnarfjordur, Iceland

C. Bittig[23], Laurent Bopp[24], Selma Bultan[10], Naveen Chandra[25,26], Frédéric Chevallier[14], Louise P.
Chini[27], Wiley Evans[28], Liesbeth Florentie[8], Piers M. Forster[29], Thomas Gasser[30], Marion Gehlen[14],
Dennis Gilfillan[31], Thanos Gkritzalis[32], Luke Gregor[33], Nicolas Gruber[33], Ian Harris[34], Kerstin
Hartung[10,35], Vanessa Haverd[13], Richard A. Houghton[36], Tatiana Ilyina[11], Atul K. Jain[37], Emilie
Joetzjer[38], Koji Kadono[39], Etsushi Kato[40], Vassilis Kitidis[41], Jan Ivar Korsbakken[4], Peter
Landschützer[11], Nathalie Lefèvre[42], Andrew Lenton[43], Sebastian Lienert[44], Zhu Liu[45], Danica
Leibniz Institute for Baltic Sea Research Warnemuende (IOW), Seestrasse 15; 18119 Rostock, Germany
Laboratoire de Météorologie Dynamique / Institut Pierre-Simon Laplace, CNRS, Ecole Normale Supérieure / Université PSL, Sorbonne Université, Ecole Polytechnique, Paris, France
Japan Agency for Marine-Earth Science and Technology (JAMSTEC), Yokohama, 236-0001
Center for Global Environmental Research, National Institute for Environmental Studies (NIES), 16-2 Onogawa, Tsukuba, Ibaraki, 305-8506, Japan
Department of Geographical Sciences, University of Maryland, College Park, Maryland 20742, USA
Hakai Institute, Heriot Bay, BC, Canada
Priestley International Centre for Climate, University of Leeds, Leeds, UK
International Institute for Applied Systems Analysis (IIASA), Schlossplatz 1 A-2361 Laxenburg, Austria
Research Institute for Environment, Energy, and Economics, Appalachian State University, Boone, North Carolina, USA
Flanders Marine Institute (VLIZ), InnovOceanSite, Wandelaarkaai 7, 8400 Ostend, Belgium
Environmental Physics Group, ETH Zürich, Institute of Biogeochemistry and Pollutant Dynamics and Center for Climate Systems Modeling (C2SM), Zurich, Switzerland
NCAS-Climate, Climatic Research Unit, School of Environmental Sciences, University of East Anglia, Norwich Research Park, Norwich, NR4 7TJ, UK
Now at: Deutsches Zentrum für Luft- und Raumfahrt, Institut für Physik der Atmosphäre, Oberpfaffenhofen, Germany.
Woods Hole Research Center (WHRC), Falmouth, MA 02540, USA
Department of Atmospheric Sciences, University of Illinois, Urbana, IL 61821, USA
CNRM, Université de Toulouse, Météo-France, CNRS, Toulouse, France
Japan Meteorological Agency, 1-3-4 Otemachi, Chiyoda-Ku, Tokyo 100-8122, Japan
Institute of Applied Energy (IAE), Minato-ku, Tokyo 105-0003, Japan
Plymouth Marine Laboratory (PML), Plymouth, PL13DH, United Kingdom
LOCEAN/IPSL laboratory, Sorbonne Université, CNRS/IRD/MNHN, Paris, France
CSIRO Oceans and Atmosphere, Hobart, TAS, Australia
Climate and Environmental Physics, Physics Institute and Oeschger Centre for Climate Change Research, University of Bern, Bern, Switzerland
Department of Earth System Science, Tsinghua University, Beijing 100084, China

Lombardozzi[46], Gregg Marland[31,47], Nicolas Metzl[42], David R. Munro[48,49], Julia E.M.S. Nabel[11], Shin-Ichiro Nakaoka[26], Yosuke Niwa[26,50], Kevin O´Brien[51,16], Tsuneo Ono[52], Paul I. Palmer[53,54], Denis Pierrot[55], Benjamin Poulter[56], Laure Resplandy[57], Eddy Robertson[58], Christian Rödenbeck[59], Jörg Schwinger[60,7], Roland Séférian[38], Ingunn Skjelvan[60,7], Adam J.P. Smith[3], Adrienne J. Sutton[16], Toste Tanhua[61], Pieter P. Tans[62], Hanqin Tian[63], Bronte Tilbrook[43,64], Guido van der Werf[65], Nicolas Vuichard[14], Anthony P. Walker[66], Rik Wanninkhof[55], Andrew J. Watson[12], David Willis[67], Andrew J. Wiltshire[58], Wenping Yuan[68], Xu Yue[69], Sönke Zaehle[59]

[46] National Center for Atmospheric Research, Climate and Global Dynamics, Terrestrial Sciences Section, Boulder, CO 80305, USA

[47] Department of Geological and Environmental Sciences, Appalachian State University, Boone, North Carolina, USA

[48] Cooperative Institute for Research in Environmental Sciences, University of Colorado, Boulder, CO, 80305, USA

[49] National Oceanic & Atmospheric Administration/Global Monitoring Laboratory (NOAA/GML), Boulder, CO, 80305, USA

[50] Meteorological Research Institute, 1-1 Nagamine, Tsukuba, Ibaraki, 305-0052 Japan

[51] Cooperative Institute for Climate, Ocean and Ecosystem Studies (CICOES), University of Washington, Seattle, WA, USA

[52] Japan Fisheries Research and Education Agency, 2-12-4 Fukuura, Kanazawa-Ku, Yokohama 236-8648, Japan

[53] National Centre for Earth Observation, University of Edinburgh, UK

[54] School of GeoSciences, University of Edinburgh, UK

[55] National Oceanic & Atmospheric Administration/Atlantic Oceanographic & Meteorological Laboratory (NOAA/AOML), Miami, FL 33149, USA

[56] NASA Goddard Space Flight Center, Biospheric Sciences Laboratory, Greenbelt, Maryland 20771, USA

[57] Princeton University, Department of Geosciences and Princeton Environmental Institute, Princeton, NJ, USA

[58] Met Office Hadley Centre, FitzRoy Road, Exeter EX1 3PB, UK

[59] Max Planck Institute for Biogeochemistry, P.O. Box 600164, Hans-Knöll-Str. 10, 07745 Jena, Germany

[60] NORCE Norwegian Research Centre, Jahnebakken 5, 5007 Bergen, Norway

[61] GEOMAR Helmholtz Centre for Ocean Research Kiel, Düsternbrooker Weg 20, 24105 Kiel, Germany

[62] National Oceanic & Atmospheric Administration, Earth System Research Laboratory (NOAA ESRL), Boulder, CO 80305, USA

[63] School of Forestry and Wildlife Sciences, Auburn University, 602 Ducan Drive, Auburn, AL 36849, USA

[64] Australian Antarctic Partnership Program, University of Tasmania, Hobart, Australia

[65] Faculty of Science, Vrije Universiteit, Amsterdam, the Netherlands

[66] Climate Change Science Institute & Environmental Sciences Division, Oak Ridge National Lab

[67] University of East Anglia, Norwich Research Park, Norwich NR4 7TJ, UK

[68] School of Atmospheric Sciences, Guangdong Province Key Laboratory for Climate Change and Natural Disaster Studies, Zhuhai Key Laboratory of Dynamics Urban Climate and Ecology, Sun Yat-sen University, Zhuhai, Guangdong 510245, China.

[69] Jiangsu Key Laboratory of Atmospheric Environment Monitoring and Pollution Control, Collaborative Innovation Center of Atmospheric Environment and Equipment

**Correspondence:** Pierre Friedlingstein (p.friedlingstein@exeter.ac.uk)

Technology, School of Environmental Science and Engineering, Nanjing University of Information Science & Technology (NUIST), Nanjing, 210044, China

**Abstract**
Accurate assessment of anthropogenic carbon dioxide ($CO_2$) emissions and their
redistribution among the atmosphere, ocean, and terrestrial biosphere in a changing
climate – the 'global carbon budget' – is important to better understand the global carbon
cycle, support the development of climate policies, and project future climate change. Here
we describe and synthesize data sets and methodology to quantify the five major
components of the global carbon budget and their uncertainties. Fossil $CO_2$ emissions ($E_{FOS}$)
are based on energy statistics and cement production data, while emissions from land-use
change ($E_{LUC}$), mainly deforestation, are based on land-use and land-use change data and
bookkeeping models. Atmospheric $CO_2$ concentration is measured directly and its growth
rate ($G_{ATM}$) is computed from the annual changes in concentration. The ocean $CO_2$ sink
($S_{OCEAN}$) and terrestrial $CO_2$ sink ($S_{LAND}$) are estimated with global process models constrained
by observations. The resulting carbon budget imbalance ($B_{IM}$), the difference between the
estimated total emissions and the estimated changes in the atmosphere, ocean, and
terrestrial biosphere, is a measure of imperfect data and understanding of the
contemporary carbon cycle. All uncertainties are reported as ±1σ. For the last decade
available (2010-2019), $E_{FOS}$ was 9.4 ± 0.5 GtC yr$^{-1}$, $E_{LUC}$ 1.6 ± 0.7 GtC yr$^{-1}$, $G_{ATM}$ 5.1 ± 0.02 GtC
yr$^{-1}$ (2.4 ± 0.01 ppm yr$^{-1}$), $S_{OCEAN}$ 2.5 ± 0.6 GtC yr$^{-1}$, and $S_{LAND}$ 3.4 ± 0.9 GtC yr$^{-1}$, with a budget
imbalance $B_{IM}$ of -0.1 GtC yr$^{-1}$ indicating a near balance between estimated sources and sinks
over the last decade. For year 2019 alone, the growth in $E_{FOS}$ was only about 0.1% with fossil
emissions increasing to 9.7 ± 0.5 GtC yr$^{-1}$, $E_{LUC}$ was 1.8 ± 0.7 GtC yr$^{-1}$, for a total
anthropogenic $CO_2$ emissions of 11.5± 0.9 GtC yr$^{-1}$ (42.2 ± 3.3 GtCO$_2$). Also for 2019, $G_{ATM}$
was 5.4 ± 0.2 GtC yr$^{-1}$ (2.5 ± 0.1 ppm yr$^{-1}$), $S_{OCEAN}$ was 2.6 ± 0.6 GtC yr$^{-1}$ and $S_{LAND}$ was 3.1 ±
1.2 GtC yr$^{-1}$, with a $B_{IM}$ of 0.3 GtC. The global atmospheric $CO_2$ concentration reached 409.85
± 0.1 ppm averaged over 2019. Preliminary data for 2020, accounting for the COVID-19
induced changes in emissions, suggest a decrease in $E_{FOS}$ relative to 2019 of about  -7%
(median estimate) based on individual estimates from four studies of -6%, -7%, -7% (-3% to -
11%), and -13%. Overall, the mean and trend in the components of the global carbon
budget are consistently estimated over the period 1959-2019, but discrepancies of up to 1
GtC yr$^{-1}$ persist for the representation of semi-decadal variability in $CO_2$ fluxes. Comparison
of estimates from diverse approaches and observations shows: (1) no consensus in the
mean and trend in land-use change emissions over the last decade, (2) a persistent low
agreement between the different methods on the magnitude of the land $CO_2$ flux in the
northern extra-tropics, and (3) an apparent discrepancy between the different methods on
the ocean sink outside the tropics, particularly in the Southern Ocean. This living data
update documents changes in the methods and data sets used in this new global carbon
budget and the progress in understanding of the global carbon cycle compared with
previous publications of this data set (Friedlingstein et al., 2019; Le Quéré et al., 2018b,
2018a, 2016, 2015b, 2015a, 2014, 2013). The data presented in  this work are available at
https://doi.org/10.18160/gcp-2020 (Friedlingstein et al., 2020).
**1    Introduction**
The concentration of carbon dioxide ($CO_2$) in the atmosphere has increased from
approximately 277 parts per million (ppm) in 1750 (Joos and Spahni, 2008), the beginning of
the Industrial Era, to 409.85 ± 0.1 ppm in 2019 (Dlugokencky and Tans, 2020); Fig. 1). The
atmospheric $CO_2$ increase above pre-industrial levels was, initially, primarily caused by the
release of carbon to the atmosphere from deforestation and other land-use change
activities (Ciais et al., 2013). While emissions from fossil fuels started before the Industrial
Era, they became the dominant source of anthropogenic emissions to the atmosphere from
around 1950 and their relative share has continued to increase until present. Anthropogenic
emissions occur on top of an active natural carbon cycle that circulates carbon between the
reservoirs of the atmosphere, ocean, and terrestrial biosphere on time scales from sub-daily
to millennia, while exchanges with geologic reservoirs occur at longer timescales (Archer et
al., 2009).
The global carbon budget presented here refers to the mean, variations, and trends in the
perturbation of $CO_2$ in the environment, referenced to the beginning of the Industrial Era
(defined here as 1750). This paper describes the components of the global carbon cycle over
the historical period with a stronger focus on the recent period (since 1958, onset of
atmospheric $CO_2$ measurements), the last decade (2010-2019), the last year (2019) and the
current year (2020). We quantify the input of $CO_2$ to the atmosphere by emissions from
human activities, the growth rate of atmospheric $CO_2$ concentration, and the resulting
changes in the storage of carbon in the land and ocean reservoirs in response to increasing
atmospheric $CO_2$ levels, climate change and variability, and other anthropogenic and natural
changes (Fig. 2). An understanding of this perturbation budget over time and the underlying
variability and trends of the natural carbon cycle is necessary to understand the response of
natural sinks to changes in climate, $CO_2$ and land-use change drivers, and to quantify the
permissible emissions for a given climate stabilization target. Note that this paper quantifies
the historical global carbon budget, but does not estimate the remaining future carbon
emissions consistent with a given climate target, often referred to as the "remaining carbon
budget" (Millar et al., 2017; Rogelj et al., 2016, 2019).
The components of the $CO_2$ budget that are reported annually in this paper include separate
estimates for the $CO_2$ emissions from (1) fossil fuel combustion and oxidation from all
energy and industrial processes; also including cement production and carbonation ($E_{FOS}$;
GtC yr$^{-1}$) and (2) the emissions resulting from deliberate human activities on land, including
those leading to land-use change ($E_{LUC}$; GtC yr$^{-1}$); and their partitioning among (3) the
growth rate of atmospheric $CO_2$ concentration ($G_{ATM}$; GtC yr$^{-1}$), and the uptake of $CO_2$ (the
'$CO_2$ sinks') in (4) the ocean ($S_{OCEAN}$; GtC yr$^{-1}$) and (5) on land ($S_{LAND}$; GtC yr$^{-1}$). The $CO_2$ sinks
as defined here conceptually include the response of the land (including inland waters and
estuaries) and ocean (including coasts and territorial seas) to elevated $CO_2$ and changes in
climate, rivers, and other environmental conditions, although in practice not all processes
are fully accounted for (see Section 2.7). Global emissions and their partitioning among the
atmosphere, ocean and land are in reality in balance. Due to combination of imperfect
spatial and/or temporal data coverage, errors in each estimate, and smaller terms not
included in our budget estimate (discussed in Section 2.7), their sum does not necessarily
add up to zero. We estimate a budget imbalance ($B_{IM}$), which is a measure of the mismatch
between the estimated emissions and the estimated changes in the atmosphere, land and
ocean, with the full global carbon budget as follows:
$$E_{FOS} + E_{LUC} = G_{ATM} + S_{OCEAN} + S_{LAND} + B_{IM} \qquad\qquad (1)$$
$G_{ATM}$ is usually reported in ppm yr$^{-1}$, which we convert to units of carbon mass per year, GtC
yr$^{-1}$, using 1 ppm = 2.124 GtC (Ballantyne et al., 2012; Table 1). All quantities are presented
in units of gigatonnes of carbon (GtC, $10^{15}$ gC), which is the same as petagrams of carbon
(PgC; Table 1). Units of gigatonnes of $CO_2$ (or billion tonnes of $CO_2$) used in policy are equal
to 3.664 multiplied by the value in units of GtC.
We also include a quantification of $E_{FOS}$ by country, computed with both territorial and
consumption-based accounting (see Section 2), and discuss missing terms from sources
other than the combustion of fossil fuels (see Section 2.7).
The global $CO_2$ budget has been assessed by the Intergovernmental Panel on Climate
Change (IPCC) in all assessment reports (Prentice et al., 2001; Schimel et al., 1995; Watson
et al., 1990; Denman et al., 2007; Ciais et al., 2013), and by others (e.g. Ballantyne et al.,
2012). The Global Carbon Project (GCP, www.globalcarbonproject.org, last access: 16
November 2020) has coordinated this cooperative community effort for the annual
publication of global carbon budgets for the year 2005 (Raupach et al., 2007; including fossil
emissions only), year 2006 (Canadell et al., 2007), year 2007 (published online; GCP, 2007),
year 2008 (Le Quéré et al., 2009), year 2009 (Friedlingstein et al., 2010), year 2010 (Peters et
al., 2012b), year 2012 (Le Quéré et al., 2013; Peters et al., 2013), year 2013 (Le Quéré et al.,
2014), year 2014 (Le Quéré et al., 2015a; Friedlingstein et al., 2014), year 2015 (Jackson et
al., 2016; Le Quéré et al., 2015b), year 2016 (Le Quéré et al., 2016), year 2017 (Le Quéré et

al., 2018a; Peters et al., 2017), year 2018 (Le Quéré et al., 2018b; Jackson et al., 2018) and most recently the year 2019 (Friedlingstein et al., 2019; Jackson et al., 2019; Peters et al., 2019). Each of these papers updated previous estimates with the latest available information for the entire time series.

We adopt a range of ±1 standard deviation (σ) to report the uncertainties in our estimates, representing a likelihood of 68% that the true value will be within the provided range if the errors have a Gaussian distribution and no bias is assumed. This choice reflects the difficulty of characterising the uncertainty in the $CO_2$ fluxes between the atmosphere and the ocean and land reservoirs individually, particularly on an annual basis, as well as the difficulty of updating the $CO_2$ emissions from land-use change. A likelihood of 68% provides an indication of our current capability to quantify each term and its uncertainty given the available information. For comparison, the Fifth Assessment Report of the IPCC (AR5; Ciais et al., 2013) generally reported a likelihood of 90% for large data sets whose uncertainty is well characterised, or for long time intervals less affected by year-to-year variability. Our 68% uncertainty value is near the 66% which the IPCC characterises as 'likely' for values falling into the ±1σ interval. The uncertainties reported here combine statistical analysis of the underlying data and expert judgement of the likelihood of results lying outside this range. The limitations of current information are discussed in the paper and have been examined in detail elsewhere (Ballantyne et al., 2015; Zscheischler et al., 2017). We also use a qualitative assessment of confidence level to characterise the annual estimates from each term based on the type, amount, quality and consistency of the evidence as defined by the IPCC (Stocker et al., 2013).

This paper provides a detailed description of the data sets and methodology used to compute the global carbon budget estimates for the industrial period, from 1750 to 2019, and in more detail for the period since 1959. It also provides decadal averages starting in 1960 including the most recent decade (2010-2019), results for the year 2019, and a projection for the year 2020. Finally it provides cumulative emissions from fossil fuels and land-use change since the year 1750, the pre-industrial period; and since the year 1850, the reference year for historical simulations in IPCC AR6 (Eyring et al., 2016). This paper is updated every year using the format of 'living data' to keep a record of budget versions and the changes in new data, revision of data, and changes in methodology that lead to changes

in estimates of the carbon budget. Additional materials associated with the release of each
new version will be posted at the Global Carbon Project (GCP) website
(http://www.globalcarbonproject.org/carbonbudget, last access: 16 November 2020), with
fossil fuel emissions also available through the Global Carbon Atlas
(http://www.globalcarbonatlas.org, last access: 16 November 2020). With this approach, we
aim to provide the highest transparency and traceability in the reporting of $CO_2$, the key
driver of climate change.

## 2   Methods

Multiple organizations and research groups around the world generated the original
measurements and data used to complete the global carbon budget. The effort presented
here is thus mainly one of synthesis, where results from individual groups are collated,
analysed and evaluated for consistency. We facilitate access to original data with the
understanding that primary data sets will be referenced in future work (see Table 2 for how
to cite the data sets). Descriptions of the measurements, models, and methodologies follow
below and detailed descriptions of each component are provided elsewhere.
This is the 15th version of the global carbon budget and the ninth revised version in the
format of a living data update in Earth System Science Data. It builds on the latest published
global carbon budget of Friedlingstein et al. (2019). The main changes are: (1) the inclusion
of data to year 2019 and a projection for the global carbon budget for year 2020; (2) the
inclusion of gross carbon fluxes associated with land use changes; and (3) the inclusion of
cement carbonation in the fossil fuel and cement component of the budget ($E_{FOS}$). The main
methodological differences between recent annual carbon budgets (2015-2019) are
summarised in Table 3 and previous changes since 2006 are provided in Table A7.

### 2.1   Fossil $CO_2$ emissions ($E_{FOS}$)

### 2.1.1   Emissions estimates

The estimates of global and national fossil $CO_2$ emissions ($E_{FOS}$) include the combustion of
fossil fuels through a wide range of activities (e.g. transport, heating and cooling, industry,
fossil industry own use & natural gas flaring), the production of cement, and other process
emissions (e.g. the production of chemicals & fertilizers) as well as $CO_2$ uptake during the
cement carbonation process. The estimates of $E_{FOS}$ in this study rely primarily on energy
consumption data, specifically data on hydrocarbon fuels, collated and archived by several
organisations (Andres et al., 2012; Andrew, 2020a). We use four main data sets for historical
emissions (1750-2019):
1. Global and national emission estimates for coal, oil, natural gas as well as peat fuel
extraction from the Carbon Dioxide Information Analysis Center (CDIAC) for the time
period 1750-2017 (Gilfillan et al., 2020), as it is the only data set that extends back to
1750 by country.
2. Official national greenhouse gas inventory reports annually for 1990-2018 for the 42
Annex I countries in the UNFCCC (UNFCCC, 2020). We assess these to be the most
accurate estimates because they are compiled by experts within countries that have
access to the most detailed data, and they are periodically reviewed.
3. The BP Statistical Review of World Energy (BP, 2020), as these are the most up-to-date
estimates of national energy statistics.
4. Global and national cement emissions updated from Andrew (2019) to include the latest
estimates of cement production and clinker ratios.
In the following section we provide more details for each data set and describe the
additional modifications that are required to make the data set consistent and usable.
*CDIAC*: The CDIAC estimates have been updated annually to the year 2017, derived primarily
from energy statistics published by the United Nations (UNSD, 2020). Fuel masses and
volumes are converted to fuel energy content using country-level coefficients provided by
the UN, and then converted to $CO_2$ emissions using conversion factors that take into
account the relationship between carbon content and energy (heat) content of the different
fuel types (coal, oil, natural gas, natural gas flaring) and the combustion efficiency (Marland
and Rotty, 1984; Andrew, 2020a). Following Andrew (2020a), we make corrections to
emissions from coal in the Soviet Union during World War II, amounting to a cumulative
reduction of 53 MtC over 1942-43, and corrections to emissions from oil in the Netherland
Antilles and Aruba prior to 1950, amounting to a cumulative reduction of 340 MtC over 23
years.
*UNFCCC*: Estimates from the national greenhouse gas inventory reports submitted to the
United Nations Framework Convention on Climate Change (UNFCCC) follow the IPCC
guidelines (IPCC, 2006; IPCC,2019), but have a slightly larger system boundary than CDIAC
by including emissions coming from carbonates other than in cement manufacture. We
reallocate the detailed UNFCCC sectoral estimates to the CDIAC definitions of coal, oil,
natural gas, cement, and other to allow more consistent comparisons over time and
between countries.
Specific country updates: India: The data reported by CDIAC for India are for the fiscal year
running from April to March (Andrew, 2020a), and various interannual variations in
emissions are not supported by official data. Given that India is the world's third-largest
emitter and that a new data source is available that resolves these issues, we replace CDIAC
estimates with calendar-year estimates through 2019 by Andrew (2020b). Norway: CDIAC's
method of apparent energy consumption results in large errors for Norway, and we
therefore overwrite emissions before 1990 with estimates derived from official Norwegian
statistics.
*BP*: For the most recent year(s) when the UNFCCC and CDIAC estimates are yet not
available, we generate preliminary estimates using energy consumption data (in EJ) from
the BP Statistical Review of World Energy (Andres et al., 2014; BP, 2020; Myhre et al., 2009).
We apply the BP growth rates by fuel type (coal, oil, natural gas) to estimate 2019 emissions
based on 2018 estimates (UNFCCC Annex I countries), and to estimate 2018-2019 emissions
based on 2017 estimates (remaining countries except India). BP's dataset explicitly covers
about 70 countries (96% of global energy emissions), and for the remaining countries we
use growth rates from the sub-region the country belongs to. For the most recent years,
natural gas flaring is assumed constant from the most recent available year of data (2018 for
Annex I countries, 2017 for the remainder). We apply two exceptions to this update using
BP data. The first is for China's coal emissions, for which we use growth rates reported in
official preliminary statistics for 2019 (NBS, 2020b). The second exception is for Australia,
for which BP reports a growth rate of natural gas consumption in Australia of almost 30%,
which is incorrect, and we use a figure of 2.2% derived from Australia's own reporting
(Department of the Environment and Energy, 2020).
*Cement*: Estimates of emissions from cement production are updated from Andrew (2019).
Other carbonate decomposition processes are not included explicitly here, except in
national inventories provided by Annex I countries, but are discussed in Section 2.7.2.

*Country mappings*: The published CDIAC data set includes 257 countries and regions. This list includes countries that no longer exist, such as the USSR and Yugoslavia. We reduce the list to 214 countries by reallocating emissions to currently defined territories, using mass-preserving aggregation or disaggregation. Examples of aggregation include merging East and West Germany to the currently defined Germany. Examples of disaggregation include reallocating the emissions from the former USSR to the resulting independent countries. For disaggregation, we use the emission shares when the current territories first appeared (e.g. USSR in 1992), and thus historical estimates of disaggregated countries should be treated with extreme care. In the case of the USSR, we were able to disaggregate 1990 and 1991 using data from the International Energy Agency (IEA). In addition, we aggregate some overseas territories (e.g. Réunion, Guadeloupe) into their governing nations (e.g. France) to align with UNFCCC reporting.

*Global total*: The global estimate is the sum of the individual countries' emissions and international aviation and marine bunkers. The CDIAC global total differs to the sum of the countries and bunkers since 1) the sum of imports in all countries is not equal to the sum of exports because of reporting inconsistencies, 2) changes in stocks, and 3) the share of non-oxidised carbon (e.g. as solvents, lubricants, feedstocks, etc.) at the global level is assumed to be fixed at the 1970's average while it varies in the country level data based on energy data (Andres et al., 2012). From the 2019 edition CDIAC now includes changes in stocks in the global total (pers. comm., Dennis Gilfillan), removing one contribution to this discrepancy. The discrepancy has grown over time from around zero in 1990 to over 500 $MtCO_2$ in recent years, consistent with the growth in non-oxidised carbon (IEA, 2019). To remove this discrepancy we now calculate the global total as the sum of the countries and international bunkers.

*Cement carbonation*: From the moment it is created, cement begins to absorb $CO_2$ from the atmosphere, a process known as 'cement carbonation'. We estimate this $CO_2$ sink, as the average of two studies in the literature (Cao et al., 2020; Guo et al., *in review*). Both studies use the same model, developed by Xi et al. (2016), with different parameterisations and input data, with the estimate of Guo and colleagues being a revision of Xi et al (2016). The trends of the two studies are very similar. Modelling cement carbonation requires estimation of a large number of parameters, including the different types of cement

material in different countries, the lifetime of the structures before demolition, of cement
waste after demolition, and the volumetric properties of structures, among others (Xi et al.,
2016). Lifetime is an important parameter because demolition results in the exposure of
new surfaces to the carbonation process. The most significant reasons for differences
between the two studies appear to be the assumed lifetimes of cement structures and the
geographic resolution, but the uncertainty bounds of the two studies overlap. In the present
budget, we include the cement carbonation carbon sink in the fossil $CO_2$ emission
component ($E_{FOS}$).

### 311 2.1.2 Uncertainty assessment for $E_{FOS}$

We estimate the uncertainty of the global fossil $CO_2$ emissions at ±5% (scaled down from
the published ±10 % at ±2$\sigma$ to the use of ±1$\sigma$ bounds reported here; Andres et al., 2012).
This is consistent with a more detailed analysis of uncertainty of ±8.4% at ±2$\sigma$ (Andres et al.,
2014) and at the high-end of the range of ±5-10% at ±2$\sigma$ reported by (Ballantyne et al.,
2015). This includes an assessment of uncertainties in the amounts of fuel consumed, the
carbon and heat contents of fuels, and the combustion efficiency. While we consider a fixed
uncertainty of ±5% for all years, the uncertainty as a percentage of the emissions is growing
with time because of the larger share of global emissions from emerging economies and
developing countries (Marland et al., 2009). Generally, emissions from mature economies
with good statistical processes have an uncertainty of only a few per cent (Marland, 2008),
while emissions from strongly developing economies such as China have uncertainties of
around ±10% (for ±1$\sigma$; Gregg et al., 2008; Andres et al., 2014). Uncertainties of emissions
are likely to be mainly systematic errors related to underlying biases of energy statistics and
to the accounting method used by each country.

### 326 2.1.3 Emissions embodied in goods and services

CDIAC, UNFCCC, and BP national emission statistics 'include greenhouse gas emissions and
removals taking place within national territory and offshore areas over which the country
has jurisdiction' (Rypdal et al., 2006), and are called territorial emission inventories.
Consumption-based emission inventories allocate emissions to products that are consumed
within a country, and are conceptually calculated as the territorial emissions minus the
'embodied' territorial emissions to produce exported products plus the emissions in other
countries to produce imported products (Consumption = Territorial – Exports + Imports).
Consumption-based emission attribution results (e.g. Davis and Caldeira, 2010) provide
additional information to territorial-based emissions that can be used to understand
emission drivers (Hertwich and Peters, 2009) and quantify emission transfers by the trade of
products between countries (Peters et al., 2011b). The consumption-based emissions have
the same global total, but reflect the trade-driven movement of emissions across the Earth's
surface in response to human activities.
We estimate consumption-based emissions from 1990-2018 by enumerating the global
supply chain using a global model of the economic relationships between economic sectors
within and between every country (Andrew and Peters, 2013; Peters et al., 2011a). Our
analysis is based on the economic and trade data from the Global Trade and Analysis Project
(GTAP; Narayanan et al., 2015), and we make detailed estimates for the years 1997 (GTAP
version 5), 2001 (GTAP6), and 2004, 2007, and 2011 (GTAP9.2), covering 57 sectors and 141
countries and regions. The detailed results are then extended into an annual time-series
from 1990 to the latest year of the Gross Domestic Product (GDP) data (2018 in this budget),
using GDP data by expenditure in current exchange rate of US dollars (USD; from the UN
National Accounts main Aggregrates database; UN, 2019) and time series of trade data from
GTAP (based on the methodology in Peters et al., 2011b). We estimate the sector-level $CO_2$
emissions using the GTAP data and methodology, include flaring and cement emissions from
CDIAC, and then scale the national totals (excluding bunker fuels) to match the emission
estimates from the carbon budget. We do not provide a separate uncertainty estimate for
the consumption-based emissions, but based on model comparisons and sensitivity analysis,
they are unlikely to be significantly different than for the territorial emission estimates
(Peters et al., 2012a).
**2.1.4 Growth rate in emissions**
We report the annual growth rate in emissions for adjacent years (in percent per year) by
calculating the difference between the two years and then normalising to the emissions in
the first year: $(E_{FOS}(t_{0+1})-E_{FOS}(t_0))/E_{FOS}(t_0)\times100\%$. We apply a leap-year adjustment where
relevant to ensure valid interpretations of annual growth rates. This affects the growth rate
by about 0.3% $yr^{-1}$ (1/366) and causes calculated growth rates to go up approximately 0.3%
if the first year is a leap year and down 0.3% if the second year is a leap year.
The relative growth rate of $E_{FOS}$ over time periods of greater than one year can be rewritten
using its logarithm equivalent as follows:
$$\frac{1}{E_{FOS}}\frac{dE_{FOS}}{dt} = \frac{d(lnE_{FOS})}{dt}$$ (2)
Here we calculate relative growth rates in emissions for multi-year periods (e.g. a decade)
by fitting a linear trend to $ln(E_{FOS})$ in Eq. (2), reported in percent per year.

**2.1.5   Emissions projections**

To gain insight on emission trends for 2020, we provide an assessment of global fossil $CO_2$
emissions, $E_{FOS}$, by combining individual assessments of emissions for China, USA, the EU,
and India (the four countries/regions with the largest emissions), and the rest of the world.
Our analysis this year is different to previous editions of the Global Carbon Budget, as there
have been several independent studies estimating 2020 global $CO_2$ emissions in response to
restrictions related to the COVID-19 pandemic, and the highly unusual nature of the year
makes the projection much more difficult. We consider three separate studies (Le Quéré et
al., 2020, Forster et al., 2020, Liu et al., 2020), in addition to building on the method used in
our previous editions. We separate each method into two parts: first we estimate emissions
for the Year To Date (YTD) and, second, we project emissions for the rest of the year 2020.
Each method is presented in the order it was published.

*2.1.5.1   UEA: Le Quéré et al. (2020)*

YTD: Le Quéré et al (2020) estimated the effect of COVID-19 on emissions using observed
changes in activity using proxy data (such as electricity use, coal use, steel production, road
traffic, aircraft departures, etc), for six sectors of the economy as a function of confinement
levels, scaled to the globe based on policy data in response to the pandemic. The analyses
employed baseline emissions by country for the latest year available (2018 or 2019) from
the Global Carbon Budget 2019 to estimate absolute daily emission changes and covered 67
countries representing 97% of global emissions. Here we use an update through to 13
November. The parameters for the changes in activity by sector were updated for the
industry and aviation sectors, to account for the slow recovery in these sectors observed
since the first peak of the pandemic. Specific country-based parameters were used for India
and the US, which improved the match to the observed monthly emissions (from Section
2.1.5.4). By design, this estimate does not include the background seasonal variability in
emissions (e.g. lower emissions in Northern Hemisphere summer; Jones et al. 2020), nor the
trends in emissions that would be caused by other factors (e.g. reduced use of coal in the EU
and the US). To account for the seasonality in emissions where data is available, the mean
seasonal variability over 2015-2019 was calculated from available monthly emissions data
for the US, EU27, and India (data from Section 2.1.5.4), and added to the UEA estimate for
these regions on Fig. B5. The uncertainty provided reflects the uncertainty in activity
parameters.
Projection: A projection is used to fill the data from 14 November to the end of December,
assuming. countries where confinement measures were at level 1 (targeted measures) on
13 November remain at that level until the end of 2020. For countries where confinement
measures were at more stringent levels 2 & 3 (see Le Quéré et al 2020) on 13 November, we
assume that the measures ease by one level after their announced end date, and then
remain at that level until the end of 2020.
### 2.1.5.2  Priestley Centre: Forster et al. (2020)
YTD: Forster et al. (2020) estimated YTD emissions based primarily on Google mobility data.
The mobility data were used to estimate daily fractional changes in emissions from power,
surface transport, industry, residential, and public and commercial sectors. The analyses
employed baseline emissions for 2019 from the Global Carbon Project to estimate absolute
emission changes and covered 123 countries representing over 99% of global emissions. For
a few countries - most notably China and Iran - Google data was not available and so data
were obtained from the high-reduction estimate from Le Quéré et al (2020). We use an
updated version of Forster et al (2020) in which emission-reduction estimates were
extended through 3 November.
Projection: The estimates were projected from the start of November to the end of
December with the assumption that the declines in emissions from their baselines remain at
66% of the level over the last 30 days with estimates.
### 2.1.5.3  Carbon Monitor: Liu et al. (2020)
YTD: Liu et al (2020) estimated YTD emissions using emission data and emission proxy
activity data including hourly to daily electrical power generation data and carbon emission
factors for each different electricity sources from national electricity operation systems of
31 countries, real-time mobility data (TomTom city congestion index data of 416 cities
worldwide calibrated to reproduce vehicle fluxes in Paris and FlightRadar24 individual flight
location data), monthly industrial production data (calculated separately by cement
production, steel production, chemical production and other industrial production of 27
industries) or indices (primarily Industrial Production Index) from national statistics of 62
countries and regions, and monthly fuel consumption data corrected for the daily
population-weighted air temperature in 206 countries using predefined heating and
temperature functions from EDGAR for residential, commercial and public buildings heating
emissions, to finally calculate the global fossil CO2 emissions, as well as the daily sectoral
emissions from power sector, industry sector, transport sector (including ground transport,
aviation and shipping), and residential sector respectively. We use an updated version of Liu
et al (2020) with data extended through the end of September.
Projection: Liu et al. (2020) did not perform a projection and only presented YTD results. For
purposes of comparison with other methods, we use a simple approach to extrapolating
their observations by assuming the remaining months of the year change by the same
relative amount compared to 2019 in the final month of observations.
### 2.1.5.4  *Global Carbon Budget Estimates*
Previous editions of the Global Carbon Budget (GCB) have estimated YTD emissions, and
performed projections, using sub-annual energy consumption data from a variety of sources
depending on the country or region. The YTD estimates have then been projected to the full
year using specific methods for each country or region. This year we make some
adjustments to this approach, as described below, with detailed descriptions provided in
Appendix C.
China:The YTD estimate is based on monthly data from China's National Bureau of Statistics
and Customs, with the projection based on the relationship between previous monthly data
and full year data to extend the 2020 monthly data to estimate full year emissions.
USA: The YTD and projection are taken directly from the US Energy Information Agency.
EU27: The YTD estimates are based on monthly consumption data of coal, oil, and gas
converted to $CO_2$ and scaled to match previous year emissions.  We use the same method
for the EU27 as for Carbon Monitor described above to generate a full-year projection.
India: YTD estimates are updated from Andrew (2020b), which calculates monthly emissions
directly from detailed energy and cement production data. We use the same method for
India as for Carbon Monitor described above to generate a full-year projection.
Rest of World: There is no YTD estimate, while the 2020 projection is based on a GDP
estimate from the IMF combined with average improvements in carbon intensity observed
in the last 10 years, as in previous editions of the Global Carbon Budget (e.g. Friedlingstein
et al. 2019).
***2.1.5.5   Synthesis***
In the results section we present the estimates from the four different methods, showing
the YTD estimates to the last common historical data point in each dataset and the
projections for 2020.
**2.2   CO$_2$ emissions from land-use, land-use change and forestry (E$_{LUC}$)**
The net CO$_2$ flux from land-use, land-use change and forestry (E$_{LUC}$, called land-use change
emissions in the rest of the text) includes CO$_2$ fluxes from deforestation, afforestation,
logging and forest degradation (including harvest activity), shifting cultivation (cycle of
cutting forest for agriculture, then abandoning), and regrowth of forests following wood
harvest or abandonment of agriculture. Emissions from peat burning and drainage are
added from external datasets (see 2.2.1). Only some land-management activities are
included in our land-use change emissions estimates (Table A1). Some of these activities
lead to emissions of CO$_2$ to the atmosphere, while others lead to CO$_2$ sinks. E$_{LUC}$ is the net
sum of emissions and removals due to all anthropogenic activities considered. Our annual
estimate for 1959-2019 is provided as the average of results from three bookkeeping
approaches (Section 2.2.1): an estimate using the Bookkeeping of Land Use Emissions model
(Hansis et al., 2015; hereafter BLUE), the estimate published by (Houghton and Nassikas,
2017; hereafter H&N2017) and the estimate published by Gasser et al. (2020) using the
compact Earth system model OSCAR, the latter two updated to 2019. All three data sets are
then extrapolated to provide a projection for 2020 (Section 2.2.4). In addition, we use
results from Dynamic Global Vegetation Models (DGVMs; see Section 2.2.2 and Table 4) to
help quantify the uncertainty in E$_{LUC}$ (Section 2.2.3), and thus better characterise our
understanding. Note that we use the scientific E$_{LUC}$ definition, which counts fluxes due to
environmental changes on managed land towards S$_{LAND}$, as opposed to the national
greenhouse gas inventories under the UNFCCC, which include them in E$_{LUC}$ and thus often
report smaller land-use emissions (Grassi et al., 2018; Petrescu et al., 2020).

### 2.2.1  Bookkeeping models

Land-use change $CO_2$ emissions and uptake fluxes are calculated by three bookkeeping models. These are based on the original bookkeeping approach of Houghton (2003) that keeps track of the carbon stored in vegetation and soils before and after a land-use change (transitions between various natural vegetation types, croplands and pastures). Literature-based response curves describe decay of vegetation and soil carbon, including transfer to product pools of different lifetimes, as well as carbon uptake due to regrowth. In addition, the bookkeeping models represent long-term degradation of primary forest as lowered standing vegetation and soil carbon stocks in secondary forests, and also include forest management practices such as wood harvests.

BLUE and H&N2017 exclude land ecosystems' transient response to changes in climate, atmospheric $CO_2$ and other environmental factors, and base the carbon densities on contemporary data from literature and inventory data. Since carbon densities thus remain fixed over time, the additional sink capacity that ecosystems provide in response to $CO_2$-fertilization and some other environmental changes is not captured by these models (Pongratz et al., 2014). On the contrary, OSCAR includes this transient response, and it follows a theoretical framework (Gasser and Ciais, 2013) that allows separating bookkeeping land-use emissions and the loss of additional sink capacity. Only the former is included here, while the latter is discussed in Section 2.7.4. The bookkeeping models differ in (1) computational units (spatially explicit treatment of land-use change for BLUE, country-level for H&N2017, 10 regions and 5 biomes for OSCAR), (2) processes represented (see Table A1), and (3) carbon densities assigned to vegetation and soil of each vegetation type (literature-based for H&N2017 and BLUE, calibrated to DGVMs for OSCAR). A notable change of H&N2017 over the original approach by Houghton (2003) used in earlier budget estimates is that no shifting cultivation or other back- and forth-transitions at a level below country are included. Only a decline in forest area in a country as indicated by the Forest Resource Assessment of the FAO that exceeds the expansion of agricultural area as indicated by FAO is assumed to represent a concurrent expansion and abandonment of cropland. In contrast, the BLUE and OSCAR models include sub-grid-scale transitions between all vegetation types). Furthermore, H&N2017 assume conversion of natural grasslands to pasture, while BLUE and OSCAR allocates pasture proportionally on all natural vegetation that exists in a grid-cell. This is one reason for generally higher emissions in BLUE

and OSCAR. Bookkeeping models do not directly capture carbon emissions from peat fires,
which can create large emissions and interannual variability due to synergies of land-use
and climate variability in Southeast Asia, in particular during El-Niño events, nor emissions
from the organic layers of drained peat soils. To correct for this, H&N2017 includes carbon
emissions from peat burning based on the Global Fire Emission Database (GFED4s; van der
Werf et al., 2017), and peat drainage based on estimates by Hooijer et al. (2010) for
Indonesia and Malaysia. We add GFED4s peat fire emissions to BLUE and OSCAR output, but
use the newly published global FAO peat drainage emissions 1990-2018 from croplands and
grasslands (Conchedda and Tubiello, 2020). We linearly increase tropical drainage emissions
from 0 in 1980, consistent with H&N2017's assumption, and keep emissions from the often
old drained areas of the extratropics constant pre-1990. This adds 8.6 GtC 1960-2019 for
FAO compared to 5.4 GtC for Hooijer et al. (2010). Peat fires add another 2.0 GtC over the
same period.
The three bookkeeping estimates used in this study differ with respect to the land-use
change data used to drive the models. H&N2017 base their estimates directly on the Forest
Resource Assessment of the FAO which provides statistics on forest-area change and
management at intervals of five years currently updated until 2015 (FAO, 2015). The data is
based on country reporting to FAO, and may include remote-sensing information in more
recent assessments. Changes in land-use other than forests are based on annual, national
changes in cropland and pasture areas reported by FAO (FAOSTAT, 2015). On the other
hand, BLUE uses the harmonised land-use change data LUH2-GCB2020 covering the entire
850-2019 period (an update to the previously released LUH2 v2h dataset;
https://doi.org/10.22033/ESGF/input4MIPs.1127; Hurtt et al., 2020), which was also used as
input to the DGVMs (Sec. 2.2.2). It describes land-use change, also based on the FAO data as
well as the HYDE dataset (Goldewijk et al., 2017a, 2017b), but provided at a quarter-degree
spatial resolution, considering sub-grid-scale transitions between primary forest, secondary
forest, primary non-forest, secondary non-forest, cropland, pasture, rangeland, and urban
land (Hurtt et al., 2020). LUH2-GCB2020 provides a distinction between rangelands and
pasture, based on inputs from HYDE. To constrain the models' interpretation on whether
rangeland implies the original natural vegetation to be transformed to grassland or not (e.g.,
browsing on shrubland), a forest mask was provided with LUH2-GCB2020; forest is assumed
to be transformed to grasslands, while other natural vegetation remains (in case of

secondary vegetation) or is degraded from primary to secondary vegetation (Ma et al., 2020). This is implemented in BLUE. OSCAR was run with both LUH2-GCB2019 850-2018 (as used in Friedlingstein et al., 2019) and FAO/FRA (as used by Houghton and Nassikas, 2017), where the latter was extended beyond 2015 with constant 2011-2015 average values. The best-guess OSCAR estimate used in our study is a combination of results for LUH2-GCB2019 and FAO/FRA land-use data and a large number of perturbed parameter simulations weighted against an observational constraint. H&N2017 was extended here for 2016 to 2019 by adding the annual change in total tropical emissions to the H&N2017 estimate for 2015, including estimates of peat drainage and peat burning as described above as well as emissions from tropical deforestation and degradation fires from GFED4.1s (van der Werf et al., 2017). Similarly, OSCAR was extended from 2018 to 2019. Gross fluxes for H&N2017 and OSCAR were extended to 2019 based on a regression of gross sources (including peat emissions) to net emissions for recent years. BLUE's 2019 value was adjusted because the LUH2-GCB2020 forcing for 2019 was an extrapolation of earlier years, thus not capturing the rising deforestation rates occurring in South America in 2019 and the anomalous fire season in Equatorial Asia (see Sec. 2.2.4 and 3.2.1). Anomalies of GFED tropical deforestation and degradation and Equatorial Asia peat fire emissions relative to 2018 are therefore added. Resulting dynamics in the Amazon are consistent with BLUE simulations using directly observed forest cover loss and forest alert data (Hansen et al., 2013; Hansen et al., 2016).

For $E_{LUC}$ from 1850 onwards we average the estimates from BLUE, H&N2017 and OSCAR. For the cumulative numbers starting 1750 an average of four earlier publications is added (30 ± 20 PgC 1750-1850, rounded to nearest 5; Le Quéré et al., 2016).

For the first time we provide estimates of the gross land use change fluxes from which the reported net land-use change flux, $E_{LUC}$, is derived as a sum. Gross fluxes are derived internally by the three bookkeeping models: Gross emissions stem from decaying material left dead on site and from products after clearing of natural vegetation for agricultural purposes, wood harvesting, emissions from peat drainage and peat burning, and, for BLUE, additionally from degradation from primary to secondary land through usage of natural vegetation as rangeland. Gross removals stem from regrowth after agricultural abandonment and wood harvesting.

### 2.2.2 Dynamic Global Vegetation Models (DGVMs)

Land-use change $CO_2$ emissions have also been estimated using an ensemble of 17 DGVM simulations. The DGVMs account for deforestation and regrowth, the most important components of $E_{LUC}$, but they do not represent all processes resulting directly from human activities on land (Table A1). All DGVMs represent processes of vegetation growth and mortality, as well as decomposition of dead organic matter associated with natural cycles, and include the vegetation and soil carbon response to increasing atmospheric $CO_2$ concentration and to climate variability and change. Some models explicitly simulate the coupling of carbon and nitrogen cycles and account for atmospheric N deposition and N fertilisers (Table A1). The DGVMs are independent from the other budget terms except for their use of atmospheric $CO_2$ concentration to calculate the fertilization effect of $CO_2$ on plant photosynthesis.

Many DGVMs used the HYDE land-use change data set (Goldewijk et al., 2017a, 2017b), which provides annual (1700-2019), half-degree, fractional data on cropland and pasture. The data are based on the available annual FAO statistics of change in agricultural land area available until 2015. HYDE version 3.2 used FAO statistics until 2012, which were supplemented using the annual change anomalies from FAO data for years 2013-2015 relative to year 2012. HYDE forcing was also corrected for Brazil for years 1951-2012. After the year 2015 HYDE extrapolates cropland, pasture, and urban land-use data until the year 2019. Some models also use the LUH2-GCB2020 data set, an update of the more comprehensive harmonised land-use data set (Hurtt et al., 2011), that further includes fractional data on primary and secondary forest vegetation, as well as all underlying transitions between land-use states (1700-2019) (https://doi.org/10.22033/ESGF/input4MIPs.1127; Hurtt et al., 2011; Hurtt et al., 2020; Table A1). This new data set is of quarter degree fractional areas of land-use states and all transitions between those states, including a new wood harvest reconstruction, new representation of shifting cultivation, crop rotations, management information including irrigation and fertilizer application. The land-use states include five different crop types in addition to the pasture-rangeland split discussed before. Wood harvest patterns are constrained with Landsat-based tree cover loss data (Hansen et al. 2013). Updates of LUH2-GCB2020 over last year's version (LUH2-GCB2019) are using the most recent HYDE/FAO release (covering the time period up to including 2015), which also corrects an error in the

version used for the 2018 budget in Brazil. The FAO wood harvest data has changed for the
years 2015 onwards and so those are now being used in this year's LUH-GCB2020 dataset.
This means the LUH-GCB2020 data is identical to LUH-GCB2019 for all years up to 2015 and
differs slightly in terms of wood harvest and resulting secondary area/age/biomass for years
after 2015.
DGVMs implement land-use change differently (e.g. an increased cropland fraction in a grid
cell can either be at the expense of grassland or shrubs, or forest, the latter resulting in
deforestation; land cover fractions of the non-agricultural land differ between models).
Similarly, model-specific assumptions are applied to convert deforested biomass or
deforested area, and other forest product pools into carbon, and different choices are made
regarding the allocation of rangelands as natural vegetation or pastures.
The DGVM model runs were forced by either the merged monthly Climate Research Unit
(CRU) and 6 hourly Japanese 55-year Reanalysis (JRA-55) data set or by the monthly CRU
data set, both providing observation-based temperature, precipitation, and incoming
surface radiation on a 0.5°x0.5° grid and updated to 2019 (Harris et al., 2014, 2019). The
combination of CRU monthly data with 6 hourly forcing from JRA-55 (Kobayashi et al., 2015)
is performed with methodology used in previous years (Viovy, 2016) adapted to the
specifics of the JRA-55 data. The forcing data also include global atmospheric $CO_2$, which
changes over time (Dlugokencky and Tans, 2020), and gridded, time dependent N
deposition and N fertilisers (as used in some models; Table A1).
Two sets of simulations were performed with each of the DGVMs. Both applied historical
changes in climate, atmospheric $CO_2$ concentration, and N inputs. The two sets of
simulations differ, however, with respect to land-use: one set applies historical changes in
land-use, the other a time-invariant pre-industrial land cover distribution and pre-industrial
wood harvest rates. By difference of the two simulations, the dynamic evolution of
vegetation biomass and soil carbon pools in response to land-use change can be quantified
in each model ($E_{LUC}$). Using the difference between these two DGVM simulations to
diagnose $E_{LUC}$ means the DGVMs account for the loss of additional sink capacity (around 0.4
± 0.3 GtC yr-1; see Section 2.7.4), while the bookkeeping models do not.
As a criterion for inclusion in this carbon budget, we only retain models that simulate a
positive $E_{LUC}$ during the 1990s, as assessed in the IPCC AR4 (Denman et al., 2007) and AR5
(Ciais et al., 2013).  All DGVMs met this criteria, although one model was not included in the
$E_{LUC}$ estimate from DGVMs as it exhibited a spurious response to the transient land cover
change forcing after its initial spin-up.

### 2.2.3    Uncertainty assessment for $E_{LUC}$

Differences between the bookkeeping models and DGVM models originate from three main
sources: the different methodologies, which among others lead to inclusion of the loss of
additional sink capacity in DGVMs (Section 2.7.4), the underlying land-use/land cover data
set, and the different processes represented (Table A1). We examine the results from the
DGVM models and of the bookkeeping method, and use the resulting variations as a way to
characterise the uncertainty in $E_{LUC}$.
Despite these differences, the $E_{LUC}$ estimate from the DGVMs multi-model mean is
consistent with the average of the emissions from the bookkeeping models (Table 5).
However there are large differences among individual DGVMs (standard deviation at around
0.5 GtC yr$^{-1}$; Table 5), between the bookkeeping estimates (average difference BLUE-
HN2017 of 0.7 GtC yr$^{-1}$, BLUE-OSCAR  of 0.3 GtC yr$^{-1}$, OSCAR-HN2017 of 0.5 GtC yr$^{-1}$), and
between the current estimate of H&N2017 and its previous model version (Houghton et al.,
2012). The uncertainty in $E_{LUC}$ of ±0.7 GtC yr$^{-1}$ reflects our best value judgment that there is
at least 68% chance (±1σ) that the true land-use change emission lies within the given
range, for the range of processes considered here. Prior to the year 1959, the uncertainty in
$E_{LUC}$ was taken from the standard deviation of the DGVMs. We assign low confidence to the
annual estimates of $E_{LUC}$ because of the inconsistencies among estimates and of the
difficulties to quantify some of the processes in DGVMs.

### 2.2.4    Emissions projections for ELUC

We project the 2020 land-use emissions for BLUE, H&N2017 and OSCAR, starting from their
estimates for 2019 assuming unaltered peat drainage, which has low interannual variability,
and the highly variable emissions from peat fires, tropical deforestation and degradation as
estimated using active fire data (MCD14ML; Giglio et al., 2016). Those latter scale almost
linearly with GFED over large areas (van der Werf et al., 2017), and thus allows for tracking
fire emissions in deforestation and tropical peat zones in near-real time. During most years,
emissions during January-September cover most of the fire season in the Amazon and
Southeast Asia, where a large part of the global deforestation takes place and our estimates
capture emissions until October 31st. By the end of October 2020 emissions from tropical
deforestation and degradation fires were estimated to be 227 TgC, down from 347 TgC in
2019 (313 TgC 1997-2019 average). Peat fire emissions in Equatorial Asia were estimated to
be 1 TgC, down from 117 TgC in 2019 (68 TgC 1997-2019 average). The lower fire emissions
for both processes in 2020 compared to 2019 are related to the transition from unusually
dry conditions for a non-El Niño year in Indonesia in 2019, which caused relatively high
emissions, to few fires due to wet conditions throughout 2020. By contrast, fire emissions in
South America remained above-average in 2020, with the slight decrease since 2019
estimated in GFED4.1s (van der Werf et al., 2017) being a conservative estimate. This is
consistent with slightly reduced deforestation rates in 2020 compared to 2019 (note that
often Amazon deforestation is reported from August of the previous to July of the current
year; for such reporting, 2020 deforestation will tend to be higher in 2020 than in 2019 by
including strong deforestation Aug-Dec 2019). Together, this results in pantropical fire
emissions from deforestation, degradation, and peat burning of about 230 Tg C projected
for 2020 as compared to 464 Tg C in 2019; this is slightly above the 2017 and 2018 values of
pantropical fire emissions. Overall, however, we have low confidence in our projection due
to the large uncertainty range we associate with past ELUC, the dependence of 2020
emissions on legacy fluxes from previous years, uncertainties related to fire emissions
estimates, and the lack of data before the end of the year that would allow deforested area
to be quantified accurately. Also, an incomplete coverage of degradation by fire data makes
our estimates conservative, considering that degradation rates in the Amazon increased
from 2019 to 2020 (INPE, 2020).
**2.3   Growth rate in atmospheric CO$_2$ concentration (G$_{ATM}$)**
**2.3.1   Global growth rate in atmospheric CO$_2$ concentration**
The rate of growth of the atmospheric CO$_2$ concentration is provided by the US National
Oceanic and Atmospheric Administration Earth System Research Laboratory (NOAA/ESRL;
Dlugokencky and Tans, 2020), which is updated from Ballantyne et al. (2012). For the 1959-
1979 period, the global growth rate is based on measurements of atmospheric CO$_2$
concentration averaged from the Mauna Loa and South Pole stations, as observed by the
CO$_2$ Program at Scripps Institution of Oceanography (Keeling et al., 1976). For the 1980-
2019 time period, the global growth rate is based on the average of multiple stations
selected from the marine boundary layer sites with well-mixed background air (Ballantyne
et al., 2012), after fitting each station with a smoothed curve as a function of time, and
averaging by latitude band (Masarie and Tans, 1995). The annual growth rate is estimated
by Dlugokencky and Tans (2020) from atmospheric $CO_2$ concentration by taking the average
of the most recent December-January months corrected for the average seasonal cycle and
subtracting this same average one year earlier. The growth rate in units of ppm yr$^{-1}$ is
converted to units of GtC yr$^{-1}$ by multiplying by a factor of 2.124 GtC per ppm (Ballantyne et
al., 2012).
The uncertainty around the atmospheric growth rate is due to four main factors. First, the
long-term reproducibility of reference gas standards (around 0.03 ppm for 1σ from the
1980s; Dlugokencky and Tans, 2020). Second, small unexplained systematic analytical errors
that may have a duration of several months to two years come and go. They have been
simulated by randomizing both the duration and the magnitude (determined from the
existing evidence) in a Monte Carlo procedure. Third, the network composition of the
marine boundary layer with some sites coming or going, gaps in the time series at each site,
etc (Dlugokencky and Tans, 2020). The latter uncertainty was estimated by NOAA/ESRL with
a Monte Carlo method by constructing 100 "alternative" networks (Masarie and Tans, 1995;
NOAA/ESRL, 2019). The second and third uncertainties, summed in quadrature, add up to
0.085 ppm on average (Dlugokencky and Tans, 2020). Fourth, the uncertainty associated
with using the average $CO_2$ concentration from a surface network to approximate the true
atmospheric average $CO_2$ concentration (mass-weighted, in 3 dimensions) as needed to
assess the total atmospheric $CO_2$ burden. In reality, $CO_2$ variations measured at the stations
will not exactly track changes in total atmospheric burden, with offsets in magnitude and
phasing due to vertical and horizontal mixing. This effect must be very small on decadal and
longer time scales, when the atmosphere can be considered well mixed. Preliminary
estimates suggest this effect would increase the annual uncertainty, but a full analysis is not
yet available. We therefore maintain an uncertainty around the annual growth rate based
on the multiple stations data set ranges between 0.11 and 0.72 GtC yr$^{-1}$, with a mean of 0.61
GtC yr$^{-1}$ for 1959-1979 and 0.17 GtC yr$^{-1}$ for 1980-2019, when a larger set of stations were
available as provided by Dlugokencky and Tans (2020), but recognise further exploration of
this uncertainty is required. At this time, we estimate the uncertainty of the decadal
averaged growth rate after 1980 at 0.02 GtC yr$^{-1}$ based on the calibration and the annual
growth rate uncertainty, but stretched over a 10-year interval. For years prior to 1980, we
estimate the decadal averaged uncertainty to be 0.07 GtC yr$^{-1}$ based on a factor
proportional to the annual uncertainty prior and after 1980 (0.02 * [0.61/0.17] GtC yr$^{-1}$).
We assign a high confidence to the annual estimates of $G_{ATM}$ because they are based on
direct measurements from multiple and consistent instruments and stations distributed
around the world (Ballantyne et al., 2012).
In order to estimate the total carbon accumulated in the atmosphere since 1750 or 1850,
we use an atmospheric $CO_2$ concentration of 277 ± 3 ppm or 286 ± 3 ppm, respectively,
based on a cubic spline fit to ice core data (Joos and Spahni, 2008). The uncertainty of ±3
ppm (converted to ±1σ) is taken directly from the IPCC's assessment (Ciais et al., 2013).
Typical uncertainties in the growth rate in atmospheric $CO_2$ concentration from ice core
data are equivalent to ±0.1-0.15 GtC yr$^{-1}$ as evaluated from the Law Dome data (Etheridge et
al., 1996) for individual 20-year intervals over the period from 1850 to 1960 (Bruno and
Joos, 1997).
**2.3.2    Atmospheric growth rate projection**
We provide an assessment of $G_{ATM}$ for 2020 based on the monthly calculated global
atmospheric $CO_2$ concentration (GLO) through August (Dlugokencky and Tans, 2020), and
bias-adjusted Holt–Winters exponential smoothing with additive seasonality (Chatfield,
1978) to project to January 2021. Additional analysis suggests that the first half of the year
shows more interannual variability than the second half of the year, so that the exact
projection method applied to the second half of the year has a relatively smaller impact on
the projection of the full year. Uncertainty is estimated from past variability using the
standard deviation of the last 5 years' monthly growth rates.
**2.4    Ocean $CO_2$ sink**
Estimates of the global ocean $CO_2$ sink $S_{OCEAN}$ are from an ensemble of global ocean
biogeochemistry models (GOBMs, Table A2) that meet observational constraints over the
1990s (see below). The GOBMs constrain the air-sea $CO_2$ flux by the transport of carbon into
the ocean interior, which is also the controlling factor of ocean carbon uptake in the real
world. They cover the full globe and all seasons and were recently evaluated against surface
ocean $pCO_2$ observations, suggesting they are suitable to estimate the annual ocean carbon
sink (Hauck et al., 2020). We use observation-based estimates of $S_{OCEAN}$ to provide a
qualitative assessment of confidence in the reported results, and two diagnostic ocean
models to estimate $S_{OCEAN}$ over the industrial era (see below).

### 2.4.1 Observation-based estimates

We primarily use the observational constraints assessed by IPCC of a mean ocean $CO_2$ sink
of $2.2 \pm 0.7$ GtC yr$^{-1}$ for the 1990s (90% confidence interval; Ciais et al., 2013) to verify that
the GOBMs provide a realistic assessment of $S_{OCEAN}$. We further test that GOBMs and data-
products fall within the IPCC estimates for the 2000s ($2.3 \pm 0.7$ GtC yr$^{-1}$), and the period
2002-2011 ($2.4 \pm 0.7$ GtC yr$^{-1}$, Ciais et al., 2013). The IPCC estimates are based on the
observational constraint of the mean 1990s sink and trends derived mainly from models and
one data-product (Ciais et al., 2013). This is based on indirect observations with seven
different methodologies and their uncertainties, using the methods that are deemed most
reliable for the assessment of this quantity (Denman et al., 2007; Ciais et al., 2013). The
observation-based estimates use the ocean/land $CO_2$ sink partitioning from observed
atmospheric $CO_2$ and $O_2/N_2$ concentration trends (Manning and Keeling, 2006; Keeling and
Manning, 2014), an oceanic inversion method constrained by ocean biogeochemistry data
(Mikaloff Fletcher et al., 2006), and a method based on penetration time scale for
chlorofluorocarbons (McNeil et al., 2003). The IPCC estimate of 2.2 GtC yr$^{-1}$ for the 1990s is
consistent with a range of methods (Wanninkhof et al., 2013).
We also use four estimates of the ocean $CO_2$ sink and its variability based on surface ocean
$pCO_2$ maps obtained by the interpolation of measurements of surface ocean fugacity of $CO_2$
($fCO_2$, which equals $pCO_2$ corrected for the non-ideal behaviour of the gas; Pfeil et al., 2013).
These estimates differ in many respects: they use different maps of surface $pCO_2$, different
atmospheric $CO_2$ concentrations, wind products and different gas-exchange formulations as
specified in Table A3. We refer to them as $pCO_2$-based flux estimates. The measurements
underlying the surface $pCO_2$ maps are from the Surface Ocean $CO_2$ Atlas version 2020
(SOCATv2020; Bakker et al., 2020), which is an update of version 3 (Bakker et al., 2016) and
contains quality-controlled data through 2019 (see data attribution Table A5). Each of the
estimates uses a different method to then map the SOCAT v2020 data to the global ocean.
The methods include a data-driven diagnostic method (Rödenbeck et al., 2013; referred to

here as Jena-MLS), a combined self-organising map and feed-forward neural network (Landschützer et al., 2014; referred to here as MPI-SOMFFN), an artificial neural network model (Denvil-Sommer et al., 2019; Copernicus Marine Environment Monitoring Service, referred to here as CMEMS), and an ensemble average of six machine learning estimates of $pCO_2$ using a cluster regression approach (Gregor et al., 2019; referred to here as CSIR). The ensemble mean of the $pCO_2$-based flux estimates is calculated from these four mapping methods. Further, we show the flux estimate of Watson et al. (2020) whose uptake is substantially larger, owing to a number of adjustments they applied to the surface ocean $fCO_2$ data and the gas-exchange parameterization. Concretely, these authors adjusted the SOCAT $fCO_2$ downward to account for differences in temperature between the depth of the ship intake and the relevant depth right near the surface, and also included a further adjustment to account for the cool surface skin temperature effect. They then used the MPI-SOMFFN method to map the adjusted $fCO_2$ data to the globe. The Watson et al. flux estimate hence differs from the others by their choice of adjusting the flux to a cool, salty ocean surface skin. Watson et al. (2020) showed that this temperature adjustment leads to an upward correction of the ocean carbon sink, up to 0.9 GtC $yr^{-1}$, that, if correct, should be applied to all $pCO_2$-based flux estimates. So far this adjustment is based on a single line of evidence and hence associated with low confidence until further evidence is available. The Watson et al flux estimate presented here is therefore not included in the ensemble mean of the $pCO_2$-based flux estimates. This choice will be reevaluated in upcoming budgets based on further lines of evidence.

The global $pCO_2$-based flux estimates were adjusted to remove the pre-industrial ocean source of $CO_2$ to the atmosphere of 0.61 GtC $yr^{-1}$ from river input to the ocean (the average of 0.45 ± 0.18 GtC yr-1 by Jacobson et al (2007) and 0.78 ± 0.41 GtC yr-1 by Resplandy et al., 2018), to satisfy our definition of $S_{OCEAN}$ (Hauck et al., 2020). The river flux adjustment was distributed over the latitudinal bands using the regional distribution of Aumont et al. (2001; North: 0.16 GtC $yr^{-1}$, Tropics: 0.15 GtC $yr^{-1}$, South: 0.30 GtC $yr^{-1}$). The $CO_2$ flux from each $pCO_2$-based product is scaled by the ratio of the total ocean area covered by the respective product to the total ocean area (361.9e6 $km^2$) from ETOPO1 (Amante and Eakins, 2009; Eakins and Sharman, 2010). In products where the covered area varies with time (MPI-SOMFFN, CMEMS) we use the maximum area coverage. The data-products cover 88% (MPI-

SOMFFN, CMEMS) to 101% (Jena-MLS) of the observed total ocean area, so two products
are effectively corrected upwards by a factor of 1.13 (Table A3, Hauck et al., 2020).
We further use results from two diagnostic ocean models, Khatiwala et al. (2013) and
DeVries (2014), to estimate the anthropogenic carbon accumulated in the ocean prior to
1959. The two approaches assume constant ocean circulation and biological fluxes, with
$S_{OCEAN}$ estimated as a response in the change in atmospheric $CO_2$ concentration calibrated to
observations. The uncertainty in cumulative uptake of ±20 GtC (converted to ±1σ) is taken
directly from the IPCC's review of the literature (Rhein et al., 2013), or about ±30% for the
annual values (Khatiwala et al., 2009).
**2.4.2   Global Ocean Biogeochemistry Models (GOBMs)**
The ocean $CO_2$ sink for 1959-2019 is estimated using nine GOBMs (Table A2). The GOBMs
represent the physical, chemical and biological processes that influence the surface ocean
concentration of $CO_2$ and thus the air-sea $CO_2$ flux. The GOBMs are forced by meteorological
reanalysis and atmospheric $CO_2$ concentration data available for the entire time period.
They mostly differ in the source of the atmospheric forcing data (meteorological reanalysis),
spin up strategies, and in their horizontal and vertical resolutions (Table A2). All GOBMs
except one (CESM-ETHZ) do not include the effects of anthropogenic changes in nutrient
supply  (Duce et al., 2008). They also do not include the perturbation associated with
changes in riverine organic carbon (see Section 2.7.3).
Two sets of simulations were performed with each of the GOBMs. Simulation A applied
historical changes in climate and atmospheric $CO_2$ concentration. Simulation B is a control
simulation with constant atmospheric forcing (normal year or repeated year forcing) and
constant pre-industrial atmospheric $CO_2$ concentration. In order to derive $S_{OCEAN}$ from the
model simulations, we subtracted the annual time-series of the control simulation B from
the annual time-series of simulation A. Assuming that drift and bias are the same in
simulations A and B, we thereby correct for any model drift. Further, this difference also
removes the natural steady state flux (assumed to be 0 GtC yr$^{-1}$ globally) which is often a
major source of biases. Simulation B of IPSL had to be treated differently as it was forced
with constant atmospheric $CO_2$, but observed historical changes in climate. For IPSL, we
fitted a linear trend to the simulation B and subtracted this linear trend from simulation A.
This approach assures that the interannual variability is not removed from IPSL simulation A.
The absolute correction for bias and drift per model in the 1990s varied between <0.01 GtC
yr$^{-1}$ and 0.35 GtC yr$^{-1}$, with six models having positive and three models having negative
biases. This correction reduces the model mean ocean carbon sink by 0.07 GtC yr$^{-1}$ in the
1990s. The $CO_2$ flux from each model is scaled by the ratio of the total ocean area covered
by the respective GOBM to the total ocean area (361.9e6 km$^2$) from ETOPO1 (Amante and
Eakins, 2009; Eakins and Sharman, 2010). The ocean models cover 99% to 101% of the total
ocean area, so the effect of this correction is small.
**2.4.3   GOBM evaluation and uncertainty assessment for S$_{OCEAN}$**
The mean ocean $CO_2$ sink for all GOBMs and the ensemble mean falls within 90% confidence
of the observed range, or 1.5 to 2.9 GtC yr$^{-1}$ for the 1990s (Ciais et al., 2013) and within the
derived constraints for the 2000s and 2002-2011 (see section 2.4.1) before and after
applying corrections. The GOBMs and flux products have been further evaluated using the
fugacity of sea surface $CO_2$ (fCO$_2$) from the SOCAT v2020 database (Bakker et al., 2016,
2020). The fugacity of $CO_2$ is 3–4‰ smaller than the partial pressure of $CO_2$ (Zeebe and
Wolf-Gladrow, 2001). We focused this evaluation on the root mean squared error (RMSE)
between observed fCO$_2$ and modelled pCO$_2$ and on a measure of the amplitude of the
interannual variability of the flux (modified after Rödenbeck et al., 2015).  The RMSE is
calculated from annually and regionally averaged time-series calculated from GOBM and
data-product pCO$_2$ subsampled to open ocean (water depth > 400 m) SOCAT sampling
points to measure the misfit between large-scale signals (Hauck et al., 2020) as opposed to
the RMSE calculated from binned monthly data as in the previous year. The amplitude of
the S$_{OCEAN}$ interannual variability (A-IAV) is calculated as the temporal standard deviation of
the detrended $CO_2$ flux time-series (Rödenbeck et al., 2015, Hauck et al., 2020). These
metrics are chosen because RMSE is the most direct measure of data-model mismatch and
the A-IAV is a direct measure of the variability of S$_{OCEAN}$ on interannual timescales. We apply
these metrics globally and by latitude bands (Fig. B1). Results are shown in Fig. B1 and
discussed in Section 3.1.3.
The 1-σ uncertainty around the mean ocean sink of anthropogenic $CO_2$ was quantified by
Denman et al. (2007) for the 1990s to be ± 0.5 GtC yr$^{-1}$. Here we scale the uncertainty of ±
0.5 GtC yr$^{-1}$ to the mean estimate of 2.2 GtC yr$^{-1}$ in the 1990s to obtain a relative uncertainty
of ± 18%, which is then applied to the full time-series. To quantify the uncertainty around
annual values, we examine the standard deviation of the GOBM ensemble, which varies
between 0.2 and 0.4 GtC yr$^{-1}$ and averages to 0.30 GtC yr$^{-1}$ during 1959-2019. We estimate
that the uncertainty in the annual ocean $CO_2$ sink increases from ± 0.3 GtC yr$^{-1}$ in the 1960s
to ± 0.6 GtC yr$^{-1}$ in the decade 2010-19 from the combined uncertainty of the mean flux
based on observations of ± 18% (Denman et al., 2007) and the standard deviation across
GOBMs of up to ± 0.4 GtC yr$^{-1}$, reflecting both the uncertainty in the mean sink from
observations during the 1990s (Denman et al., 2007; Section 2.4.1) and the uncertainty in
annual estimates from the standard deviation across the GOBM ensemble.
We examine the consistency between the variability of the model-based and the $pCO_2$-
based flux products to assess confidence in $S_{OCEAN}$. The interannual variability of the ocean
fluxes (quantified as A-IAV, the standard deviation after detrending, Figure B1) of the four
$pCO_2$-based flux products plus the Watson et al. product for 1992-2019, ranges from 0.16 to
0.25 GtC yr$^{-1}$ with the lower estimates by the two ensemble methods (CSIR, CMEMS). The
inter-annual variability in the GOBMs ranges between 0.11 and 0.17 GtC yr$^{-1}$, hence there is
overlap with the lower A-IAV estimates of two data-products.
Individual estimates (both GOBM and flux products) generally produce a higher ocean $CO_2$
sink during strong El Niño events. There is emerging agreement between GOBMs and data-
products on the patterns of decadal variability of $S_{OCEAN}$ with a global stagnation in the
1990s and an extra-tropical strengthening in the 2000s (McKinley et al., 2020, Hauck et al.,

2020).

The annual $pCO_2$-based flux products correlate with the ocean $CO_2$ sink estimated here with
a correlation coefficient *r* ranging from 0.80 to 0.97 (1985-2019). The central estimates of
the annual flux from the GOBMs and the $pCO_2$-based flux products have a correlation *r* of
0.97 (1985-2019). The agreement between the models and the flux products reflects some
consistency in their representation of underlying variability since there is little overlap in
their methodology or use of observations. We assess a medium confidence level to the
annual ocean $CO_2$ sink and its uncertainty because it is based on multiple lines of evidence,
it is consistent with ocean interior carbon estimates (Gruber et al., 2019, see section 3.1.2)
and the results are consistent in that the interannual variability in the GOBMs and data-
based estimates are all generally small compared to the variability in the growth rate of
atmospheric $CO_2$ concentration.

## 2.5    Terrestrial CO$_2$ sink

### 2.5.1    DGVM simulations

The terrestrial land sink (S$_{LAND}$) is thought to be due to the combined effects of fertilisation by rising atmospheric CO$_2$ and N inputs on plant growth, as well as the effects of climate change such as the lengthening of the growing season in northern temperate and boreal areas. S$_{LAND}$ does not include land sinks directly resulting from land-use and land-use change (e.g. regrowth of vegetation) as these are part of the land-use flux (E$_{LUC}$), although system boundaries make it difficult to attribute exactly CO$_2$ fluxes on land between S$_{LAND}$ and E$_{LUC}$ (Erb et al., 2013).

S$_{LAND}$ is estimated from the multi-model mean of 17 DGVMs (Table 4). As described in section 2.2.2, DGVM simulations include all climate variability and CO$_2$ effects over land, with 12 DGVMs also including the effect of N inputs. The DGVMs estimate of S$_{LAND}$ does not include the export of carbon to aquatic systems or its historical perturbation, which is discussed in section 2.7.3.

### 2.5.2    DGVM evaluation and uncertainty assessment for S$_{LAND}$

We apply three criteria for minimum DGVM realism by including only those DGVMs with (1) steady state after spin up, (2) global net land flux (S$_{LAND}$ – E$_{LUC}$) that is an atmosphere-to-land carbon flux over the 1990s ranging between -0.3 and 2.3 GtC yr$^{-1}$, within 90% confidence of constraints by global atmospheric and oceanic observations (Keeling and Manning, 2014; Wanninkhof et al., 2013), and (3) global E$_{LUC}$ that is a carbon source to the atmosphere over the 1990s, as already mentioned in section 2.2.2. All 17 DGVMs meet these three criteria.

In addition, the DGVM results are also evaluated using the International Land Model Benchmarking system (ILAMB; Collier et al., 2018). This evaluation is provided here to document, encourage and support model improvements through time. ILAMB variables cover key processes that are relevant for the quantification of S$_{LAND}$ and resulting aggregated outcomes. The selected variables are vegetation biomass, gross primary productivity, leaf area index, net ecosystem exchange, ecosystem respiration, evapotranspiration, soil carbon, and runoff (see Fig. B2 for the results and for the list of observed databases). Results are shown in Fig. B2 and discussed in Section 3.1.3.

For the uncertainty for $S_{LAND}$, we use the standard deviation of the annual $CO_2$ sink across
the DGVMs, averaging to about ± 0.6 GtC yr$^{-1}$ for the period 1959 to 2019. We attach a
medium confidence level to the annual land $CO_2$ sink and its uncertainty because the
estimates from the residual budget and averaged DGVMs match well within their respective
uncertainties (Table 5).
**2.6   The atmospheric inversion perspective**
The world-wide network of in-situ atmospheric measurements and satellite derived
atmospheric $CO_2$ column ($xCO_2$) observations can be used with atmospheric inversion
methods to constrain the location of the combined total surface $CO_2$ fluxes from all sources,
including fossil and land-use change emissions and land and ocean $CO_2$ fluxes. The
inversions assume $E_{FOS}$ to be well known, and they solve for the spatial and temporal
distribution of land and ocean fluxes from the residual gradients of $CO_2$ between stations
that are not explained by fossil fuel emissions.
Six atmospheric inversions (Table A4) used atmospheric $CO_2$ data to the end of 2019
(including preliminary values in some cases) to infer the spatio-temporal distribution of the
$CO_2$ flux exchanged between the atmosphere and the land or oceans. We focus here on the
total land and ocean $CO_2$ fluxes and their partitioning among the Northern extratropics
(30°N-90°N), the tropics (30°S-30°N) and the Southern extratropics (30°S-90°S). We also
break down those estimates for the land and ocean regions separately. We use these
estimates to comment on the consistency across various data streams and process-based
estimates.
The six inversion systems used in this release are described in Table A4. The inversions are
based on Bayesian inversion principles with prior information on fluxes and their
uncertainties. The inversion systems are based on near-identical observations of surface
measurements of $CO_2$ time series (or subsets thereof) from various flask and in situ
networks. Two  inversion systems (UoE and CAMS) were also applied using only satellite
$xCO_2$ measurements from GOSAT or OCO-2, but their results at the larger scales discussed in
this work did not deviate substantially from their in-situ counterparts, and are therefore not
separately included (Palmer et al., 2019). Each inversion system uses different
methodologies and input data but is rooted in Bayesian inversion principles (Table A4).
These differences mainly concern the selection of atmospheric $CO_2$ data and prior fluxes, as
well as the spatial resolution, assumed correlation structures, and mathematical approach
of the models. The details of each model's approach are documented extensively in the
references provided in Table A4. Each system uses a different transport model, which was
demonstrated to be a driving factor behind differences in atmospheric inversion based flux
estimates, and specifically their distribution across latitudinal bands (Gaubert et al., 2019;
Schuh et al., 2019).
The inversion systems prescribe global fossil fuel emissions. For the first time in this year's
budget, most (five of the six) inversion systems prescribed the same estimate for $E_{FOS}$;
specifically, the GCP's Gridded Fossil Emissions Dataset version 2020.1 (GCP-
GridFEDv2020.1), which is an update to 2019 of the first version of GCP-GridFED presented
by Jones et al. (2020). GCP-GridFEDv2020.1 scales gridded estimates of $CO_2$ emissions from
EDGARv4.3.2 (Janssens-Maenhout et al., 2019) within national territories to match national
emissions estimates provided by the GCP for the years 1959-2019, which were compiled
following the methodology described in section 2.1 with all datasets available on 31st July
2020 (R. Andrew, *pers. comm.*).
A new feature in this edition of the global carbon budget is the use of a consistent prior
emissions dataset for $E_{FOS}$ across almost all inversion models, which avoids the need to
correct the estimated land sink (by up to 0.5 GtC in the Northern extratropics) for most
models. Only the UoE inversion used an alternative dataset and required a post-processing
correction (see Table A4). Further, the use of GCP-GridFEDv2020.1 for $E_{FOS}$ ensures a close
alignment with the estimate of $E_{FOS}$ used in this budget assessment, enhancing the
comparability of the inversion-based estimate with the flux estimates deriving from DGVMs,
GOBMs and $pCO_2$-based methods.
The land and ocean $CO_2$ fluxes from atmospheric inversions contain anthropogenic
perturbation and natural pre-industrial $CO_2$ fluxes. On annual time scales, natural pre-
industrial fluxes are primarily land $CO_2$ sinks and ocean $CO_2$ sources corresponding to carbon
taken up on land, transported by rivers from land to ocean, and outgassed by the ocean.
These pre-industrial land $CO_2$ sinks are thus compensated over the globe by ocean $CO_2$
sources corresponding to the outgassing of riverine carbon inputs to the ocean. We apply
the distribution of land-to-ocean C fluxes from rivers in three latitude bands using estimates
from Resplandy et al. (2018), which are constrained by ocean heat transport to a total land-
to-ocean carbon transfer of 0.61 GtC $yr^{-1}$. The latitude distribution of river-induced ocean
$CO_2$ sources (North: 0.16 GtC $yr^{-1}$, Tropics: 0.15 GtC $yr^{-1}$, South: 0.30 GtC $yr^{-1}$) from carbon
originating from land ( North: 0.29 GtC $yr^{-1}$, Tropics: 0.32 GtC $yr^{-1}$, South: <0.01 GtC $yr^{-1}$ ) are
derived by scaling the outgassing per latitude band from Aumont et al. (2001) to the global
estimate of 0.61 GtC $yr^{-1}$ . To facilitate the comparison, we adjusted the inverse estimates of
the land and ocean fluxes per latitude band with these numbers to produce historical
perturbation $CO_2$ fluxes from inversions.
The atmospheric inversions are also evaluated using vertical profiles of atmospheric $CO_2$
concentrations (Fig. B3). More than 30 aircraft programs over the globe, either regular
programs or repeated surveys over at least 9 months, have been used in order to draw a
robust picture of the model performance (with space-time data coverage irregular and
denser in the 0-45°N latitude band; Table A6). The six models are compared to the
independent aircraft $CO_2$ measurements between 2 and 7 km above sea level between 2001
and 2018. Results are shown in Fig. B3 and discussed in Section 3.1.3.
**2.7     Processes not included in the global carbon budget**
The contribution of anthropogenic CO and $CH_4$ to the global carbon budget is not fully
accounted for in Eq. (1) and is described in Section 2.7.1. The contributions of other
carbonates to $CO_2$ emissions is described in Section 2.7.2. The contribution of anthropogenic
changes in river fluxes is conceptually included in Eq. (1) in $S_{OCEAN}$ and in $S_{LAND}$, but it is not
represented in the process models used to quantify these fluxes. This effect is discussed in
Section 2.7.3. Similarly, the loss of additional sink capacity from reduced forest cover is
missing in the combination of approaches used here to estimate both land fluxes ($E_{LUC}$ and
$S_{LAND}$) and its potential effect is discussed and quantified in Section 2.7.4.
**2.7.1     Contribution of anthropogenic CO and $CH_4$ to the global carbon budget**
Equation (1) includes only partly the net input of $CO_2$ to the atmosphere from the chemical
oxidation of reactive carbon-containing gases from sources other than the combustion of
fossil fuels, such as: (1) cement process emissions, since these do not come from
combustion of fossil fuels, (2) the oxidation of fossil fuels, (3) the assumption of immediate
oxidation of vented methane in oil production. It omits however any other anthropogenic
carbon-containing gases that are eventually oxidised in the atmosphere, such as
anthropogenic emissions of CO and $CH_4$. An attempt is made in this section to estimate their
magnitude, and identify the sources of uncertainty. Anthropogenic CO emissions are from
incomplete fossil fuel and biofuel burning and deforestation fires. The main anthropogenic
emissions of fossil $CH_4$ that matter for the global (anthropogenic) carbon budget are the
fugitive emissions of coal, oil and gas sectors (see below). These emissions of CO and $CH_4$
contribute a net addition of fossil carbon to the atmosphere.
In our estimate of $E_{FOS}$ we assumed (Section 2.1.1) that all the fuel burned is emitted as $CO_2$,
thus CO anthropogenic emissions associated with incomplete fossil fuel combustion and its
atmospheric oxidation into $CO_2$ within a few months are already counted implicitly in $E_{FOS}$
and should not be counted twice (same for $E_{LUC}$ and anthropogenic CO emissions by
deforestation fires). Anthropogenic emissions of fossil $CH_4$ are however not included in $E_{FOS}$,
because these fugitive emissions are not included in the fuel inventories. Yet they
contribute to the annual $CO_2$ growth rate after $CH_4$ gets oxidized into $CO_2$. Emissions of fossil
$CH_4$ represent 30% of total anthropogenic $CH_4$ emissions (Saunois et al. 2020 ; their top-
down estimate is used because it is consistent with the observed $CH_4$ growth rate), that is
0.083 GtC $yr^{-1}$ for the decade 2008-2017. Assuming steady state, an amount equal to this
fossil CH4 emission is all converted to $CO_2$ by OH oxidation, and thus explain 0.083 GtC $yr^{-1}$
of the global $CO_2$ growth rate with an uncertainty range of 0.061 to 0.098  GtC $yr^{-1}$ taken
from the min-max of top-down estimates in Saunois et al. (2020). If this min-max range is
assumed to be 2 σ because Saunois et al. (2020) did not account for the internal uncertainty
of their min and max top-down estimates, it translates into a 1-σ uncertainty of 0.019 GtC
$yr^{-1}$.
Other anthropogenic changes in the sources of CO and $CH_4$ from wildfires, vegetation
biomass, wetlands, ruminants or permafrost changes are similarly assumed to have a small
effect on the $CO_2$ growth rate. The $CH_4$ and CO emissions and sinks are published and
analysed separately in the Global Methane Budget and Global Carbon Monoxide Budget
publications, which follow a similar approach to that presented here (Saunois et al., 2020;
Zheng et al., 2019).
**2.7.2   Contribution of other carbonates to $CO_2$ emissions**
This year we account for cement carbonation (a carbon sink) for the first time. The
contribution of emissions of fossil carbonates (carbon sources) other than cement
production is not systematically included in estimates of $E_{FOS}$, except at the national level
where they are accounted in the UNFCCC national inventories. The missing processes
include $CO_2$ emissions associated with the calcination of lime and limestone outside cement
production. Carbonates are also used in various industries, including in iron and steel
manufacture and in agriculture. They are found naturally in some coals. $CO_2$ emissions from
fossil carbonates other than cement are estimated to amount to about 1% of $E_{FOS}$ (Crippa et
al., 2019), though some of these carbonate emissions are included in our estimates (e.g., via
UNFCCC inventories).
**2.7.3  Anthropogenic carbon fluxes in the land-to-ocean aquatic continuum**
The approach used to determine the global carbon budget refers to the mean, variations,
and trends in the perturbation of $CO_2$ in the atmosphere, referenced to the pre-industrial
era. Carbon is continuously displaced from the land to the ocean through the land-ocean
aquatic continuum (LOAC) comprising freshwaters, estuaries and coastal areas (Bauer et al.,
2013; Regnier et al., 2013). A significant fraction of this lateral carbon flux is entirely
'natural' and is thus a steady state component of the pre-industrial carbon cycle. We
account for this pre-industrial flux where appropriate in our study. However, changes in
environmental conditions and land-use change have caused an increase in the lateral
transport of carbon into the LOAC – a perturbation that is relevant for the global carbon
budget presented here.
The results of the analysis of Regnier et al. (2013) can be summarized in two points of
relevance for the anthropogenic $CO_2$ budget. First, the anthropogenic perturbation of the
LOAC has increased the organic carbon export from terrestrial ecosystems to the
hydrosphere by as much as $1.0 \pm 0.5$ GtC yr$^{-1}$ since pre-industrial, mainly owing to enhanced
carbon export from soils. Second, this exported anthropogenic carbon is partly respired
through the LOAC, partly sequestered in sediments along the LOAC and to a lesser extent,
transferred to the open ocean where it may accumulate or be outgassed. The increase in
storage of land-derived organic carbon in the LOAC carbon reservoirs (burial) and in the
open ocean combined is estimated by Regnier et al. (2013) at $0.65 \pm 0.35$GtC yr$^{-1}$. The
inclusion of LOAC related anthropogenic $CO_2$ fluxes should affect estimates of $S_{LAND}$ and
$S_{OCEAN}$ in Eq. (1), but does not affect the other terms. Representation of the anthropogenic
perturbation of LOAC $CO_2$ fluxes is however not included in the GOBMs and DGVMs used in
our global carbon budget analysis presented here.

### 2.7.4 Loss of additional sink capacity

Historical land-cover change was dominated by transitions from vegetation types that can
provide a large carbon sink per area unit (typically, forests) to others less efficient in
removing $CO_2$ from the atmosphere (typically, croplands). The resultant decrease in land
sink, called the 'loss of additional sink capacity', can be calculated as the difference between
the actual land sink under changing land-cover and the counterfactual land sink under pre-
industrial land-cover. This term is not accounted for in our global carbon budget estimate.
Here, we provide a quantitative estimate of this term to be used in the discussion. Seven of
the DGVMs used in Friedlingstein et al (2019) performed additional simulations with and
without land-use change under cycled pre-industrial environmental conditions. The
resulting loss of additional sink capacity amounts to $0.9 \pm 0.3$ GtC $yr^{-1}$ on average over 2009-
2018 and $42 \pm 16$ GtC accumulated between 1850 and 2018. OSCAR, emulating the
behaviour of 11 DGVMs finds values of the loss of additional sink capacity of $0.7 \pm 0.6$ GtC
$yr^{-1}$ and $31 \pm 23$ GtC for the same time period (Gasser et al., 2020). Since the DGVM-based
$E_{LUC}$ estimates are only used to quantify the uncertainty around the bookkeeping models'
$E_{LUC}$ we do not add the loss of additional sink capacity to the bookkeeping estimate.

## 3   Results

### 3.1   Global carbon budget mean and variability for 1959-2019

The global carbon budget averaged over the historical period (1850-2019) is shown in Fig. 3.
For the more recent 1959-2019 period where direct atmospheric $CO_2$ measurements are
available, 81% of the total emissions ($E_{FOS}$ + $E_{LUC}$) were caused by fossil $CO_2$ emissions, and
19% by land-use change. The total emissions were partitioned among the atmosphere
(45%), ocean (24%) and land (32%), with a near-zero unattributed budget imbalance (0%).
All components except land-use change emissions have significantly grown since 1959, with
important interannual variability in the growth rate in atmospheric $CO_2$ concentration and in
the land $CO_2$ sink (Fig. 4), and some decadal variability in all terms (Table 6). Differences
with previous budget releases are documented in Fig. B4.

### 3.1.1 CO₂ emissions

Global fossil $CO_2$ emissions have increased every decade from an average of $3.0 \pm 0.2$ GtC yr⁻¹ for the decade of the 1960s to an average of $9.4 \pm 0.5$ GtC yr⁻¹ during 2010-2019 (Table 6, Fig. 2 and Fig. 5). The growth rate in these emissions decreased between the 1960s and the 1990s, from 4.3% yr⁻¹ in the 1960s (1960-1969), 3.1% yr⁻¹ in the 1970s (1970-1979), 1.6% yr⁻¹ in the 1980s (1980-1989), to 0.9% yr⁻¹ in the 1990s (1990-1999). After this period, the growth rate began increasing again in the 2000s at an average growth rate of 3.0% yr⁻¹, decreasing to 1.2% yr⁻¹ for the last decade (2010-2019).

In contrast, $CO_2$ emissions from land-use, land-use change and forestry have remained relatively constant, at around $1.4 \pm 0.7$ GtC yr⁻¹ over the past half-century (Table 6) but with large spread across estimates (Table 5, Fig. 6). These emissions are also relatively constant in the DGVM ensemble of models, except during the last decade when they increase to $2.1 \pm 0.5$ GtC yr⁻¹. However, there is no agreement on this recent increase between the bookkeeping estimates, with H&N2017 suggesting a downward trend as compared to a weak and strong upward trend in OSCAR and the BLUE estimates respectively (Fig. 6).

$E_{LUC}$ is a net term of various gross fluxes, which comprise emissions and removals (see Sec. 2.2.1). Gross emissions are on average 2-3 times larger than the net $E_{LUC}$ emissions, increasing from an average of $3.5 \pm 1.2$ GtC yr⁻¹ for the decade of the 1960s to an average of $4.4 \pm 1.6$ GtC yr⁻¹ during 2010-2019 (Fig. 6, Table 5), showing the relevance of land management such as harvesting or rotational agriculture. They differ more across the three bookkeeping estimates than net fluxes, which is expected due to different process representation; in particular explicit inclusion of shifting cultivation (BLUE, OSCAR) increases both gross emissions and removals.

The uptake of $CO_2$ by cement via carbonation has increased with increasing stocks of cement products, from an average of 20 MtC yr⁻¹ in the 1960s to an average of 190 MtC yr⁻¹ during 2010-2019 (Fig. 5). The growth rate declined from 6.7% yr⁻¹ in the 1960s to 3.3% yr⁻¹ in the 1980s, rising again to 6.2% yr⁻¹ in the 2000s, before declining again to 3.5% yr⁻¹ in the 2010s.

### 3.1.2 Partitioning among the atmosphere, ocean and land

The growth rate in atmospheric $CO_2$ level increased from $1.8 \pm 0.07$ GtC yr$^{-1}$ in the 1960s to $5.1 \pm 0.02$ GtC yr$^{-1}$ during 2010-2019 with important decadal variations (Table 6 and Fig. 3). Both ocean and land $CO_2$ sinks have increased roughly in line with the atmospheric increase, but with significant decadal variability on land (Table 6 and Fig. 6), and possibly in the ocean (Fig. 7).

The ocean $CO_2$ sink increased from $1.0 \pm 0.3$ GtC yr$^{-1}$ in the 1960s to $2.5 \pm 0.6$ GtC yr$^{-1}$ during 2010-2019, with interannual variations of the order of a few tenths of GtC yr$^{-1}$ generally showing an increased ocean sink during large El Niño events (i.e. 1997-1998) (Fig. 7; Rödenbeck et al., 2014, Hauck et al., 2020). The GOBMs show the same patterns of decadal variability as the mean of the pCO$_2$-based flux products, but of weaker magnitude (Section 2.4.3 and Fig. 7; DeVries et al., 2019, Hauck et al., 2020). The pCO$_2$-based flux products and the ocean inverse model highlight different regions as the main origin of this decadal variability, with the pCO$_2$-based flux products placing more of the weakening trend in the Southern Ocean and the ocean inverse model suggesting that more of the weakening trend occurred in the North Atlantic and North Pacific (DeVries et al., 2019). Both approaches show also decadal trends in the low-latitude oceans (DeVries et al., 2019).

Although all individual GOBMs and data-products fall within the observational constraint, the ensemble means of GOBMs and data-products adjusted for the riverine flux diverge over time with a mean offset of 0.15 GtC yr$^{-1}$ in the 1990s to 0.55 GtC yr$^{-1}$ in the decade 2010-2019 and $\geq 0.70$ GtC yr$^{-1}$ since 2017. The GOBMs best estimate of $S_{OCEAN}$ over the period 1994-2007 is $2.1 \pm 0.5$ GtC yr$^{-1}$ and is in agreement with the ocean interior estimate of $2.2 \pm 0.4$ GtC yr$^{-1}$ when taking into account the interior ocean carbon changes of $2.6 \pm 0.3$ GtC yr$^{-1}$ due to the increase of atmospheric $CO_2$ and $-0.4 \pm 0.24$ GtC yr$^{-1}$ due to anthropogenic climate change and variability effects on the natural $CO_2$ flux (Gruber et al., 2019) to match the definition of $S_{OCEAN}$ used here (Hauck et al., 2020). The discrepancy between GOBMs and data-products stems from the southern and northern extratropics prior to 2005, and mostly from the Southern Ocean since the mid-2000s. Possible explanations for the discrepancy in the Southern Ocean could be missing winter observations or uncertainties in the regional river flux adjustment (see section 3.2.3.1, Hauck et al., 2020).

The terrestrial $CO_2$ sink increased from 1.3 ± 0.4 GtC yr$^{-1}$ in the 1960s to 3.4 ± 0.9 GtC yr$^{-1}$
during 2010-2019, with important interannual variations of up to 2 GtC yr$^{-1}$ generally
showing a decreased land sink during El Niño events (Fig. 6), responsible for the
corresponding enhanced growth rate in atmospheric $CO_2$ concentration. The larger land $CO_2$
sink during 2010-2019 compared to the 1960s is reproduced by all the DGVMs in response
to the combined atmospheric $CO_2$ increase and the changes in climate, and consistent with
constraints from the other budget terms (Table 5).
The total atmosphere-to-land fluxes ($S_{LAND} - E_{LUC}$), calculated here as the difference between
$S_{LAND}$ from the DGVMs and $E_{LUC}$ from the bookkeeping models, increased from a 0.2 ± 0.9
GtC yr$^{-1}$ source in the 1960s to a 1.9 ± 1.1 GtC yr$^{-1}$ sink during 2010-2019 (Table 5). Estimates
of total atmosphere-to-land fluxes ($S_{LAND} - E_{LUC}$) from the DGVMs alone are consistent with
our estimate and also with the global carbon budget constraint ($E_{FOS}-G_{ATM}-S_{OCEAN}$, Table 5).
Over the last decade, the land use emission estimate from the DGVMs is significantly larger
than the bookkeeping estimate, mainly explaining why the DGVMs total atmosphere-to-land
flux estimate is lower than the other estimates.
**3.1.3    Model evaluation**
The evaluation of the ocean estimates (Fig. B1) shows an RMSE from annually detrended
data of 0.5 to 1.6 µatm for the five pCO$_2$-based flux products over the globe, relative to the
fCO$_2$ observations from the SOCAT v2020 database for the period 1985-2019. The GOBM
RMSEs are larger and range from 3.5 to 6.9 µatm. The RMSEs are generally larger at high
latitudes compared to the tropics, for both the flux products and the GOBMs. The five flux
products have RMSEs of 0.4 to 1.9 µatm in the tropics, 0.6 to 1.9 µatm in the north, and 1.5
to 2.8 µatm in the south. Note that the flux products are based on the SOCAT v2020
database, hence the latter are no independent data set for the evaluation of the flux
products. The GOBM RMSEs are more spread across regions, ranging from 2.7 to 4.0 µatm
in the tropics, 3.1 to 7.3 µatm in the North, and 6.6 to 11.4 µatm in the South. The higher
RMSEs occur in regions with stronger climate variability, such as the northern and southern
high latitudes (poleward of the subtropical gyres).
The evaluation of the DGVMs (Fig. B2) shows generally high skill scores across models for
runoff, and to a lesser extent for vegetation biomass, GPP, and ecosystem respiration (Fig.
B2, left panel). Skill score was lowest for leaf area index and net ecosystem exchange, with a

widest disparity among models for soil carbon. Further analysis of the results will be provided separately, focusing on the strengths and weaknesses in the DGVM ensemble and its validity for use in the global carbon budget.

The evaluation of the atmospheric inversions (Fig. B3) shows long-term mean biases in the free troposphere lower than 0.4 ppm in absolute values for each product. These biases show some dependency on latitude and are different for each inverse model, which may reveal biases in the surface fluxes (Gaubert et al., 2019, Houweling et al., 2015). Despite tracking surface and in-situ $CO_2$ observations, the systems reproduce NOAA's global annual $CO_2$ growth rate (Section 2.3.1) with mixed skill: where decadal biases are typically small for all systems (<0.08 ppm/yr), interannual differences are larger (1-$\sigma$: 0.10-0.25 ppm/yr, N=19 years) but can be as large as 0.6 ppm/yr for the model/year with worst performance on this metric.

### 3.1.4  Budget imbalance

The carbon budget imbalance ($B_{IM}$; Eq. 1) quantifies the mismatch between the estimated total emissions and the estimated changes in the atmosphere, land and ocean reservoirs. The mean budget imbalance from 1959 to 2019 is small (average of -0.03 GtC yr$^{-1}$) and shows no trend over the full time series. The process models (GOBMs and DGVMs) have been selected to match observational constraints in the 1990s and derived constraints for the 2000s and 2002-2011, but no further constraints have been applied to their representation of trend and variability. Therefore, the near-zero mean and trend in the budget imbalance is an indirect evidence of a coherent community understanding of the emissions and their partitioning on those time scales (Fig. 4). However, the budget imbalance shows substantial variability of the order of ±1 GtC yr$^{-1}$, particularly over semi-decadal time scales, although most of the variability is within the uncertainty of the estimates. The positive carbon imbalance during the 1960s, and early 1990s, suggests that either the emissions were overestimated or the sinks were underestimated during these periods. The reverse is true for the 1980s and late 1990s (Fig. 4).

We cannot attribute the cause of the variability in the budget imbalance with our analysis, we only note that the budget imbalance is unlikely to be explained by errors or biases in the emissions alone because of its large semi-decadal variability component, a variability that is untypical of emissions and has not changed in the past 50 years in spite of a near tripling in

emissions (Fig. 4). Errors in $S_{LAND}$ and $S_{OCEAN}$ are more likely to be the main cause for the
budget imbalance. For example, underestimation of the $S_{LAND}$ by DGVMs has been reported
following the eruption of Mount Pinatubo in 1991 possibly due to missing responses to
changes in diffuse radiation (Mercado et al., 2009) or other yet unknown factors, and
DGVMs are suspected to overestimate the land sink in response to the wet decade of the
1970s (Sitch et al., 2008). Quasi-decadal variability in the ocean sink has also been reported
recently (DeVries et al., 2019, 2017; Landschützer et al., 2015), with all methods agreeing on
a smaller than expected ocean $CO_2$ sink in the 1990s and a larger than expected sink in the
2000s (Fig. 7; DeVries et al., 2019, McKinley et al., 2020). The decadal variability is possibly
caused by changes in ocean circulation (DeVries et al., 2017) not captured in coarse
resolution GOBMs used here (Dufour et al., 2013), but also by external forcing from
decadally varying atmospheric $CO_2$ growth rates and cooling effects through the eruption of
Mount Pinatubo in 1991 which is captured by GOBMs (McKinley et al., 2020).
The decadal variability is thought to be largest in the high latitude ocean regions (poleward
of the subtropical gyres) and the equatorial Pacific (Li and Ilyina, 2018; McKinley et al., 2016,
McKinley et al., 2020). Some of these errors could be driven by errors in the climatic forcing
data, particularly precipitation (for $S_{LAND}$) and wind (for $S_{OCEAN}$) rather than in the models.

### 3.2 Global carbon budget for the last decade (2010 – 2019)

The global carbon budget averaged over the last decade (2010-2019) is shown in Fig. 2 and
Fig. 9 (right panel). For this time period, 86% of the total emissions ($E_{FOS}$ + $E_{LUC}$) were from
fossil $CO_2$ emissions ($E_{FOS}$), and 14% from land-use change ($E_{LUC}$). The total emissions were
partitioned among the atmosphere (46%), ocean (23%) and land (31%), with an
unattributed budget imbalance (-1%).

### 3.2.1 $CO_2$ emissions

Global fossil $CO_2$ emissions grew at a rate of 1.2% $yr^{-1}$ for the last decade (2010-2019), with
a decadal average of 9.4 ± 0.5 GtC $yr^{-1}$ (Fig.5, Table 6). China's emissions increased by +1.2%
$yr^{-1}$ on average (increasing by +0.046 GtC $yr^{-1}$ during the 10-year period) dominating the
global trend, followed by India's emissions increase by +5.1% $yr^{-1}$ (increasing by +0.025 GtC
$yr^{-1}$), while emissions decreased in EU27 by −1.4% $yr^{-1}$ (decreasing by −0.014 GtC $yr^{-1}$), and in
the USA by −0.7% $yr^{-1}$ (decreasing by −0.01 GtC $yr^{-1}$). In the past decade, fossil $CO_2$ emissions
decreased significantly (at the 95% level) in 24 growing economies: Barbados, Belgium,
Croatia, Czech Republic, Denmark, Finland, France, Germany, Israel, Italy, Japan,
Luxembourg, Malta, Mexico, Netherlands, Norway, Romania, Slovakia, Slovenia, Solomon
Islands, Sweden, Switzerland, United Kingdom and the USA. The drivers of recent
decarbonisation are examined in Le Quéré et al. (2019).
In contrast, there is no clear trend in $CO_2$ emissions from land-use change over the last
decade (Fig. 6, Table 6), though the data are very uncertain, with partly diverging trends
over the last decade (Sec. 3.1.1). Larger emissions are expected increasingly over time for
DGVM-based estimates as they include the loss of additional sink capacity, while the
bookkeeping estimates do not. The LUH2-GCB2020 data set also features large dynamics in
land-use in particular in the tropics in recent years, causing higher emissions in DGVMs,
BLUE and the OSCAR best-guess, which includes simulations based on LUH2-GCB2020, than
in H&N2017.
**3.2.2    Partitioning among the atmosphere, ocean and land**
The growth rate in atmospheric $CO_2$ concentration increased during 2010-2019, with a
decadal average of 5.1 ± 0.02 GtC $yr^{-1}$, albeit with large interannual variability (Fig. 4).
Averaged over that decade, the ocean and land sinks amount to 2.5 ± 0.6 GtC $yr^{-1}$ and 3.4 ±
0.9 GtC $yr^{-1}$ respectively. During 2010-2017, the ocean $CO_2$ sink appears to have intensified
in line with the expected increase from atmospheric $CO_2$ (McKinley et al., 2020). This effect
is stronger in the $pCO_2$-based flux products (Fig. 7, McKinley et al., 2020). The reduction of -
0.16 GtC $yr^{-1}$ (range: -0.43 to +0.03 GtC $yr^{-1}$ ) in the ocean $CO_2$ sink in 2017 is consistent with
the return to normal conditions after the El Niño in 2015/16, which caused an enhanced
sink in previous years.
The budget imbalance (Table 6) and the residual sink from global budget (Table 5) include
an error term due to the inconsistency that arises from using $E_{LUC}$ from bookkeeping
models, and $S_{LAND}$ from DGVMs. This error term includes the fundamental differences
between bookkeeping models and DGVMs, most notably the loss of additional sink capacity.
Other differences include: an incomplete accounting of LUC practices and processes in
DGVMs, while they are all accounted for in bookkeeping models by using observed carbon
densities, and bookkeeping error of keeping present-day carbon densities fixed in the past.
That the budget imbalance shows no clear trend towards larger values over time is an
indication that the loss of additional sink capacity plays a minor role compared to other
errors in $S_{LAND}$ or $S_{OCEAN}$ (discussed in Section 3.1.4).

### 3.2.3 Inter-comparison of flux estimates

#### 3.2.3.1 Regionality

Fig. 8 shows the partitioning of the total atmosphere-to-surface fluxes excluding fossil $CO_2$
emissions ($S_{OCEAN}$ + $S_{LAND}$ − $E_{LUC}$) according to the multi-model average estimates from
process models (GOBMs and DGVMs), atmospheric inversions and ocean $pCO_2$-based
products. Fig. 8 provides information on the regional distribution of those fluxes by latitude
bands. The global mean total atmosphere-to-surface $CO_2$ fluxes from process models for
2010-2019 is 3.8 ± 0.7 GtC yr$^{-1}$, below the global mean atmosphere-to-surface flux of 4.3 ±
0.5 GtC yr$^{-1}$ inferred by the carbon budget ($E_{FOS}$ − $G_{ATM}$ in Equation 1; Table 6). The total
atmosphere-to-surface $CO_2$ fluxes from the inversions (4.5 ± 0.1 GtC yr$^{-1}$) almost matches
the value inferred by the carbon budget, which is expected due to the constraint on $G_{ATM}$
incorporated within the inversion approach and the adjustment of the fossil emissions prior
to a value consistent with the $E_{FOS}$ budget term (Jones et al., 2020; See Section 2.6).
In the southern extratropics (south of 30°S), the atmospheric inversions suggest a total
atmosphere-to-surface sink ($S_{OCEAN}$+$S_{LAND}$-$E_{LUC}$) for 2010-2019 of 1.4 ± 0.3 GtC yr$^{-1}$, similar to
the process models' estimate of 1.4 ± 0.3 GtC yr$^{-1}$ (Fig. 8). An approximately neutral total
land flux ($S_{LAND}$-$E_{LUC}$) for the southern extratropics is estimated by both the DGVMs (0.0 ± 0.1
GtC yr$^{-1}$) and the inversion models (sink of 0.1 ± 0.2 GtC yr$^{-1}$). The GOBMs (1.4 ± 0.3 GtC yr$^{-1}$)
produce a lower estimate for the ocean sink than the inversion models (1.6 ± 0.2 GtC yr$^{-1}$) or
$pCO_2$-based flux products (1.7 ± 0.1 GtC yr$^{-1}$; discussed further below).
In the tropics (30°S-30°N), both the atmospheric inversions and process models suggest that
the total carbon balance in this region ($S_{OCEAN}$+$S_{LAND}$-$E_{LUC}$) is close to neutral over the past
decade. The inversion models indicate a small tropical source to the atmosphere of -0.2 ±
0.6 GtC yr$^{-1}$, whereas the process models indicate a small sink of 0.2 ± 0.7 GtC yr$^{-1}$. The
GOBMs (-0.1 ± 0.2 GtC yr$^{-1}$ source), inversion models (-0.1 ± 0.2 GtC yr$^{-1}$ source) and $pCO_2$-
based flux products (-0.05 ± 0.02 GtC yr$^{-1}$ source) all indicate an approximately neutral
tropical ocean flux, meaning that the difference in sign of the total fluxes stems from the
land component. Indeed, the DGVMs indicate a total land sink ($S_{LAND}$-$E_{LUC}$) of 0.2 ± 0.7 GtC
yr$^{-1}$, whereas the inversion models indicate a small land source of -0.1 ± 0.7 GtC yr$^{-1}$, though
with high uncertainty in both cases. Overall, the GOBMs, pCO$_2$-based flux products and
inversion models suggest either a neutral ocean flux or a small ocean source, while the
DGVMs and inversion models suggest either a small sink or source on land. The agreement
between inversions and process models is significantly better for the last decade than for
any previous decade (Fig. 8), although the reasons for this better agreement are still
unclear.
In the northern extratropics (north of 30°N) the atmospheric inversions suggest an
atmosphere-to-surface sink (S$_{OCEAN}$+S$_{LAND}$-E$_{LUC}$) for 2010-2019 of 2.9 ± 0.6 GtC yr$^{-1}$, which is
higher than the process models' estimate of 2.3 ± 0.6 GtC yr$^{-1}$ (Fig. 8). The difference derives
from the total land flux (S$_{LAND}$-E$_{LUC}$) estimate, which is 1.1 ± 0.6 GtC yr$^{-1}$ in the DGVMs
compared with 1.7 ± 0.8 GtC yr$^{-1}$ in the inversion models. The GOBMs (1.2 ± 0.2 GtC yr$^{-1}$),
inversion models (1.2 ± 0.2 GtC yr$^{-1}$) and pCO$_2$-based flux products (1.2 ± 0.2 GtC yr$^{-1}$)
produce consistent estimates of the ocean sink.
The noteworthy differences between the annual estimates produced by different data
sources are as follows:
(i) the southern S$_{OCEAN}$ flux in the pCO$_2$-based flux products and inversion models is
higher than in the GOBMs. This might be explained by the data-products potentially
underestimating the winter CO$_2$ outgassing south of the Polar Front (Bushinsky et al.,
2019), or by the uncertainty in the regional distribution of the river flux adjustment
(Aumont et al., 2001, Lacroix et al., 2020) applied to pCO$_2$-based flux products to
isolate the anthropogenic S$_{OCEAN}$ flux.
(ii) the larger magnitude of the northern net land flux (S$_{LAND}$-E$_{LUC}$) in inversion models
than in the DGVMs. Discrepancies in the northern and tropical land fluxes conforms
with persistent issues surrounding the quantification of the drivers of the global net
land CO$_2$ flux (Arneth et al., 2017; Huntzinger et al., 2017) and the distribution of
atmosphere-to-land fluxes between the tropics and high northern latitudes (Baccini
et al., 2017; Schimel et al., 2015; Stephens et al., 2007; Ciais et al. 2019). These
differences cannot be simply explained. They could either reflect a bias in the
inversions or missing processes or biases in the process models, such as the lack of

adequate parameterizations for land management for the DGVMs. In fact, the 6

inversions shown in Fig. 8 form two categories, one with a large northern land sink

and a tropical land source and another with a moderate northern land sink and a

small tropical sink (3.2.3.3). The estimated contribution of the north and its

uncertainty from process models is sensitive both to the ensemble of process

models used e.g. the inclusion of northern forest management in DGVMs and

possibly too strong emissions from LUC (Bastos et al. 2020), and to the specifics of

each inversion e.g. zonal and latitudinal transport and its covariance with seasonal

fluxes (Denning et al. 1995).

### 3.2.3.2  Interannual Variability

The interannual variability in the southern extratropics is low because of the dominance of
ocean area with low variability compared to land areas. The split between land ($S_{LAND}-E_{LUC}$)
and ocean ($S_{OCEAN}$) shows a small contribution to variability in the south coming from the
land, with no consistency between the DGVMs and the inversions or among inversions. This
is expected due to the difficulty of separating exactly the land and oceanic fluxes when
viewed from atmospheric observations alone. The interannual variability, calculated as the
standard deviation from detrended time-series around the mean, was found to be similar in
the $pCO_2$-based flux products including Watson et al (0.05 to 0.10 GtC yr−1) and GOBMs
(0.06 to 0.17 GtC yr−1) in 2010-2019 (Fig. B1).
Both the process models and the inversions consistently allocate more year-to-year
variability of $CO_2$ fluxes to the tropics compared to the northern extratropics (Fig. 8). The
land is the origin of most of the tropical variability, consistently among the process models
and inversions. The interannual variability in the tropics is similar among the ocean flux
products (0.03 to 0.09 GtC yr−1) and the models (0.02 to 0.09 GtC yr−1; Sect. 3.1.3, Fig. B1).
The inversions indicate that atmosphere-to-land $CO_2$ fluxes are more variable than
atmosphere-to-ocean $CO_2$ fluxes in the tropics, and produce slightly higher IAV than the
ocean flux products or GOBMs. With a sparsity of tropical atmospheric measurements, an
aliasing of the large land flux variations onto the tropical ocean fluxes in the inversions is
one likely cause of this difference.

In the northern extratropics, the models, inversions, and $pCO_2$-based flux products

consistently suggest that most of the variability stems from the land (Fig. 8). Inversions,

GOBMs, and $pCO_2$-based flux products agree on the mean of $S_{OCEAN}$, but with a higher

interannual variability in the $pCO_2$-based flux products (0.05 to 0.08 GtC yr−1) than in the

GOBMs (0.04 to 0.10 GtC yr−1, Fig. B1).

### 3.2.3.3  *Atmospheric inversion models differences*

The expanded ensemble of atmospheric inversions (from N=3 to N=6) allows to have a more

representative sample of model-model differences e.g. in latitudinal transport and other

inversion settings (Table A4). When assessed for their tropical/northern land+ocean fluxes

we see a dipole arise, where three models estimate a Northern extratropical sink close to

2.5 GtC/year, and the other three a sink of close to 3.5 GtC/year. The inversions resulting in

a large Northern sink estimate also a tropical source. Both groups of models perform equally

well on the evaluation metric of the misfit of optimized $CO_2$ from inversions against

independent aircraft data in Fig B3 though, and resolving this difference will require the

consideration and inclusion of larger volumes of semi-continuous observations of

concentrations, fluxes as well as auxiliary variables collected from (tall) towers close to the

surface $CO_2$ exchange. Improvements in model resolution and atmospheric transport

realism together with expansion of the observational record (also in the data sparse Boreal

Eurasian area) may help anchor the mid-latitude NH fluxes per continent. In addition, new

metrics could potentially differentiate between the more and less realistic realisations of

the Northern Hemisphere land sink shown in Fig.8.

In previous versions of this publication, another hypothesised explanation was that

differences in the prior dataset used by the inversion models, and related adjustments to

posterior estimates, drove inter-model disparity. However, separate analysis has shown that

the influence of the chosen prior land and ocean fluxes is minor compared to other aspects

of each inversion, and the majority (5 of 6) of the inversion models presented in this update

now use a consistent prior for fossil emissions (Jones et al., 2020; see Section 2.6).

Finally, in the 2020 effort, two inverse systems (UoE and CAMS) used column $CO_2$ products

derived from GoSAT and OCO-2, respectively. Their estimated fluxes and performance on

the metrics evaluated in this work were similar to their counterparts driven by in-situ and

flask observations, and hence these solutions were not included separately (as noted by
Chevallier et al., 2019). Nevertheless, this convergence of solutions is an important
prerequisite for the use of longer remote sensing $CO_2$ time series in the future, and could
help to further study differences driven by observational coverage and/or sparseness of the
current network. Also, column-$CO_2$ products are likely to be less sensitive to vertical
transport differences between models, believed to be a remaining source of uncertainty
(Basu et al., 2018).

### 3.2.4 Budget imbalance

The budget imbalance ($B_{IM}$) was low, -0.1 GtC yr$^{-1}$ on average over 2010-2019, although the
$B_{IM}$ uncertainty is large (1.4 GtC yr$^{-1}$ over the decade). Also, the $B_{IM}$ shows significant
departure from zero on yearly time scales (Fig. 4), highlighting unresolved variability of the
carbon cycle, likely in the land sink ($S_{LAND}$), given its large year to year variability (Fig. 4e and
6b), while the decadal variability could originate from both the land and ocean sinks, given
unresolved discussions on the strength of the ocean carbon sink (Bushinsky et al., 2019;
Watson et al., 2020) and its decadal variability (DeVries et al., 2019).
Although the budget imbalance is near zero for the recent decades, it could be due to
compensation of errors. We cannot exclude an overestimation of $CO_2$ emissions, in
particular from land-use change, given their large uncertainty, as has been suggested
elsewhere (Piao et al., 2018), combined with an underestimate of the sinks. A larger $S_{LAND}$
would reconcile model results with inversion estimates for fluxes in the total land during the
past decade (Fig. 8; Table 5). Likewise, a larger $S_{OCEAN}$ is also possible given the higher
estimates from the data-products (see section 3.1.2, Fig. 7 and 8) and the recently
suggested upward correction of the ocean carbon sink (Watson et al., 2020, Fig. 7). If data-
products with the Watson et al adjustment were to be used instead of GOBMs to estimate
$S_{OCEAN}$, this would result in a $B_{IM}$ on the order of -1 GtC yr$^{-1}$ indicating that a closure of the
budget could only be achieved with either anthropogenic emissions being larger and/or the
net land sink being smaller than estimated here.
More integrated use of observations in the Global Carbon Budget, either on their own or for
further constraining model results, should help resolve some of the budget imbalance
(Peters et al., 2017; Section 4).

**3.3 Global carbon budget for year 2019**

**3.3.1 CO$_2$ emissions**

Preliminary estimates of global fossil CO$_2$ emissions are for growth of only 0.1% between 2018 and 2019 to remain at 9.7 ± 0.5 GtC in 2019 (Fig. 5), distributed among coal (39%), oil (34%), natural gas (21%), cement (4%) and others (1.5%). Compared to the previous year, emissions from coal decreased by 1.8%, while emissions from oil, natural gas, and cement increased by 0.8%, 2.0%, and 3.2%, respectively. All growth rates presented are adjusted for the leap year, unless stated otherwise.

In 2019, the largest absolute contributions to global fossil CO$_2$ emissions were from China (28%), the USA (14%), the EU (27-member states; 8%), and India (7%). These four regions account for 57% of global CO$_2$ emissions, while the rest of the world contributed 43% which includes aviation and marine bunker fuels (3.5% of the total). Growth rates for these countries from 2018 to 2019 were +2.2% (China), -2.6% (USA), –4.5% (EU27), and +1.0% (India), with +1.8% for the rest of the world. The per-capita fossil CO$_2$ emissions in 2019 were 1.3 tC person$^{-1}$ yr$^{-1}$ for the globe, and were 4.4 (USA), 1.9 (China), 1.8 (EU27) and 0.5 (India) tC person$^{-1}$ yr$^{-1}$ for the four highest emitting countries (Fig. 5).

The growth in emissions of 0.1% in 2019 is within the range of the projected growth of 0.6% (range of -0.2 to 1.5%) published in Friedlingstein et al. (2019) based on national emissions projections for China, the USA, the EU27, and India and projections of gross domestic product corrected for I$_{FOS}$ trends for the rest of the world. The growth in emissions in 2019 for China, the USA, EU27, India, and the rest of the world were all within their previously projected range (Table 7).

The largest absolute contributions to global CO$_2$ emissions from a consumption perspective were China (25%), USA (16%), the EU (10%), and India (6%) for 2016, the last year with available data. The difference between territorial and consumption emissions (the net emission transfer via international trade) has generally increased from 1990 to around 2005 and remained relatively stable afterwards until the last year available (2016; Fig. 5).

The global CO$_2$ emissions from land-use change are estimated as 1.8 ± 0.7 GtC in 2019, slightly larger than  the previous decade, which results in particular from the high peat and tropical deforestation/degradation fires. First, unusually dry conditions for a non-El Niño year occurred in Indonesia in 2019, which led to fire emissions from peat burning,

deforestation and degradation in equatorial Asia to be about twice as large as the average
over the previous decade (GFED4.1s, van der Werf et al., 2017). Second, 2019 saw a surge
in deforestation fires in the Amazon, causing about 30% higher emissions from
deforestation and degradation fires over the previous decade (GFED4.1s, van der Werf et
al., 2017). This development was evident also in deforestation rates, where 2019 (August
2018-July 2019), with 10.1 $km^2$ forest clear-cut, saw the highest rate since 2008 (INPE,
2020). However, confidence in the annual change remains low. This brings the total $CO_2$
emissions from fossil plus land-use change ($E_{FOS}+E_{LUC}$) to 11.5 ± 0.9 GtC (42.2 ± 3.3 $GtCO_2$).
**3.3.2   Partitioning among the atmosphere, ocean and land**
The growth rate in atmospheric $CO_2$ concentration corresponded to  5.4 ± 0.2 GtC in 2019
(2.54 ± 0.08 ppm; Fig. 4; Dlugokencky and Tans, 2020), slightly above the 2010-2019 average
of 5.1 ± 0.02 GtC $yr^{-1}$.
The estimated ocean $CO_2$ sink was 2.6 ± 0.6 GtC in 2019. Although there is a significant
difference of $S_{OCEAN}$ between GOBMs (2.6 GtC) and $pCO_2$-based products (3.4 GtC), they
both suggest an average increase of 0.06-0.07 GtC in 2019 compared to 2018.  Six models
and two flux products show an increase of $S_{OCEAN}$ (GOBM up to +0.30 GtC, data-product up
to +0.29 GtC), while three models and two flux products show no change or a decrease of
$S_{OCEAN}$ (GOBMs down to -0.03 GtC, data-products down to -0.17 GtC; Fig. 7).
The terrestrial $CO_2$ sink from the DGVM model ensemble was 3.1 ± 1.2 GtC in 2019, slightly
below the decadal average (Fig. 4) and consistent with constraints from the rest of the
budget (Table 5). Atmospheric inversions confirm a lower-than-average land sink in 2019,
and consistently estimate this as an increased source from the tropical land (+0.3 GtC). The
budget imbalance was +0.3 GtC in 2019, which is above the average over the last decade
(Table 6). This imbalance is indicative only, given its significant year to year variability and
large uncertainty (1.4 GtC $yr^{-1}$).
**3.4   Global carbon budget projection for year 2020**
**3.4.1   Fossil $CO_2$ emissions**
We present the results from the four separate methods in Table A8, with monthly results
for each country, region, and globally shown in Figure B5. The restrictions implemented in
response to COVID-19 led to dramatic and unprecedented changes in society, and this
caused large changes in $CO_2$ emissions. All countries had significant deviations from their
previous emission trends.

### 3.4.1.1  Year To Date (YTD)

The four methods presented here use a mix of direct emissions estimates from energy
consumption data to the use of proxies as indicators of changes in activity levels. Annual
historical $CO_2$ emissions estimates (pre-2020) are largely derived from reported energy data.
For 2020, we do not have sufficient information to say that the use of monthly energy data
gives any more accurate estimates than proxy approaches. Monthly energy consumption
data are subject to revisions and can be estimated or incomplete, and it is not known if
proxy data may perform better. A full evaluation of monthly and proxy methods can only be
made when full year data comes available. As noted in Forster et al (2020) the reductions in
$CO_2$ emissions may be about 20% overestimated based on meteorologically adjusted NOx
observations.
The YTD results (Fig. B5, Table A8) run to September for all regions and methods, except the
EU27 which is to July (limited by the Eurostat data used by the GCB method). To September
(July) 2020, the four methods indicate fossil $CO_2$ emissions were down in all regions and
globally. However, the background for these declines varies by countries. The EU and the US
had declining emission trends before COVID-19, so the pandemic effect is on top of these
existing emission reductions. In both the EU and the US, reductions in coal use have been
accelerated by COVID-19. Similarly, India's emissions were in decline through 2019, but this
time because of economic troubles (Andrew, 2020b), but COVID-19 is potentially
superimposed on the longer term trend of increasing emissions in India. In contrast, China
and the Rest of the World have the COVID-19 effect on the top of rising emissions. China
has lower reductions, but this may also indicate that the full impact of the COVID-19
restrictions occurred earlier and the economy has had a longer time to recover.
Based on the three studies providing sufficient data, from January to September, global
emissions may have declined around 8% (median, based on model estimates of -7.6% UEA, -
7.6% Carbon Monitor, -14.1% Priestley Centre). This range between estimates does not
include the uncertainty inherent in each method, which would increase the spread.

### 3.4.1.2 2020 projections

The full-year projection for 2020 must necessarily be interpreted cautiously. Only Le Quéré et al (2020) include a formal projection, by assuming confinement measures in place on 13 November remain in place until the end of the year at current or lower levels in each country. Forster et al (2020) use a simple extrapolation, assuming the declines in emissions from their baselines remain at 66% of the level over the last 30 days with estimates. Liu et al (2020) and the GCB method did not perform a projection for 2020, and for purposes of comparison we use a simple approach to extrapolating their observations by assuming the remaining months of the year change by the same relative amount compared to 2019 as the final month of observations.

Based on these assumptions, the countries and regions considered are all expected to see a decline in annual total emissions, with the potential exception of China which may have a slight increase according to Carbon Monitor and the GCB method (Fig. B5). The year 2020 is behaving in many ways entirely differently to any year in history, and the confidence in the 2020 projection is therefore currently low, due to both the spread in results and the uncertain developments of the disease itself, strength of future societal and industrial restrictions, and stimulus packages throughout the remainder of 2020. The largest source of uncertainty comes from the emissions in China, because of the limited available information both on monthly emissions and for proxy data, and emissions for the RoW, because it represents around 40% of the world's emissions in aggregate.

Based on the median value of the four methods considered, global emissions may decline by about 7% in 2020 (-5.8% GCB, -6.5% Carbon Monitor, -6.9% (range -2.7 to -10.8%) UEA, -13.0% Priestley Centre), with additional uncertainty from each method on top of this (Fig B5, Table A8). Using a purely GDP-based projection, based on the IMF GDP forecast as of June 2020, and assuming the 10-year trend in $CO_2$/GDP continues in 2020, emissions would decline 7.5% – well within the range of other estimates. In October 2020, the IEA forecasted a drop of 7% in fossil energy emissions (IEA, 2020). The decrease in emissions for the full year 2020 appears more pronounced in the US, EU27 and India, partly due to pre-existing trends. In contrast the decrease in emissions appears least pronounced in China, where restrictions measures associated with COVID-19 occurred early in the year and lockdown measures were more limited in time.

### 3.4.1.3  Synthesis

Given a negative median growth rate of about -7% across methods, global fossil $CO_2$ emissions ($E_{FOS}$) in 2020 would be around 9.0 GtC (33.2 $GtCO_2$) in 2020 (Table A8). These figures do not include the uncertainty from this method in projecting 2020 emissions.

Our preliminary estimates of fire emissions in deforestation zones and Amazon deforestation rates indicate that emissions from land-use change ($E_{LUC}$) for 2020 are similar to the 2010-2019 average (Sec. 2.2.4). We therefore expect $E_{LUC}$ emissions of around 1.6 GtC in 2020. The apparent decrease in the mean value of $E_{LUC}$ emissions compared to 2019 is largely related to the transition from an anomalously dry to a wet year in Indonesia (see Section 2.2.4 and 3.2.1 for detail).

We hence project global total anthropogenic $CO_2$ emissions from fossil and land use changes to be around 10.6 GtC (39 $GtCO_2$) in 2020.

### 3.4.2  Partitioning among the atmosphere, ocean and land

The 2020 growth in atmospheric $CO_2$ concentration ($G_{ATM}$) is projected to be about 5.3 GtC (2.5 ppm) based on GLO observations until the end of August 2020, bringing the atmospheric $CO_2$ concentration to an expected level of 412 ppm averaged over the year. Combining projected $E_{FOS}$, $E_{LUC}$ and $G_{ATM}$ suggests a combined land and ocean sink ($S_{LAND}$ + $S_{OCEAN}$) of about 5.3 GtC for 2020. Although each term has large uncertainty, the oceanic sink $S_{OCEAN}$ has generally low interannual variability and is likely to remain close to its 2019 value of around 2.6 GtC, leaving a rough estimated land sink $S_{LAND}$ (including any budget imbalance) of around 2.7 GtC, slightly below the 2019 estimate.

### 3.5  Cumulative sources and sinks

Cumulative historical sources and sinks are estimated as in Eq. (1) with semi-independent estimates for each term and a global carbon budget imbalance. Cumulative fossil $CO_2$ emissions for 1850-2019 were 445 ± 20 GtC for $E_{FOS}$ and 210 ± 60 GtC for $E_{LUC}$ (Table 8; Fig. 9), for a total of 650 ± 65 GtC. The cumulative emissions from $E_{LUC}$ are particularly uncertain, with large spread among individual estimates of 150 GtC (H&N2017), 275 GtC (BLUE), and 200 GtC (OSCAR) for the three bookkeeping models and a similar wide estimate of 200 ± 60 GtC for the DGVMs. These estimates are consistent with indirect constraints from

vegetation biomass observations (Li et al., 2017), but given the large spread a best estimate
is difficult to ascertain.
Emissions during the period 1850-2019 were partitioned among the atmosphere (265 ± 5
GtC; 40%), ocean (160 ± 20 GtC; 25%), and the land (210 ± 55 GtC; 32%). This cumulative
land sink is broadly equal to the cumulative land-use emissions, making the global land near
neutral over the 1850-2019 period. The use of nearly independent estimates for the
individual terms shows a cumulative budget imbalance of 20 GtC (3%) during 1850-2019
(Fig. 2), which, if correct, suggests that emissions are too high by the same proportion or
that the land or ocean sinks are underestimated. The bulk of the imbalance could originate
from the estimation of large $E_{LUC}$ between the mid 1920s and the mid 1960s which is
unmatched by a growth in atmospheric $CO_2$ concentration as recorded in ice cores (Fig. 3).
The known loss of additional sink capacity of 30-40 GtC due to reduced forest cover has not
been accounted for in our method and would further exacerbate the budget imbalance
(Section 2.7.4).
Cumulative emissions through to year 2020 increase to 655 ± 65 GtC (2340 ± 240 $GtCO_2$),
with about 70% contribution from $E_{FOS}$ and about 30% contribution from $E_{LUC}$. Cumulative
emissions and their partitioning for different periods are provided in Table 8.
Given the large and persistent uncertainties in historical cumulative emissions, we suggest
extreme caution is needed if using this estimate to determine the remaining cumulative $CO_2$
emissions consistent with an ambition to stay below a given temperature limit (Millar et al.,
2017; Rogelj et al., 2016, 2019).
**4    Discussion**
Each year when the global carbon budget is published, each flux component is updated for
all previous years to consider corrections that are the result of further scrutiny and
verification of the underlying data in the primary input data sets. Annual estimates may be
updated with improvements in data quality and timeliness (e.g. to eliminate the need for
extrapolation of forcing data such as land-use). Of all terms in the global budget, only the
fossil $CO_2$ emissions and the growth rate in atmospheric $CO_2$ concentration are based
primarily on empirical inputs supporting annual estimates in this carbon budget. The carbon
budget imbalance, yet an imperfect measure, provides a strong indication of the limitations
in observations in understanding and representing processes in models, and/or in the
integration of the carbon budget components.
The persistent unexplained variability in the carbon budget imbalance limits our ability to
verify reported emissions (Peters et al., 2017) and suggests we do not yet have a complete
understanding of the underlying carbon cycle dynamics. Resolving most of this unexplained
variability should be possible through different and complementary approaches. First, as
intended with our annual updates, the imbalance as an error term is reduced by
improvements of individual components of the global carbon budget that follow from
improving the underlying data and statistics and by improving the models through the
resolution of some of the key uncertainties detailed in Table 9. Second, additional clues to
the origin and processes responsible for the variability in the budget imbalance could be
obtained through a closer scrutiny of carbon variability in light of other Earth system data
(e.g. heat balance, water balance), and the use of a wider range of biogeochemical
observations to better understand the land-ocean partitioning of the carbon imbalance (e.g.
oxygen, carbon isotopes). Finally, additional information could also be obtained through
higher resolution and process knowledge at the regional level, and through the introduction
of inferred fluxes such as those based on satellite $CO_2$ retrievals. The limit of the resolution
of the carbon budget imbalance is yet unclear, but most certainly not yet reached given the
possibilities for improvements that lie ahead.
Estimates of global fossil $CO_2$ emissions from different datasets are in relatively good
agreement when the different system boundaries of these datasets are taken into account
(Andrew, 2020a). But while estimates of $E_{FOS}$ are derived from reported activity data
requiring much less complex transformations than some other components of the budget,
uncertainties remain, and one reason for the apparently low variation between datasets is
precisely the reliance on the same underlying reported energy data.  This year we have
added cement carbonation, a carbon sink, to $E_{FOS}$.The budget excludes some sources of
fossil $CO_2$ emissions, which available evidence suggests are relatively small (<1%). In non-
Annex I countries, and before 1990 in Annex I countries, we still omit emissions from
carbonate decomposition apart from those in cement production, a focus of future updates.
We have also included new estimates for India, which are now for the calendar year instead
of its fiscal year and include the significant changes in coal stocks missing from other
datasets. Estimates for Japan and Australia, two other large emitters, are still reported for
fiscal years not aligned with the calendar year. Some errors in pre-1950 emissions were
uncovered by Andrew (2020a), and these have been corrected this year.
Estimates of $E_{LUC}$ suffer from a range of intertwined issues, including the poor quality of
historical land-cover and land-use change maps, the rudimentary representation of
management processes in most models, and the confusion in methodologies and boundary
conditions used across methods (e.g. Arneth et al., 2017; Pongratz et al., 2014, see also
Section 2.7.4 on the loss of sink capacity). Uncertainties in current and historical carbon
stocks in soils and vegetation also add uncertainty in the LUC flux estimates. Unless a major
effort to resolve these issues is made, little progress is expected in the resolution of $E_{LUC}$.
This is particularly concerning given the growing importance of $E_{LUC}$ for climate mitigation
strategies, and the large issues in the quantification of the cumulative emissions over the
historical period that arise from large uncertainties in $E_{LUC}$.
The assessment of the GOBMs used for $S_{OCEAN}$ with flux products based on observations
highlights substantial discrepancy in the Southern Ocean (Figure 8, Hauck et al., 2020). The
long-standing sparse data coverage of $pCO_2$ observations in the Southern compared to the
Northern Hemisphere (e.g. Takahashi et al., 2009) continues to exist (Bakker et al., 2016,
2020) and to lead to substantially higher uncertainty in the  $S_{OCEAN}$ estimate for the Southern
Hemisphere (Watson et al., 2020). This discrepancy points to the need for increased high-
quality $pCO_2$ observations especially in the Southern Ocean. Further uncertainty stems from
the regional distribution of the river flux adjustment term being based on one model study
yielding the largest riverine outgassing flux south of 20°S (Aumont et al., 2001), with a
recent study questioning this distribution (Lacroix et al., 2020). The data-products suggest
an underestimation of variability in the GOBMs globally and consequently, the variability in
$S_{OCEAN}$ appears to be underestimated. The size of the underestimation of the amplitude of
interannual variability (order of <0.1 GtC yr$^{-1}$, A-IAV, see Fig. B1) could account for some of
the budget imbalance, but not all.
The assessment of the net land-atmosphere exchange derived from land sink and net land-
use change flux with atmospheric inversions also shows substantial discrepancy, particularly
for the estimate of the total land flux over the northern extra-tropics in the past decade.
This discrepancy highlights the difficulty to quantify complex processes ($CO_2$ fertilisation,
nitrogen deposition, N fertilisers, climate change and variability, land management, etc.)
that collectively determine the net land $CO_2$ flux. Resolving the differences in the Northern
Hemisphere land sink will require the consideration and inclusion of larger volumes of
observations (Section 3.2.3).
As introduced in 2018, we provide metrics for the evaluation of the ocean and land models
and the atmospheric inversions. These metrics expand the use of observations in the global
carbon budget, helping 1) to support improvements in the ocean and land carbon models
that produce the sink estimates, and 2) to constrain the representation of key underlying
processes in the models and to allocate the regional partitioning of the $CO_2$ fluxes. However,
GOBMs have changed little since the introduction of the ocean model evaluation. This is an
initial step towards the introduction of a broader range of observations that we hope will
support continued improvements in the annual estimates of the global carbon budget.
We assessed before that a sustained decrease of $-1\%$ in global emissions could be detected
at the 66% likelihood level after a decade only (Peters et al., 2017). Similarly, a change in
behaviour of the land and/or ocean carbon sink would take as long to detect, and much
longer if it emerges more slowly. To continue reducing the carbon imbalance on annual to
decadal time scales, regionalising the carbon budget, and integrating multiple variables are
powerful ways to shorten the detection limit and ensure the research community can
rapidly identify issues of concern in the evolution of the global carbon cycle under the
current rapid and unprecedented changing environmental conditions.
**5    Conclusions**
The estimation of global $CO_2$ emissions and sinks is a major effort by the carbon cycle
research community that requires a careful compilation and synthesis of measurements,
statistical estimates and model results. The delivery of an annual carbon budget serves two
purposes. First, there is a large demand for up-to-date information on the state of the
anthropogenic perturbation of the climate system and its underpinning causes. A broad
stakeholder community relies on the data sets associated with the annual carbon budget
including scientists, policy makers, businesses, journalists, and non-governmental
organizations engaged in adapting to and mitigating human-driven climate change. Second,
over the last decade we have seen unprecedented changes in the human and biophysical
environments (e.g. changes in the growth of fossil fuel emissions, impact of COVID-19
pandemic, Earth's warming, and strength of the carbon sinks), which call for frequent
assessments of the state of the planet, a better quantification of the causes of changes in
the contemporary global carbon cycle, and an improved capacity to anticipate its evolution
in the future. Building this scientific understanding to meet the extraordinary climate
mitigation challenge requires frequent, robust, transparent and traceable data sets and
methods that can be scrutinized and replicated. This paper via 'living data' helps to keep
track of new budget updates.

## 1747 6   Data availability

The data presented here are made available in the belief that their wide dissemination will
lead to greater understanding and new scientific insights of how the carbon cycle works,
how humans are altering it, and how we can mitigate the resulting human-driven climate
change. The free availability of these data does not constitute permission for publication of
the data. For research projects, if the data are essential to the work, or if an important
result or conclusion depends on the data, co-authorship may need to be considered for the
relevant data providers. Full contact details and information on how to cite the data shown
here are given at the top of each page in the accompanying database and summarised in
Table 2.
The accompanying database includes two Excel files organised in the following
spreadsheets:
File Global_Carbon_Budget_2020v1.0.xlsx includes the following:
1.  Summary
2.  The global carbon budget (1959-2019);
3.  Global $CO_2$ emissions from fossil fuels and cement production by fuel type, and the per-
capita emissions (1959-2019);

4. $CO_2$ emissions from land-use change from the individual methods and models (1959-2019);

5. Ocean $CO_2$ sink from the individual ocean models and $pCO_2$-based products (1959-2019);

6. Terrestrial $CO_2$ sink from the DGVMs (1959-2019);

7.  Additional information on the historical global carbon budget prior to 1959 (1750-2019).

File National_Carbon_Emissions_2020v1.0.xlsx includes the following:

1. Summary

2. Territorial country $CO_2$ emissions from fossil $CO_2$ emissions (1959-2019) from CDIAC with UNFCCC data overwritten where available, extended to 2019 using BP data;

3. Consumption country $CO_2$ emissions from fossil $CO_2$ emissions and emissions transfer from the international trade of goods and services (1990-2016) using CDIAC/UNFCCC data (worksheet 3 above) as reference;

4. Emissions transfers (Consumption minus territorial emissions; 1990-2016);

5. Country definitions;

6. Details of disaggregated countries;

7. Details of aggregated countries.

Both spreadsheets are published by the Integrated Carbon Observation System (ICOS) Carbon Portal and are available at https://doi.org/10.18160/gcp-2020 (Friedlingstein et al., 2020). National emissions data are also available from the Global Carbon Atlas (http://www.globalcarbonatlas.org/, last access: 16 November 2020).

**Author contributions.** PF, MOS, MWJ, CLQ, RMA, JH, GPP, WP, JP, SS, AO, JGC, PC and RBJ

designed the study, conducted the analysis, and wrote the paper. RMA, GPP and JIK

produced the emissions and their uncertainties, the GCB 2020 emission projections, and

analysed the emissions data. DG and GM provided emission data. PPT provided key

atmospheric $CO_2$ data. WP, PC, FC, CR, NC, YN, PIP and LF provided an updated atmospheric

inversion, developed the protocol and produced the evaluation. JP, KH, SB, TG and RAH

provided updated bookkeeping land-use change emissions. LPC, LEOCA, and GRvdW

provided forcing data for land-use change. AA, VH, AKJ, EJ, EK, SL, DLL, JRM, JEMSN, BP, HT,

NV, APW, AJW, WY, XY and SZ provided an update of a DGVM. IH provided the climate
forcing data for the DGVMs. ER provided the evaluation of the DGVMs. JH, LBo, NG, TI, AL,
LR, JS, RS, and DW provided an update of a GOBM. MG, LG, PL, CR, and AJW provided an
update of an ocean flux product. SA, NRB, MB, AB, HCB, WE, TG, KK, VK, NL, NM, DRM, SN,
KO, AO, TO, DP, IS, AJS, TT, BT, and RW provided ocean $pCO_2$ measurements for the year
2019, with synthesis by AO and KO. PF, MOS, and MWJ revised all figures, tables, text
and/or numbers to ensure the update is clear from the 2019 edition and in phase with the
globalcarbonatlas.org.

**Competing interests.** The authors declare that they have no conflict of interest.
**Acknowledgements.** We thank all people and institutions who provided the data used in
this carbon budget; I.G.C. Ashton, Matthew Chamberlain, Ed Chan,  Laique Djeutchouang,
Christian Ethé, Liang Feng, M. Fortier, L. Goddijn-Murphy, T. Holding, George Hurtt, Joe
Melton, Tristan Quaife, Marine Remaud, Shijie Shu, J.D. Shutler, Anthony Walker, Ulrich
Weber, and D.K. Woolf for their involvement in the development, use and analysis of the
models and data-products used here. We thank Ed Dlugokencky for providing atmospheric
$CO_2$ measurements; We thank Benjamin Pfeil, Steve Jones, Rocío Castaño-Primo and Maren
Karlsen of the Ocean Thematic Centre of the EU Integrated Carbon Observation System
(ICOS) Research Infrastructure for their contribution, as well as Karl Smith of NOAA's Pacific
Marine Environmental Laboratory; and Kim Currie, Joe Salisbury, Doug Vandermark, Chris
Hunt, Douglas Wallace and Dariia Atamanchuck, who contributed to the provision of ocean
pCO2 observations for the year 2019 (see Table A5). This is NOAA-PMEL contribution
number 5167. We thank the institutions and funding agencies responsible for the collection
and quality control of the data in SOCAT, and the International Ocean Carbon Coordination
Project (IOCCP) for its support. We thank FAO and its member countries for the collection
and free dissemination of data relevant to this work. We thank data providers ObsPack
GLOBALVIEWplus v5.0 and NRT v5.2 for atmospheric CO2 observations. We thank Trang
Chau who produced the CMEMS pCO2-based ocean flux data and designed the system
together with MG, Anna Denvil-Sommer, and FC. We thank the individuals and institutions
that provided the databases used for the model evaluations introduced here, and Nigel
Hawtin for producing Figure 2 and Figure 9. We thank Fortunat Joos, Samar Khatiwala and
Timothy DeVries for providing historical data. We thank all people and institutions who
provided the data used in this carbon budget and the Global Carbon Project members for
their input throughout the development of this update. Finally, we thank all funders who
have supported the individual and joint contributions to this work (see Table A9), as well as
the reviewers of this manuscript and previous versions, and the many researchers who have
provided feedback.

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

**Tables**

**Table 1.** Factors used to convert carbon in various units (by convention, Unit 1 = Unit 2 × conversion).

| Unit 1 | Unit 2 | Conversion | Source |
|---|---|---|---|
| GtC (gigatonnes of carbon) | ppm (parts per million)[a] | 2.124[b] | Ballantyne et al. (2012) |
| GtC (gigatonnes of carbon) | PgC (petagrams of carbon) | 1 | SI unit conversion |
| GtCO$_2$ (gigatonnes of carbon dioxide) | GtC (gigatonnes of carbon) | 3.664 | 44.01/12.011 in mass equivalent |
| GtC (gigatonnes of carbon) | MtC (megatonnes of carbon) | 1000 | SI unit conversion |

[a] Measurements of atmospheric $CO_2$ concentration have units of dry-air mole fraction. 'ppm' is an abbreviation for micromole/mol, dry air.

[b]The use of a factor of 2.124 assumes that all the atmosphere is well mixed within one year. In reality, only the troposphere is well mixed and the growth rate of $CO_2$ concentration in the less well-mixed stratosphere is not measured by sites from the NOAA network. Using a factor of 2.124 makes the approximation that the growth rate of $CO_2$ concentration in the stratosphere equals that of the troposphere on a yearly basis.

3029

| Table 2. How to cite the individual components of the global carbon budget presented here. | |
|---|---|
| **Component** | **Primary reference** |
| Global fossil CO2 emissions (EFOS), total and by fuel type | Global Carbon Project (2020) |
| National territorial fossil CO2 emissions (EFOS) | CDIAC source: Gilfillan et al. (2020) |
| | UNFCCC (2020) |
| National consumption-based fossil CO2 emissions (EFOS) by country (consumption) | Peters et al. (2011b) updated as described in this paper |
| Net land-use change flux (ELUC) | Average from Houghton and Nassikas (2017), Hansis et al. (2015), Gasser et al. (2020), all updated as described in this paper |
| Growth rate in atmospheric CO2 concentration (GATM) | Dlugokencky and Tans (2020) |
| Ocean and land CO2 sinks (SOCEAN and SLAND) | This paper for SOCEAN and SLAND and references in Table 4 for individual models. |

3030

**Table 3. Main methodological changes in the global carbon budget since 2016. Methodological changes introduced in one year are kept for the following years unless noted. Empty cells mean there were no methodological changes introduced that year. Table A7 lists methodological changes from the first global carbon budget publication up to 2015.**

| Publication year | Fossil fuel emissions | | | LUC emissions | Reservoirs | | | Uncertainty & other changes |
|---|---|---|---|---|---|---|---|---|
| | Global | Country (territorial) | Country (consumption) | | Atmosphere | Ocean | Land | |
| 2016<br><br>Le Quéré et al. (2016) | Two years of BP data | Added three small countries; China's emissions from 1990 from BP data (this release only) | | Preliminary ELUC using FRA-2015 shown for comparison; use of five DGVMs | | Based on seven models | Based on fourteen models | Discussion of projection for full budget for current year |
| 2017<br><br>Le Quéré et al. (2018a) GCB2017 | Projection includes India-specific data | | | Average of two bookkeeping models; use of twelve DGVMs | | Based on eight models that match the observed sink for the 1990s; no longer normalised | Based on fifteen models that meet observation-based criteria (see Sect. 2.5) | Land multi-model average now used in main carbon budget, with the carbon imbalance presented separately; new table of key uncertainties |
| 2018<br><br>Le Quéré et al. (2018b) GCB2018 | Revision in cement emissions; Projection includes EU-specific data | Aggregation of overseas territories into governing nations for total of 213 countries a | | Use of sixteen DGVMs | Use of four atmospheric inversions | Based on seven models | Based on sixteen models; revised atmospheric forcing from CRUNCEP to CRU-JRA-55 | Introduction of metrics for evaluation of individual models using observations |

| | | | | | | | |
|---|---|---|---|---|---|---|---|
| 2019<br><br>Friedlingstein et al. (2019) GCB2019 | Global emissions calculated as sum of all countries plus bunkers, rather than taken directly from CDIAC. | | Use of fifteen DGVMs (a) | Use of three atmospheric inversions | Based on nine models | Based on sixteen models | |
| 2020<br><br>(this study) GCB2020 | Cement carbonation now included in the EFOS estimate, reducing EFOS by about 0.2GtC yr-1 for the last decade | India's emissions from Andrew (2020: India); Corrections to Netherland Antilles and Aruba and Soviet emissions before 1950 as per Andrew (2020: $CO_2$); China's coal emissions in 2019 derived from official statistics, emissions now shown for EU27 instead of EU28.Projection for 2020 based on assessment of four approaches | Average of three bookkeeping models; use of 17 DGVMs (a) | Use of six atmospheric inversions | Based on nine models. River flux revised and partitioned NH, Tropics, SH | Based on seventeen models | |

| | | . | | | | | | |
|---|---|---|---|---|---|---|---|---|
| | | | | | | | | |

(a) ELUC is still estimated based on bookkeeping models, as in 2018 (Le Quéré et al., 2018b), but the number of DGVMs used to characterise the uncertainty has changed.

**Table 4. References for the process models, pCO2-based ocean flux products, and atmospheric inversions included in Figs. 6-8. All models and products are updated with new data to end of year 2019, and the atmospheric forcing for the DGVMs has been updated as described in Section 2.2.2.**

| Model/data name | Reference | Change from Global Carbon Budget 2019 (Friedlingstein et al., 2019) |
|---|---|---|
| *Bookkeeping models for land-use change emissions* | | |
| BLUE | Hansis et al. (2015) | No change. |
| H&N2017 | Houghton and Nassikas (2017) | No change. |
| OSCAR | Gasser et al. (2020) (a) | New this year |
| *Dynamic global vegetation models* | | |
| CABLE-POP | Haverd et al. (2018) | no change |
| CLASSIC | Melton et al. (2020) | Formerly called CLASS-CTEM. Evaporation from top soil layer is reduced which increases soil moisture and yields better GPP especially in dry and semi-arid regions. |
| CLM5.0 | Lawrence et al. (2019) | No Change. |
| DLEM | Tian et al. (2015) (b) | Updated algorithms for land use change processes. |
| IBIS | Yuan et al. (2014) | New this year |
| ISAM | Meiyappan et al. (2015) | No Change. |
| ISBA-CTRIP | Delire et al. (2020) (c) | Updated spinup protocol + model name updated (SURFEXv8 in GCB2017) + inclusion of crop harvesting module |
| JSBACH | Mauritsen et al. (2019) | No Change. |
| JULES-ES | Sellar et al., (2019) (d) | No Change. |
| LPJ-GUESS | Smith et al. (2014) (e) | Bug fixes and output code restructuring. |
| LPJ | Poulter et al. (2011) (f) | No Change. |
| LPX-Bern | Lienert and Joos (2018) | Changed compiler to Intel Fortran from PGI. |
| OCN | Zaehle and Friend (2010) (g) | No change (uses r294). |

| | | |
|---|---|---|
| ORCHIDEEv3 | Vuichard et al. (2019) (h) | Inclusion of N cycle and CN interactions in ORCHIDEE2.2 (ie CMIP6) version |
| SDGVM | Walker et al. (2017) (i) | No changes from version used in Friedlingstein et al. (2019). |
| VISIT | Kato et al. (2013) (j) | Change to distinguish managed pasture/rangeland information when conversion from natural vegetation to pasture occurs. Add upper limit of deforested biomass from secondary land using the mean biomass density data of LUH2. |
| YIBs | Yue and Unger (2015) | New this year |
| *Global ocean biogeochemistry models* | | |
| NEMO-PlankTOM5 | Buitenhuis et al. (2013) | No change |
| MICOM-HAMOCC (NorESM-OCv1.2) | Schwinger et al. (2016) | No change |
| MPIOM-HAMOCC6 | Paulsen et al. (2017) | No change |
| NEMO3.6-PISCESv2-gas (CNRM) | Berthet et al. (2019) (k) | minor bug fixes and updated spin-up procedures |
| CSIRO | Law et al (2017) | small bug fixes and revised model-spin-up |
| FESOM-1.4-REcoM2 | Hauck et al. (2020) (l) | new physical model this year |
| MOM6-COBALT (Princeton) | Liao et al. (2020) | No change |
| CESM-ETHZ | Doney et al. (2009) | included water vapor correction when converting from xCO2 to pCO2. |
| NEMO-PISCES (IPSL) | Aumont et al. (2015) | updated spin-up procedure |
| *pCO2-based flux ocean products* | | |
| Landschützer (MPI-SOMFFN) | Landschützer et al. (2016) | update to SOCATv2020 measurements and time period 1982-2019; Now use of ERA5 winds instead of ERA interim |
| Rödenbeck (Jena-MLS) | Rödenbeck et al. (2014) | update to SOCATv2020 measurements, involvement of a multi-linear regression for extrapolation (combined with an explicitly interannual correction), use of OCIM (deVries et al., 2014) as decadal prior, carbonate chemistry parameterization now time-dependent, grid resolution increased to 2.5*2 degrees, adjustable degrees of freedom now also covering shallow areas and Arctic |
| CMEMS | Chau et al. (2020) | Update to SOCATv2020 measurements and extend time period 1985-2019. Use the parameterization of air-sea CO2 fluxes as in Wanninkhof 2014 instead of Wanninkhof 1992 |
| CSIR-ML6 | Gregor et al. (2019) | New this year |
| Watson et al. | Watson et al. (2020) | New this year |

| _Atmospheric inversions_ | | |
|---|---|---|
| CAMS | Chevallier et al. (2005) with updates given in https://atmosphere.copernicus.eu/ (m) | No change. |
| CarbonTracker Europe (CTE) | van der Laan-Luijkx et al. (2017) | Model transport driven by ERA5 reanalysis. GFAS fire emissions applied instead of SIBCASA-GFED. Rodenbeck et al ocean fluxes used as priors instead of Jacobson et al., (2007) |
| Jena CarboScope | Rödenbeck et al. (2003, 2018) | No change. |
| UoE in-situ | Feng et al., (2016) (n) | New this year |
| NISMON-CO2 | Niwa et al., (2017) | New this year |
| MIROC4-ACTM | Patra et al., (2018) | New this year |

(a) see also Gasser et al. (2017)

(b) See also Tian et al. (2011)

(c) See also Decharme et al. (2019) and Seferian et al. (2019)

(d) JULES-ES is the Earth System configuration of the Joint UK Land Environment Simulator. See also Best et al. (2011), Clark et al. (2011) and Wiltshire et al., (2020).

(e) To account for the differences between the derivation of shortwave radiation from CRU cloudiness and DSWRF from CRUJRA, the photosythesis scaling parameter $\alpha a$ was modified (-15%) to yield similar results.

(f) Lund-Potsdam-Jena. Compared to published version, decreased LPJ wood harvest efficiency so that 50 % of biomass was removed off-site compared to 85 % used in the 2012 budget. Residue management of managed grasslands increased so that 100 % of harvested grass enters the litter pool.

(g) See also Zaehle et al. (2011).

(h) See Zaehle and Friend (2010) and Krinner et al. (2005)

(i) See also Woodward and Lomas (2004)

(j) See also Ito and Inatomi (2012)

(k) See also Seferian et al (2019)

(l) Longer spin-up than in Hauck et al (2020); see also Schourup-Kristensen et al (2014)

(m) See also Remaud et al. (2018)

(n) See also Feng et al., (2009) and Palmer et al., (2019)

**Table 5. Comparison of results from the bookkeeping method and budget residuals with results from the DGVMs and inverse estimates for different periods, the last decade, and the last year available. All values are in GtCyr–1. The DGVM uncertainties represent ±1σ of the decadal or annual (for 2019 only) estimates from the individual DGVMs: for the inverse models the range of available results is given. All values are rounded to the nearest 0.1 GtC and therefore columns do not necessarily add to zero.**

| | Mean (GtC yr-1) | | | | | | |
|---|---|---|---|---|---|---|---|
| | 1960-1969 | 1970-1979 | 1980-1989 | 1990-1999 | 2000-2009 | 2010-2019 | 2019 |
| *Land-use change emissions (ELUC)* | | | | | | | |
| **Bookkeeping methods - Net flux (1a)** | 1.5±0.7 | 1.3±0.7 | 1.3±0.7 | 1.4±0.7 | 1.4±0.7 | 1.6±0.7 | 1.8±0.7 |
| Bookkeeping methods - Source | 3.5±1.2 | 3.3±1.1 | 3.5±1.3 | 3.8±0.9 | 4.1±1.2 | 4.4±1.6 | 4.6±1.8 |
| Bookkeeping methods - Sink | -2±0.7 | -2.1±0.7 | -2.2±0.8 | -2.4±0.9 | -2.7±1.1 | -2.9±1.2 | -2.9±1.2 |
| DGVMs - Net flux (1b) | 1.4±0.5 | 1.4±0.5 | 1.5±0.5 | 1.4±0.5 | 1.6±0.5 | 2.1±0.5 | 2.2±0.7 |
| *Terrestrial sink (SLAND)* | | | | | | | |
| Residual sink from global budget (EFF+ELUC-GATM-SOCEAN) (2a) | 1.7±0.8 | 1.9±0.8 | 1.6±0.9 | 2.6±0.9 | 2.9±0.9 | 3.3±1.0 | 3.5±1.1 |
| **DGVMs (2b)** | 1.3±0.4 | 2.1±0.4 | 2.0±0.7 | 2.6±0.7 | 2.9±0.8 | 3.4±0.9 | 3.1±1.2 |
| *Total land fluxes (SLAND – ELUC)* | | | | | | | |
| **GCB2020 Budget (2b - 1a)** | -0.2±0.9 | 0.8±0.8 | 0.7±1.0 | 1.2±1.0 | 1.5±1.1 | 1.9±1.1 | 1.3±1.4 |
| Budget constraint (2a - 1a) | 0.3±0.6 | 0.6±0.6 | 0.3±0.7 | 1.2±0.7 | 1.5±0.7 | 1.8±0.8 | 1.7±0.7 |
| DGVMs (2b - 1b) | -0.2±0.5 | 0.7±0.4 | 0.5±0.6 | 1.2±0.4 | 1.3±0.6 | 1.3±0.6 | 1.0±1.1 |
| Inversions* | - | - | 0.1 - 0.6 (2) | 0.6 - 1.1 (3) | 1.0 - 1.8 (4) | 1.2 - 2.3 (6) | 0.7-1.9 (6) |

*Estimates are adjusted for the pre-industrial influence of river fluxes and adjusted to common EFOS (Sect. 2.6.1). The ranges given include varying numbers (in parentheses) of inversions in each decade (Table A4).

**Table 6. Decadal mean in the five components of the anthropogenic CO2 budget for different periods, and last year available. All values are in GtC yr-1, and uncertainties are reported as ±1σ. The table also shows the budget imbalance (BIM), which provides a measure of the discrepancies among the nearly independent estimates and has an uncertainty exceeding ± 1 GtC yr-1. A positive imbalance means the emissions are overestimated and/or the sinks are too small. All values are rounded to the nearest 0.1 GtC and therefore columns do not necessarily add to zero.**

|  | Mean (GtC yr-1) | | | | | | |
|---|---|---|---|---|---|---|---|
|  | 1960-1969 | 1970-1979 | 1980-1989 | 1990-1999 | 2000-2009 | 2010-2019 | 2019 |
| *Total emissions (EFOS+ELUC)* |  |  |  |  |  |  |  |
| Fossil CO2 emissions (EFOS) | 3±0.2 | 4.7±0.2 | 5.4±0.3 | 6.3±0.3 | 7.7±0.4 | 9.4±0.5 | 9.7±0.5 |
| Land-use change emissions (ELUC) | 1.5±0.7 | 1.3±0.7 | 1.3±0.7 | 1.4±0.7 | 1.4±0.7 | 1.6±0.7 | 1.8±0.7 |
| Total emissions | 4.5±0.7 | 5.9±0.7 | 6.7±0.8 | 7.6±0.8 | 9.1±0.8 | 10.9±0.9 | 11.5±0.9 |
| *Partitioning* |  |  |  |  |  |  |  |
| Growth rate in atmospheric CO2 concentration (GATM) | 1.8±0.07 | 2.8±0.07 | 3.4±0.02 | 3.2±0.02 | 4.1±0.02 | 5.1±0.02 | 5.4±0.2 |
| Ocean sink (SOCEAN) | 1±0.3 | 1.3±0.4 | 1.7±0.4 | 2±0.5 | 2.1±0.5 | 2.5±0.6 | 2.6±0.6 |
| Terrestrial sink (SLAND) | 1.3±0.4 | 2.1±0.4 | 2.0±0.7 | 2.6±0.7 | 2.9±0.8 | 3.4±0.9 | 3.1±1.2 |
| *Budget imbalance* |  |  |  |  |  |  |  |
| BIM = EFOS+ELUC - (GATM+SOCEAN+SLAND) | 0.5 | -0.2 | -0.4 | -0.1 | 0 | -0.1 | 0.3 |

**Table 7. Comparison of the projection with realised fossil CO2 emissions ($E_{FOS}$). The 'Actual' values are first the estimate available using actual data, and the 'Projected' values refers to estimates made before the end of the year for each publication. Projections based on a different method from that described here during 2008-2014 are available in Le Quéré et al., (2016). All values are adjusted for leap years.**

| | World | | China | | USA | | EU28 | | India | | Rest of World | |
|---|---|---|---|---|---|---|---|---|---|---|---|---|
| | Projected | Actual | Projected | Actual | Projected | Actual | Projected | Actual | Projected | Actual | Projected | Actual |
| 2015 (a) | −0.6% (−1.6 to 0.5) | 0.06% | −3.9% (−4.6 to −1.1) | −0.7% | −1.5% (−5.5 to 0.3) | −2.5% | – | – | – | – | 1.2% (−0.2 to 2.6) | 1.2% |
| 2016 (b) | −0.2% (−1.0 to +1.8) | 0.20% | −0.5% (−3.8 to +1.3) | −0.3% | −1.7% (−4.0 to +0.6) | −2.1% | – | – | – | – | 1.0% (−0.4 to +2.5) | 1.3% |
| 2017 (c) | 2.0% (+0.8 to +3.0) | 1.6% | 3.5% (+0.7 to +5.4) | 1.5% | −0.4% (−2.7 to +1.0) | −0.5% | – | – | 2.00% (+0.2 to +3.8) | 3.9% | 1.6% (0.0 to +3.2) | 1.9% |
| 2018 (d) | 2.7% (+1.8 to +3.7) | 2.1% | 4.7% (+2.0 to +7.4) | 2.3% | 2.5% (+0.5 to +4.5) | 2.8% | -0.7% (-2.6 to +1.3) | -2.1% | 6.3% (+4.3 to +8.3) | 8.0% | 1.8% (+0.5 to +3.0) | 1.7% |
| 2019 (e) | 0.5% (-0.3 to +1.4) | 0.1% | 2.6% (+0.7 to +4.4) | 2.2% | -2.4% (-4.7 to -0.1) | -2.6% | -1.7% (-5.1% to +1.8%) | -4.3% | 1.8% (-0.7 to +3.7) | 1.0% | 0.5% (-0.8 to +1.8) | 0.5% |
| 2020 (f) | -6.7% | | -1.7% | | -12.2% | | -11.3% (EU27) | | -9.1% | | -7.4% | |

(a) Jackson et al. (2016) and Le Quéré et al. (2015a). (b) Le Quéré et al. (2016). (c) Le Quéré et al. (2018a). (d) Le Quéré et al. (2018b). (e) Friedlingstein et al., (2019), (f) This study (median of four reported estimates, Section 3.4.1.2)

**Table 8. Cumulative CO2 for different time periods in gigatonnes of carbon (GtC). All uncertainties are reported as ±1σ. The budget imbalance provides a measure of the discrepancies among the nearly independent estimates. Its uncertainty exceeds ±60 GtC. The method used here does not capture the loss of additional sink capacity from reduced forest cover, which is about 20 GtC and would exacerbate the budget imbalance (see Sect. 2.7.4). All values are rounded to the nearest 5 GtC and therefore columns do not necessarily add to zero.**

| Units of GtC | 1750-2019 | 1850-2014 | 1959-2019 | 1850-2019 | 1850-2020 (a) |
|---|---|---|---|---|---|
| *Emissions* | | | | | |
| Fossil $CO_2$ emissions (EFOS) | 445±20 | 395±20 | 365±20 | 445±20 | 455±20 |
| Land-use change $CO_2$ emissions (ELUC) | 255±70b | 200±60c | 85±45d | 210±60c | 210±60 |
| Total emissions | 700±75 | 595±65 | 450±50 | 650±65 | 665±65 |
| *Partitioning* | | | | | |
| Growth rate in atmospheric $CO_2$ concentration (GATM) | 285±5 | 235±5 | 205±5 | 265±5 | 270±5 |
| Ocean sink (SOCEAN) (e) | 170±20 | 145±20 | 105±20 | 160±20 | 165±20 |
| Terrestrial sink (SLAND) | 230±60 | 195±50 | 145±35 | 210±55 | 215±55 |
| *Budget imbalance* | | | | | |
| BIM = EFOS+ELUC - (GATM+SOCEAN+ SLAND) | 20 | 20 | 0 | 20 | 20 |

a Using projections for year 2020 (Sect. 3.4). Uncertainties are the same as 1850-2019 period

b Cumulative ELUC 1750-1849 of 30 GtC based on multi-model mean of Pongratz et al. (2009), Shevliakova et al. (2009), Zaehle et al. (2011), Van Minnen et al. (2009). 1850-2019 from mean of H&N (Houghton and Nassikas, 2017) and BLUE (Hansis et al., 2015). 1750-2019 uncertainty is estimated from standard deviation of DGVMs over 1870-2019 scaled by 1750-2019 emissions.

c Cumulative ELUC based on H&N, BLUE, and OSCAR. Uncertainty is estimated from the standard deviation of DGVM estimates

d Cumulative ELUC based on H&N, BLUE, and OSCAR. Uncertainty is formed from the uncertainty in annual ELUC over 1959-2019, which is 0.7 GtC/yr multiplied by length of the time series

e Ocean sink uncertainty from IPCC (Denman et al., 2007)

**Table 9. Major known sources of uncertainties in each component of the Global Carbon Budget, defined as input data or processes that have a demonstrated effect of at least ±0.3 GtC yr-1.**

| Source of uncertainty | Time scale (years) | Location | Status | Evidence |
|---|---|---|---|---|
| Fossil CO2 emissions (EFOS; Section 2.1) | | | | |
| energy statistics | annual to decadal | global, but mainly China & major developing countries | see Sect. 2.1 | (Korsbakken et al., 2016, Guan et al., 2012) |
| carbon content of coal | annual to decadal | global, but mainly China & major developing countries | see Sect. 2.1 | (Liu et al., 2015) |
| system boundary | annual to decadal | all countries | see Sect. 2.1 | |
| Net land-use change flux (ELUC; section 2.2) | | | | |
| land-cover and land-use change statistics | continuous | global; in particular tropics | see Sect. 2.2 | (Houghton et al., 2012; Gasser et al., 2020) |
| sub-grid-scale transitions | annual to decadal | global | see Table A1 | (Wilkenskjeld et al., 2014) |
| vegetation biomass | annual to decadal | global; in particular tropics | see Table A1 | (Houghton et al., 2012) |
| wood and crop harvest | annual to decadal | global; SE Asia | see Table A1 | (Arneth et al., 2017, Erb et al., 2018) |
| peat burning (a) | multi-decadal trend | global | see Table A1 | (van der Werf et al., 2010) |
| loss of additional sink capacity | multi-decadal trend | global | not included; Section 2.7.4 | (Pongratz et al, 2014, Gasser et al, 2020) |
| Atmospheric growth rate (GATM; section 2.3) no demonstrated uncertainties larger than ±0.3 GtC yr-1 (b) | | | | |
| Ocean sink (SOCEAN; section 2.4) | | | | |
| variability in oceanic circulation (c) | semi-decadal to decadal | global | see Sect. 2.4 | (DeVries et al., 2017, 2019) |

| internal variability | annual to decadal | high latitudes; Equatorial Pacific | no ensembles/ coarse resolution | (McKinley et al., 2016) |
|---|---|---|---|---|
| anthropogenic changes in nutrient supply | multi-decadal trend | global | not included | (Duce et al., 2008) |
| Land sink (SLAND; section 2.5) | | | | |
| strength of CO2 fertilisation | multi-decadal trend | global | see Sect. 2.5 | (Wenzel et al., 2016) |
| response to variability in temperature and rainfall | annual to decadal | global; in particular tropics | see Sect. 2.5 | (Cox et al., 2013) |
| nutrient limitation and supply | | | | |
| response to diffuse radiation | annual | global | see Sect. 2.5 | (Mercado et al., 2009) |

a As result of interactions between land-use and climate

b The uncertainties in GATM have been estimated as ±0.2 GtC yr-1, although the conversion of the growth rate into a global annual flux assuming instantaneous mixing throughout the atmosphere introduces additional errors that have not yet been quantified.

c Could in part be due to uncertainties in atmospheric forcing (Swart et al., 2014)

# Appendix A. Supplementary tables.

Table A1. Comparison of the processes included in the bookkeeping method and DGVMs in their estimates of ELUC and SLAND. See Table 4 for model references. All models include deforestation and forest regrowth after abandonment of agriculture (or from afforestation activities on agricultural land). Processes relevant for ELUC are only described for the DGVMs used with land-cover change in this study (Fig. 6 top panel).

| | Bookkeeping Models | | | DGVMs | | | | | | | | | | | | | | | | |
| --- | --- | --- | --- | --- | --- | --- | --- | --- | --- | --- | --- | --- | --- | --- | --- | --- | --- | --- | --- | --- |
| | H&N | BLUE | OSCAR | CAB LE-PO P | CLA SSI C | CL M5. 0 | DLE M | IBIS | ISA M | ISB A-CTR IP(h ) | JSB AC H | JUL ES-ES | LPJ-GU ESS | LPJ | LPX -Ber n | OC Nv2 | OR CHI DEE v3 | SD GV M | VISI T | YIB s |
| **Processes relevant for ELUC** | | | | | | | | | | | | | | | | | | | | |
| Wood harvest and forest degradation (a) | yes | yes | yes | yes | no | yes | yes | yes | yes | no | yes | no | yes | yes | no (d) | yes | yes | no | yes | no |
| Shifting cultivation / Subgrid scale transitions | no (b) | yes | yes | yes | no | yes | no | no | no | no | yes | no | yes | yes | no (d) | no | no | no | yes | no |
| Cropland harvest (removed, R, or added to litter, L) | yes (R) (z) | yes (R) (z) | yes (R) | yes (R) | yes (L) | yes (R) | yes | yes (R) | yes | yes (R+ L) | yes (R+ L) | yes (R) | yes (R) | yes (L) | yes (R) | yes (R+ L) | yes (R) | yes (R) | yse (R) | no |
| Peat fires | yes | yes | yes | no | no | yes | no | no | no | no | no | no | no | no | no | no | no | no | no | no |
| fire as a management tool | yes (z) | yes (z) | yes (j) | no | no | no | no | no | no | no | no | no | no | no | no | no | no | no | no | no |
| N fertilization | yes (z) | yes (z) | yes (j) | no | no | yes | yes | no | yes | no | no | yes( k) | yes | no | yes | yes | yes | no | no | no |
| tillage | yes (z) | yes (z) | yes (j) | yes | yes (g) | no | no | no | no | no | no | no | yes | no | no | no | yes (g) | no | no | no |
| irrigation | yes (z) | yes (z) | yes (j) | no | no | yes | yes | no | yes | no | no | no | yes | no | no | no | no | no | no | no |
| wetland drainage | yes (z) | yes (z) | yes (j) | no | no | no | no | no | no | no | no | no | no | no | no | no | no | no | no | no |
| erosion | yes (z) | yes (z) | yes (j) | no | no | no | yes | no | no | no | no | no | no | no | no | no | no | no | yes | no |
| peat drainage | yes | yes | yes | no | no | no | no | no | no | no | no | no | no | no | no | no | no | no | no | no |
| Grazing and mowing Harvest (removed, r, or added to litter, l) | yes (r) (z) | yes (r) (z) | yes (r) | yes (r) | no | no | no | no | yes (l) | no | yes (l) | no | yes (r) | yes (l) | no | yes (r+l) | no | no | no | no |
| **Processes also relevant for SLAND** | | | | | | | | | | | | | | | | | | | | |
| Fire simulation and/or suppression | for US only | no | yes (m) | no | yes | yes | yes | yes | no | yes | yes | no | yes | yes | yes | no | no | yes | yes | no |
| Climate and variability | no | no | yes | yes | yes | yes | yes | yes | yes | yes | yes | yes | yes | yes | yes | yes | yes | yes | yes | yes |
| CO2 fertilisation | no (i) | no (i) | yes | yes | yes | yes | yes | yes | yes | yes | yes | yes | yes | yes | yes | yes | yes | yes | yes | yes |

| Carbon-nitrogen interactions, including N deposition | no (z) | no (z) | no (j) | yes | no (f) | yes | yes | no | yes | no (e) | yes | yes | yes | no | yes | yes | yes | yes (c) | no | no |
|---|---|---|---|---|---|---|---|---|---|---|---|---|---|---|---|---|---|---|---|---|

**(z) Process captured implicitly by use of observed carbon densities.**

**(a) Refers to the routine harvest of established managed forests rather than pools of harvested products.**

**(b) No back- and forth-transitions between vegetation types at the country-level, but if forest loss based on FRA exceeded agricultural expansion based on FAO, then this amount of area was cleared for cropland and the same amount of area of old croplands abandoned.**

**(c) Limited. Nitrogen uptake is simulated as a function of soil C, and Vcmax is an empirical function of canopy N. Does not consider N deposition.**

**(d) Available but not active.**

**(e) Simple parameterization of nitrogen limitation based on Yin (2002; assessed on FACE experiments)**

**(f) Although C-N cycle interactions are not represented, the model includes a parameterization of down-regulation of photosynthesis as $CO_2$ increases to emulate nutrient constraints (Arora et al., 2009)**

**(g) Tillage is represented over croplands by increased soil carbon decomposition rate and reduced humification of litter to soil carbon.**

**(h) ISBA-CTRIP corresponds to SURFEXv8 in GCB2018**

**(i) Bookkeeping models include the effect of $CO_2$-fertilization as captured by present-day carbon densities, but not as an effect transient in time.**

(j) as far as the DGVMs that OSCAR is calibrated to include it

(k) perfect fertilisation assumed, i.e. crops are not nitrogen limited and the implied fertiliser diagnosed

(m) fire intensity responds to climate and $CO_2$, but no fire suppression

| | NEMO-PlankTOM5 | NEMO-PISCES (IPSL) | MICOM-HAMOCC (NorESM1-OCv1.2) | MPIOM-HAMOCC6 | CSIRO | FESOM-1.4-REcoM2 | NEMO3.6-PISCESv2-gas (CNRM) | MOM6-COBALT (Princeton) | CESM-ETHZ |
|---|---|---|---|---|---|---|---|---|---|
| **SPIN-UP procedure** | | | | | | | | | |
| Initialisation of carbon chemistry | GLODAPv1 corrected for anthropogenic carbon from Sabine et al (2004) | GLODAPv2 | GLODAPv1 (preindustrial DIC) | initialization from previous model simulations | GLODAPv1 preindustrial | GLODAPv2 alkalinity and preindustrial DIC | GLODAPv2 | GLODAPv2 for Alkalinity and DIC. DIC is corrected to 1959 level for simulation A and corrected to pre-industrial level for simulation B using Khatiwala et al 2009, 2013 | GLODAPv2 preindustrial |
| Preindustrial spin-up prior to 1850 | spin-up 1750-1947 | spin-up starting in 1836 with 3 loops of JRA55 | 1000 year spin up | spin-up with ERA20C | 800 years | no | long spin-up (> 1000 years) | Other biogeochemical tracers are initialized from a GFDL-ESM2M spin-up (> 1000 years) | spinup 1655-1849 |

Table A2. Comparison of the processes and model set up for the Global Ocean Biogeochemistry Models for their estimates of SOCEAN. See Table 4 for model references.

| | | | | | | | | | |
|---|---|---|---|---|---|---|---|---|---|
| Atmospheric forcing for pre-industrial spin-up | looping NCEP year 1980 | JRA55 | CORE-I (normal year) forcing | ERA20C | CORE+JRA55 | not applicable | NCEP2 repeat year 1948 perpetually | GFDL-ESM2M internal forcing | COREv2 forcing until 1835, three cycles of conditions from 1949-2009. from 1835-1850: JRA forcing |
| Atmospheric forcing for historical spin-up 1850-1958 for simulation A | 1750-1947: looping NCEP year 1980; 1948-2019: NCEP | 1836-1958 : looping full JRA55 reanalysis | CORE-I (normal year) forcing; from 1948 onwards NCEP-R1 with CORE-II corrections | NCEP / NCEP+ERA20C (spin-up) | JRA55do cyclic 1958 | JRA55-do-v1.3.1 repeat year 1961 | NCEP2 repeat year 1948 perpetually | JRA55-do-v1.4 repeat year 1959 (81 years) | JRA55 version 1.3, repeat cycle between 1958-2018. |
| Atmospheric CO2 for historical spin-up 1850-1958 for simulation A | provided by the GCP; converted to pCO2 temperature formulation (Sarmiento et al., JGR 1992), monthly resolution | xCO2 as provided by the GCB, global mean, annual resolution, converted to pCO2 with sea-level pressure and water vapour pressure | xCO2 as provided by the GCB, converted to pCO2 assuming constant standard seal level pressure, no water vapour correction | xCO2 provided by the GCB, no conversion | xCO2 provided by GCP converted to pCO2 with SLP, no water vapour correction | xCO2 as provided by the GCB, converted to pCO2 with sea-level pressure and water vapour pressure, global mean, monthly resolution | xCO2 as provided by the GCB, converted to pCO2 with constant sea-level pressure and water vapour pressure, global mean, yearly resolution | xCO2 at year 1959 level (315 ppm), converted to pCO2 with sea-level pressure and water vapour pressure, global mean, yearly resolution | xCO2 as provided by the GCB, converted to pCO2 with atmospheric pressure, and locally determined water vapour pressure from SST and SSS (100% saturation) |
| Atmospheric forcing for control spin-up 1850-1958 for simulation B | 1750-2019: looping NCEP 1980 | not available | CORE-I (normal year) forcing | spin-up initial restart file with cyclic 1957 NCEP; run 1957-2017 | JRA55do cyclic 1958 | JRA55-do-v1.3.1 repeat year 1961 | NCEP2 repeat year 1948 perpetually | JRA55-do-v1.4 repeat year 1959 (81 years) | normal year forcing created from JRA-55 version 1.3, NYF = climatology with anomalies from the year 2001 |
| Atmospheric CO2 for control spin-up 1850-1958 for simulation B (ppm) | constant 278ppm; converted to pCO2 temperature formulation | N/A | xCO2 of 278 ppm, converted to pCO2 assuming constant standard seal level | 278, no conversion, assuming constant standard sea level pressure | 280, converted to pCO2 with SLP, no water vapour correction | xCO2 of 278ppm, converted to pCO2 with sea-level pressure and water | xCO2 of 278ppm, converted to pCO2 with constant sea-level pressure | xCO2 of 278ppm, converted to pCO2 with sea-level pressure and water | xCO2 as provided by the GCB for 1850, converted to pCO2 with |

| | | | | | | | | | |
|---|---|---|---|---|---|---|---|---|---|
| | (Sarmiento et al., JGR 1992), monthly resolution | | pressure | | | vapour pressure | and water vapour pressure | vapour pressure | atmospheric pressure, and locally determined water vapour pressure from SST and SSS (100% saturation) |
| **Simulation A** | | | | | | | | | |
| Atmospheric forcing for simulation A | NCEP | JRA55 | NCEP-R1 with CORE-II corrections | NCEP / NCEP+ERA-20C (spin-up) | JRA55do | JRA55-do-v1.4.0 1958-2018 and JRA55-do-v1.4.0.1b for 2019 | NCEP with CORE-II corrections | JRA55-do-v1.4.0 1959-2018 and JRA55-do-v1.4.0.1b for 2019 | JRA-55 version 1.3 |
| Atmospheric CO2 for simulation A | provided by the GCP; converted to pCO2 temperature formulation (Sarmiento et al., JGR 1992), monthly resolution | xCO2 as provided by the GCB, global mean, annual resolution, converted to pCO2 with sea-level pressure and water vapour pressure | monthly xCO2 as provided by the GCB, converted to pCO2 assuming constant standard seal level pressure | monthly xCO2 as provided by the GCB, no conversion | xCO2 provided by GCP converted to pCO2 with SLP, no water vapour correction | xCO2 as provided by the GCB, converted to pCO2 with sea-level pressure and water vapour pressure, global mean, monthly resolution | xCO2 as provided by the GCB, converted to pCO2 with constant sea-level pressure and water vapour pressure, global mean, yearly resolution | xCO2 as provided by the GCB, converted to pCO2 with sea-level pressure and water vapour pressure, global mean, yearly resolution | xCO2 as provided by the GCB, converted to pCO2 with atmospheric pressure, and locally determined water vapour pressure from SST and SSS (100% saturation) |
| **Simulation B** | | | | | | | | | |
| Atmospheric forcing for simulation B | NCEP 1980 | N/A | CORE-I (normal year) forcing | spin-up initial restart file (278) with cyclic 1957 NCEP; run 1957-2017 with 278 | JRA55do cyclic 1958 | JRA55-do-v1.3.1 repeat year 1961 | NCEP with CORE-II corrections cycling over 1948-1957 | JRA55-do-v1.4.0 repeat year 1959 | normal year forcing created from JRA-55 version 1.3, NYF = climatology with anomalies from the year 2001 |

| Atmospheric CO2 for simulation B | constant 278ppm; converted to pCO2 temperature formulation (Sarmiento et al., JGR 1992), monthly resolution | N/A | xCO2 of 278 ppm, converted to pCO2 assuming constant standard seal level pressure | | 280 | xCO2 of 278ppm, converted to pCO2with sea-level pressure and water vapour pressure | xCO2 of 278ppm, converted to pCO2 with constant sea-level pressure and water vapour pressure | xCO2 of 278ppm, converted to pCO2 with sea-level pressure and water vapour pressure | xCO2 as provided by the GCB for 1850, converted to pCO2 with atmospheric pressure, and locally determined water vapour pressure from SST and SSS (100% saturation) |
|---|---|---|---|---|---|---|---|---|---|
| **Model specifics** | | | | | | | | | |
| Physical ocean model | NEMOv2.3-ORCA2 | NEMOv3.6-eORCA1L75 | MICOM (NorESM1-OCv1.2) | MPIOM | MOM5 | FESOM-1.4 | NEMOv3.6-GELATOv6-eORCA1L75 | MOM6-SIS2 | CESMv1.4 (ocean model based on POP2) |
| Biogeochemistry model | PlankTOM5.3 | PISCESv2 | HAMOCC (NorESM1-OCv1.2) | HAMOCC6 | WOMBAT | REcoM-2 | PISCESv2-gas | COBALTv2 | BEC (modified & extended) |
| Horizontal resolution | 2o lon, 0.3 to 1.5o lat | 1° lon, 0.3 to 1° lat | 1° lon, 0.17 to 0.25 lat (nominally 1°) | 1.5° | 1o x1o with enhanced resolution at the tropics and in the high lat Southern Ocean | unstructured multi-resolution mesh. CORE-mesh, with 20-120 km resolution. Highest resolution north of 50N, intermediate in the equatorial belt and Southern Ocean, lowest in the subtropical gyres | 1° lon, 0.3 to 1° lat | 0.5° lon, 0.25 to 0.5° lat | Lon: 1.125°, Lat varying from 0.53° in the extratropics to 0.27° near the equator |
| Vertical resolution | 31 levels | 75 levels, 1m at the surface | 51 isopycnic layers + 2 layers representing a bulk mixed layer | 40 levels, layer thickness increase with depth | 50 levels, 20 in the 200m | 46 levels, 10 m spacing in the top 100 m | 75 levels, 1m at surface | 75 levels hybrid coordinates, 2 m at surface | 60 levels (z-coordinates) |
| Total ocean area | 3.6080E+ | 3.6270E+ | 3.6006E+ | 3.6598E+ | 3.6134E+ | 3.6475E+ | 3.6270E+ | 3.6110E+ | 3.5926E+ |

| | | | | | | | | | |
|---|---|---|---|---|---|---|---|---|---|
| on native grid (km2) | 08 | 08 | 08 | 08 | 08 | 08 | 08 | 08 | 08 |
| Gas-exchange parameterization | Quadratic exchange formulation (function of T + 0.3*U^2)*(Sc/660)^-0.5) ; Wanninkhof et al. 1992 (Equation 8) | see Orr et al 2017: kw parameterized from Wanninkhof of 1992, with kw = a*(Sc/660)^-0.5) *u2*(1-f_ice) with a from Wanninkhof et al 2014 | see Orr et al 2017: kw parameterized from Wanninkhof of 1992, with kw = a*(Sc/660)^-0.5) *u2*(1-f_ice) with a=0.337 following the OCMIP2 protocols | Gas transfer velocity formulation and parameter setup of Wanninkhof of (2014), including updated Schmidt number parameterizations for CO2 to comply with OMIP protocol (Orr et al., 2017) | Quadratic exchange formulation (function of T + 0.3*U^2)*(Sc/660)^-0.5) ; Wanninkhof et al. 1992 (Equation 8) | see Orr et al 2017: kw parameterized from Wanninkhof of 1992, with kw = a*(Sc/660)^-0.5) *u2*(1-f_ice) with a from Wanninkhof of et al 2014 | see Orr et al 2017: kw parameterized from Wanninkhof of 1992, with kw = a*(Sc/660)^-0.5) *u2*(1-f_ice) with a from Wanninkhof of et al 2014 | see Orr et al 2017: kw parameterized from Wanninkhof of 1992, with kw = a*(Sc/660)^-0.5) *u2*(1-f_ice) with a from Wanninkhof of et al 2014 | Gas exchange is parameterized using the Wanninkhof of (1992) quadratic windspeed dependency formulation, but with the coefficient scaled down to reflect the recent 14C inventories. Concretely, we used a coefficent a of 0:31 cm hr-1 s2 m-2 to read kw = 0:31 ws^2 (1-fice) (Sc=660)^{-1/2} |
| Time-step | 96 mins | 45 min | 3200 sec | 60 mins | 15 min | 15 min | 15 min | 30 min | 3757 sec |
| Output frequency | Monthly | monthly | monthly/daily | monthly | monthly | monthly | monthly | monthly | monthly |
| CO2 chemistry routines | Following Broecker et al. (1982) | mocsy | Following Dickson et al. (2007) | as in Ilyina et al. (2013) adapted to comply with OMIP protocol (Orr et al., 2017). | OCMIP2 (Orr et al.) | mocsy | mocsy | mocsy | OCMIP2 (Orr et al.) |
| River carbon input (GtC/yr) | 60.24 Tmol/yr; 0.723 GtC/yr | 0.61 GtC y-1 | 0 | none | 0 | 0 | ~0.6 GtC y-1 | ~0.11 GtC y-1 | 0.33 Gt C yr-1 |
| Burial/net flux into the sediment (GtC/yr) | 0.723 GtC/yr | 0.59 GtC y-1 | 0 | around 0.4 GtC/yr | 0 | 0 | ~0.7 GtC y-1 | ~0.21 GtC y-1 | 0.25 Gt C yr-1 |

Table A3: Description of ocean data-products used for assessment of SOCEAN. See Table 4 for references.

| data-products | Jena-MLS | MPI-SOMFFN | CMEMS | CSIR | Watson et al |
|---|---|---|---|---|---|
| Method | Spatio-temporal interpolation (update of Rödenbeck et al., 2013, version oc_v2020). Specifically, the sea-air CO2 fluxes and the pCO2 field are numerically linked to each other and to the spatio-temporal field of ocean-internal carbon sources/sinks through process parametrizations, and the ocean-internal sources/sink field is then fit to the SOCATv2020 pCO2 data (Bakker et al. 2020). The fit includes a multi-linear regression against environmental drivers to bridge data gaps, and interannually explicit corrections to represent the data signals more completely. | 2-step neural network method where in a first step the global ocean is clustered into 16 biogeochemical provinces using a self-organizing map (SOM). In a second step, the non-linear relationship between available pCO2 measurements from the SOCAT database (Bakker et al 2016) and environmental predictor data (SST, SSS, MLD, CHL-a, atmospheric CO2 - references see Landschützer et al 2016) are established using a feed-forward neural network (FFN) for each province separately. The established relationship is then used to fill the existing data gaps (see Landschützer et al 2013, Landschützer et al 2016). | An ensemble of neural network models trained on 100 subsampled datasets from the Surface Ocean CO2 Atlas (SOCAT, Bakker et al 2016) . Like the original data, subsamples are distributed after interpolation on 1x1 grid cells along ship tracks. Sea surface salinity, temperature, sea surface height, mixed layer depth, atmospheric CO2 mole fraction, chlorophyll, spco2 climatology, latitude and longitude are used as predictors. The models are used to reconstruct sea surface pCO2, and then convert to air-sea CO2 fluxes. | An ensemble average of six machine learning estimates of pCO2 using the approach described in Gregor et al. (2019) with the updated product using SOCAT v2020. All ensemble members use a cluster-regression approach. Two different cluster configurations are used: 1) based on K-means clustering; 2) Fay and McKinley (2014) 's CO2 biomes. Three regression algorithms are used: 1) gradient boosted decision trees, 2) feed-forward neural network, 3) support vector regression. The product of the cluster configurations and the regression algorithms results in an ensemble with six members. | Derived from the the SOCAT(v2020) pCO2 database, but corrected to the subskin temperature of the ocean as measured by satellite, using the methodology described by Goddijn-Murphy et al (2015). A correction to the flux calculation is also applied for the cool and salty surface skin. In other respects the product uses interpolation of the data using the two step neural network based on MPI-SOMFFN :in the first step the ocean is divided into a monthly climatology of 16 biogeochemical provinces using a SOM, In the second step a feed-forward nerual network establishes non-linear relationships between pCO2 and SST, SSS, mixed layer depth(MLD) and atmospheric xCO2 in each of the 16 provinces. Further description in Watson et al (2020). |
| Gas-exchange parameterization | Quadratic exchange formulation ($k*U^2* (Sc/660)^{-0.5}$) (Wanninkhof 1992) with the transfer coefficient k scaled to match a global mean transfer rate of 16 cm/hr by Naegler (2009) | Quadratic exchange formulation ($k*U^2* (Sc/660)^{-0.5}$) (Wanninkhof 1992) with the transfer coefficient k scaled to match a global mean transfer rate of 16 cm/hr (calculated myself over the full period 1982-2019 - not follwing Naegler) | Quadratic exchange formulation ($k*U^2* (Sc/660)^{-0.5}$) (Wanninkhof et al. 2014) with the transfer coefficient k scaled to match a global mean transfer rate of 16 cm/hr by Naegler (2009) | Quadratic exchange formulation ($k*U^2* (Sc/660)^{-0.5}$) (Wanninkhof 1992) with the transfer coefficient k scaled to match a global mean transfer rate of 16 cm/hr by Naegler (2009) | Nightingale et al. (2000) formulation : $K=((Sc/600)^{-0.5})*(0.333*U +0.222*U^2)$ |

| | | | | | |
|---|---|---|---|---|---|
| Wind product | NCEP reanalysis (Kalnay et al., 1996) | ERA 5 | ERA5 | ERA5 | CCMP wind product, 0.25 x 0.25 degrees x 6-hourly, from which we calculate mean and mean square winds over 1 x 1 degree and 1 month intervals. |
| Spatial resolution | 2.5 degrees longitude * 2 degrees latitude | 1x1 degree | 1x1 degree | 1x1 degree | 1x1 degree |
| Temporal resolution | daily | monthly | monthly | monthly | monthly |
| Atmospheric CO2 | spatially and temporally varying field based on atmospheric $CO_2$ data from 156 stations (Jena CarboScope atmospheric inversion sEXTALL_v2020) | atmospheric pCO2_wet calculated from the NOAA ESRL marine boundary layer xCO2 and the NCEP sea level pressure with the moisture correction by Dickson et al 2007 (details and references can be obtained from Appendix A3 in Landschützer et al 2013) | Spatially and monthly varying fields of atmospheric pCO2 computed from $CO_2$ mole fraction (Chevallier, 2013), and atmospheric dry-air pressure which is derived from monthly surface pressure (ERA5) and water vapour pressure fitted by Weiss and Price (1980) | Mole fraction of $CO_2$ from NOAA marine boundary layer product interoplated longitudinally onto ERA5 monthly mean sea level pressure (MSLP). A water vapour pressure correction is applied to MSLP using the equation from Dickson et al. (2007). | Atmospheric pCO2 (wet) calculated from NOAA marine boundary layer XCO2 and NCEP sea level pressure, with pH2O calculated from Cooper et al, 1998. (2019 XCO2 marine boundary values were not available at submission so we used preliminary values, estimated from 2018 values and increase at Mauna Loa.) |
| Total ocean area on native grid (km2) | 3.63E+08 | 3.21E+08 | 3.21E+08 | 3.35E+08 | 3.48E+08 |

**Table A4. Comparison of the inversion set up and input fields for the atmospheric inversions. Atmospheric inversions see the full CO2 fluxes, including the anthropogenic and pre-industrial fluxes. Hence they need to be adjusted for the pre-industrial flux of CO2 from the land to the ocean that is part of the natural carbon cycle before they can be compared with SOCEAN and SLAND from process models. See Table 4 for references.**

| | CarbonTracker Europe (CTE) | Jena CarboScope | Copernicus Atmosphere Monitoring Service (CAMS) | UoE | MIROC | NISMON-CO2 |
|---|---|---|---|---|---|---|
| **Version number** | CTE2020 | sEXTocNEET_v2020 | v19r1 | in-situ | 4 | |
| **Observations** | | | | | | |
| Atmospheric observations | Hourly resolution (well-mixed conditions) obspack GLOBALVIEWplus v5.0 and NRT_v5.2 (a) | Flasks and hourly (outliers removed by 2-sigma criterion) | Daily averages of well-mixed conditions - OBSPACK GLOBALVIEWplus v5.0& NRT v5.2, WDCGG, RAMCES and ICOS ATC | Hourly resolution (well-mixed conditions) obspack GLOBALVIEW plus v5.0 and NRT_v 5.2 (a) | 34 surface sites from obspack GLOBALVIEW plus v5.0 and NRT_v 5.2 (a) | Hourly resolution (well-mixed conditions) obspack GLOBALVIEW plus v5.0 and NRT_v 5.2 (a) + NIES observations |
| Period covered | 2001-2019 | 1957-2019 | 1979-2019 | 2001-2019 | 1996-2019 | 1990-2019 |
| **Prior fluxes** | | | | | | |
| Biosphere and fires | SIBCASA biosphere (b) with 2019 climatological, GFAS fires | No prior | ORCHIDEE (climatological), GFEDv4.1 & GFAS after 2019 | CASA v1.0, climatology after 2016 & GFED4.0 | CASA | VISIT & GFEDv 4.1s |

| | | | | | | |
|---|---|---|---|---|---|---|
| Ocean | oc_v1.7 (Rodenbeck et al., 2014) with updates, 2019 climatology + anomalies from oc_v2020 | oc_v2020 (Rodenbeck et al., 2014) with updates | CMEMS Copernicus ocean fluxes (Denvil-Sommer et al., 2019), with updates | Takahashi climatology | Takahashi climatology | JMA global ocean mapping (Iida et al., 2015) |
| Fossil fuels | GridFED v2020 (Jones et al., 2020) | GridFED v2020 (Jones et al., 2020) | GridFED v2020 (Jones et al., 2020) | ODIAC v2016, after 2015 constant | GridFED v2020 (Jones et al., 2020) | GridFED v2020 (Jones et al., 2020) |
| **Transport and optimization** | | | | | | |
| Transport model | TM5 | TM3 | LMDZ v6 | GEOS-CHEM | ACTM | NICAM-TM |
| Weather forcing | ECMWF | NCEP | ECMWF | MERRA 2 | JRA55 | JRA55 |
| Horizontal Resolution | Global: 3° x 2°, Europe: 1° x 1°, North America: 1° x 1° | Global: 4° x 5° | Global: 3.75° x 1.875° | Global: 4° x 5° | Global: 2.8° x 2.8° | isocahedral gl5: ~225km mx225 km |
| Optimization | Ensemble Kalman filter | Conjugate gradient (re-ortho-normalization) (c) | Variational | Ensemble Kalman filter | Matrix inversion with 84 big regions | Variational |
| a (GLOBALVIEW, 2020;Carbontracker Team, 2020) | | | | | | |
| b (van der Velde et al., 2014) | | | | | | |
| c ocean prior not optimised | | | | | | |

**Table A5** Attribution of fCO2 measurements for the year 2019 included in SOCATv2020 (Bakker et al., 2016, 2020) to inform ocean pCO2-based flux products.

| Platform | Regions | No. of samples | Principal Investigators | No. of data sets | Platform type |
|---|---|---|---|---|---|
| *Allure of the Seas* | Tropical Atlantic | 110103 | Wanninkhof, R.; Pierrot, D. | 46 | Ship |
| *Atlantic Condor* | North Atlantic | 5051 | Wallace, D.; Atamanchuk, D. | 1 | Ship |
| *Atlantic Explorer* | North Atlantic | 24534 | Bates, N. R. | 19 | Ship |
| *Aurora Australis* | Southern Ocean | 24269 | Tilbrook, B. | 2 | Ship |
| *Bell M. Shimada* | North Pacific | 20176 | Alin, S.; Feely, R. A. | 6 | Ship |
| *Bjarni Saemundsson* | North Atlantic | 17364 | Benoit-Cattin, A.; Ólafsdóttir, S. R. | 3 | Ship |
| *Bluefin* | North Pacific, tropical Pacific | 40110 | Alin, S. R.; Feely, R. A. | 6 | Ship |
| *Cap San Lorenzo* | North Atlantic, tropical Atlantic | 17496 | Lefèvre, N. | 4 | Ship |
| CB-06_125W_43N | North Pacific | 223 | Sutton, A.; Hales, B. | 1 | Mooring |
| *Colibri* | North Atlantic; tropical Atlantic | 27823 | Lefèvre, N. | 5 | Ship |
| *Columbia* | North Pacific | 76458 | Evans, W.; Lebon, G. T.; Harrington, C. D.; Bidlack, A. | 1 | Ship |
| *Discovery* | North Atlantic | 1457 | Kitidis, V. | 1 | Ship |
| *Equinox* | Tropical Atlantic | 84273 | Wanninkhof, R.; Pierrot, D. | 41 | Ship |
| *Finnmaid* | North Atlantic | 144037 | Rehder, G.; Glockzin, M.; Bittig, H. C. | 3 | Ship |
| *Flora* | North Atlantic, tropical Atlantic, tropical Pacific | 58550 | Wanninkhof, R.; Pierrot, D. | 21 | Ship |
| *G.O. Sars* | North Atlantic | 93203 | Skjelvan, I. | 11 | Ship |
| *Gordon Gunter* | North Atlantic | 48162 | Wanninkhof, R.; Pierrot, D. | 9 | Ship |
| *Gulf Challenger* | North Atlantic | 6072 | Salisbury, J.; Vandemark, D.; Hunt, C. | 6 | Ship |
| *Healy* | North Pacific, Arctic | 28988 | Takahashi, T.; Sweeney, C.; | 2 | Ship |

| | | | Newberger, T.; Sutherland S. C.; Munro, D. R. | | |
|---|---|---|---|---|---|
| *Henry B. Bigelow* | North Atlantic | 66186 | Wanninkhof, R.; Pierrot, D. | 12 | Ship |
| *Investigator* | Indian Ocean, South Pacific, Southern Ocean | 126943 | Tilbrook, B. | 7 | Ship |
| *James Clark Ross* | North Atlantic, Southern Ocean | 10305 | Kitidis, V. | 3 | Ship |
| *Keifu Maru II* | North Pacific, Tropical Pacific | 8935 | Kadono, K. | 6 | Ship |
| *Laurence M. Gould* | Southern Ocean | 38380 | Sweeney, C.; Takahashi, T.; Newberger, T.; Sutherland, S. C.; Munro, D. R. | 4 | Ship |
| *Malizia* | North Atlantic | 88495 | Landschützer, P.; Tanhua, T. | 3 | Ship |
| *Marion Dufresne* | Indian, Southern oceans | 9107 | Lo Monaco, C.; Metzl, N.; Tribollet, A. | 2 | Ship |
| *New Century 2* | North Pacific, tropical Pacific, North Atlantic | 28434 | Nakaoka, S.-I. | 13 | Ship |
| *Newrest - Art and Fenetres* | North Atlantic, tropical Atlantic | 37651 | Tanhua, T.; Landschützer, P. | 2 | Ship |
| *Nuka Arctica* | North Atlantic | 65462 | Becker, M.; Olsen, A. | 20 | Ship |
| *Oscar Dyson* | North Pacific | 30373 | Alin, S.; Feely, R. A. | 6 | Ship |
| *R/V Sikuliaq* | North Pacific, Arctic | 68540 | Takahashi, T.; Sweeney, C.; Newberger, T.; Sutherland, S. C.; Munro, D. R. | 11 | Ship |
| *Ronald H. Brown* | North Atlantic, tropical Atlantic | 25605 | Wanninkhof, R.; Pierrot, D. | 4 | Ship |
| *RVIB Nathaniel B. Palmer* | Southern Ocean | 22759 | Takahashi, T.; Sweeney, C.; Newberger, T.; Sutherland, S. C.; Munro D. R. | 2 | Ship |
| *Ryofu Maru III* | North Pacific, tropical Pacific | 9981 | Kadono, K. | 6 | Ship |
| *Simon Stevin* | North Atlantic | 26389 | Gkritzalis, T. | 6 | Ship |
| *Tangaroa* | Southern Ocean | 34 | Currie, K. I. | 2 | Ship |
| TAO110W_0N | Tropical Pacific | 180 | Sutton, A. | 1 | Mooring |
| *Thomas G. Thompson* | North Atlantic, tropical Atlantic, South Atlantic, Southern Ocean | 28965 | Alin, S.; Feely, R. A. | 3 | Ship |
| *Trans Carrier* | North Atlantic | 10767 | Omar, A. | 1 | Ship |
| *Trans Future 5* | North Pacific, tropical Pacific, South Pacific, | 16694 | Nakaoka, S.-I.; Nojiri, Y. | 16 | Ship |
| *Wakataka Maru* | North Pacific | 69661 | Tadokoro, K.; Ono, T. | 4 | Ship |
| Waveglider1741 | South Pacific | 2287 | Sutton, A. | 1 | ASV |

**Table A6. Aircraft measurement programs archived by Cooperative Global Atmospheric Data Integration Project (CGADIP, 2019) that contribute to the evaluation of the atmospheric inversions (Figure B3).**

| Site code | Measurement program name in Obspack | Specific doi | Data providers |
|---|---|---|---|
| AAO | Airborne Aerosol Observatory, Bondville, Illinois | | Sweeney, C.; Dlugokencky, E.J. |
| ACG | Alaska Coast Guard | | Sweeney, C.; McKain, K.; Karion, A.; Dlugokencky, E.J. |
| ALF | Alta Floresta | | Gatti, L.V.; Gloor, E.; Miller, J.B.; |
| AOA | Aircraft Observation of Atmospheric trace gases by JMA | | ghg_obs@met.kishou.go.jp |
| ACT | Atmospheric Carbon and Transport - America | | Sweeney, C.; Dlugokencky, E.J.; Baier, B; Montzka, S.; Davis, K. |
| BNE | Beaver Crossing, Nebraska | | Sweeney, C.; Dlugokencky, E.J. |
| BGI | Bradgate, Iowa | | Sweeney, C.; Dlugokencky, E.J. |
| CAR | Briggsdale, Colorado | | Sweeney, C.; Dlugokencky, E.J. |
| CMA | Cape May, New Jersey | | Sweeney, C.; Dlugokencky, E.J. |
| CON | CONTRAIL (Comprehensive Observation Network for TRace gases by AIrLiner) | http://dx.doi.org/10.17595/20180208.001 | Machida, T.; Matsueda, H.; Sawa, Y. Niwa, Y. |
| CRV | Carbon in Arctic Reservoirs Vulnerability Experiment (CARVE) | | Sweeney, C.; Karion, A.; Miller, J.B.; Miller, C.E.; Dlugokencky, E.J. |
| DND | Dahlen, North Dakota | | Sweeney, C.; Dlugokencky, E.J. |
| ESP | Estevan Point, British Columbia | | Sweeney, C.; Dlugokencky, E.J. |
| ETL | East Trout Lake, Saskatchewan | | Sweeney, C.; Dlugokencky, E.J. |
| FWI | Fairchild, Wisconsin | | Sweeney, C.; Dlugokencky, E.J. |
| GSFC | NASA Goddard Space Flight Center Aircraft Campaign | | Kawa, S.R.; Abshire, J.B.; Riris, H. |
| HAA | Molokai Island, Hawaii | | Sweeney, C.; Dlugokencky, E.J. |
| HFM | Harvard University Aircraft Campaign | | Wofsy, S.C. |
| HIL | Homer, Illinois | | Sweeney, C.; Dlugokencky, E.J. |
| HIP | HIPPO (HIAPER Pole-to-Pole Observations) | https://doi.org/10.3334/CDIAC/HIPPO_010 | Wofsy, S.C.; Stephens, B.B.; Elkins, J.W.; Hintsa, E.J.; Moore, F. |
| INX | INFLUX (Indianapolis Flux Experiment) | | Sweeney, C.; Dlugokencky, E.J.; Shepson, P.B.; Turnbull, J. |
| LEF | Park Falls, Wisconsin | | Sweeney, C.; Dlugokencky, E.J. |
| NHA | Offshore Portsmouth, New Hampshire (Isles of Shoals) | | Sweeney, C.; Dlugokencky, E.J. |
| OIL | Oglesby, Illinois | | Sweeney, C.; Dlugokencky, E.J. |
| PFA | Poker Flat, Alaska | | Sweeney, C.; Dlugokencky, E.J. |
| RBA-B | Rio Branco | | Gatti, L.V.; Gloor, E.; Miller, J.B. |

| | | | |
|---|---|---|---|
| RTA | Rarotonga | | Sweeney, C.; Dlugokencky, E.J. |
| SCA | Charleston, South Carolina | | Sweeney, C.; Dlugokencky, E.J. |
| SGP | Southern Great Plains, Oklahoma | | Sweeney, C.; Dlugokencky, E.J.; Biraud, S. |
| TAB | Tabatinga | | Gatti, L.V.; Gloor, E.; Miller, J.B. |
| THD | Trinidad Head, California | | Sweeney, C.; Dlugokencky, E.J. |
| TGC | Offshore Corpus Christi, Texas | | Sweeney, C.; Dlugokencky, E.J. |
| WBI | West Branch, Iowa | | Sweeney, C.; Dlugokencky, E.J. |

| Publication year | Fossil fuel emissions | | | LUC emissions | Reservoirs | | | Uncertainty & other changes |
|---|---|---|---|---|---|---|---|---|
| | Global | Country (territorial) | Country (consumption) | | Atmosphere | Ocean | Land | |
| 2006 (a) | | Split in regions | | | | | | |
| 2007 (b) | | | | ELUC based on FAO-FRA 2005; constant ELUC for 2006 | 1959-1979 data from Mauna Loa; data after 1980 from global average | Based on one ocean model tuned to reproduced observed 1990s sink | | ±1σ provided for all components |
| 2008 (c) | | | | Constant ELUC for 2007 | | | | |
| 2009 (d) | | Split between Annex B and non-Annex B | Results from an independent study discussed | Fire-based emission anomalies used for 2006-2008 | | Based on four ocean models normalised to observations with constant delta | First use of five DGVMs to compare with budget residual | |
| 2010 (e) | Projection for current year based on GDP | Emissions for top emitters | | ELUC updated with FAO-FRA 2010 | | | | |
| 2011 (f) | | | Split between Annex B and non-Annex B | | | | | |

| | | | | | | | |
|---|---|---|---|---|---|---|---|
| 2012 (g) | | 129 countries from 1959 | 129 countries and regions from 1990-2010 based on GTAP8.0 | ELUC for 1997-2011 includes interannual anomalies from fire-based emissions | All years from global average | Based on 5 ocean models normalised to observations with ratio | Ten DGVMs available for SLAND; First use of four models to compare with ELUC | |
| 2013 (h) | | 250 countries b | 134 countries and regions 1990-2011 based on GTAP8.1, with detailed estimates for years 1997, 2001, 2004, and 2007 | ELUC for 2012 estimated from 2001-2010 average | | Based on six models compared with two data-products to year 2011 | Coordinated DGVM experiments for SLAND and ELUC | Confidence levels; cumulative emissions; budget from 1750 |
| 2014 (i) | Three years of BP data | Three years of BP data | Extended to 2012 with updated GDP data | ELUC for 1997-2013 includes interannual anomalies from fire-based emissions | | Based on seven models | Based on ten models | Inclusion of breakdown of the sinks in three latitude bands and comparison with three atmospheric inversions |

| 2015 (j) | Projection for current year based Jan-Aug data | National emissions from UNFCCC extended to 2014 also provided | Detailed estimates introduced for 2011 based on GTAP9 | 136 | | Based on eight models | Based on ten models with assessment of minimum realism | The decadal uncertainty for the DGVM ensemble mean now uses ±1σ of the decadal spread across models |
|---|---|---|---|---|---|---|---|---|
| a Raupach et al. (2007) | | | | | | | | |
| b Canadell et al. (2007) | | | | | | | | |
| c Online | | | | | | | | |
| d Le Quéré et al. (2009) | | | | | | | | |
| e Friedlingstein et al. (2010) | | | | | | | | |
| f Peters et al. (2012b) | | | | | | | | |
| g Le Quéré et al. (2013), Peters et al. (2013) | | | | | | | | |
| h Le Quéré et al. (2014) | | | | | | | | |
| i Le Quéré et al. (2015b) | | | | | | | | |
| j Le Quéré et al. (2016) | | | | | | | | |

| Table A8 Relative changes in fossil CO2 emissions (EFOS) for the year 2020 to date and projections for the full year. Methods of the four approaches are described in Section 2.1.5 and Appendix C. | | | | | | | | | | |
|---|---|---|---|---|---|---|---|---|---|---|
| **2020 Year to Date fossil emissions** | | | | | | | | | | |
| | UEA | Priestley | Carbon Monitor | GCB | | Median | Average | Min | Max | Range |
| China (September) | -4.1 | -10.5 | -1.8 | 0.5 | | -2.9 | -4.0 | -10.5 | 0.5 | 11.0 |
| USA (September) | -11.1 | -17.0 | -13.4 | -12.1 | | -12.8 | -13.4 | -17.0 | -11.1 | 5.9 |
| EU27 (July) | -10.0 | -14.8 | -11.6 | -16.9 | | -13.2 | -13.3 | -16.9 | -10.0 | 6.8 |
| India (September) | -12.4 | -21.2 | -12.0 | -12.7 | | -12.6 | -14.6 | -21.2 | -12.0 | 9.2 |
| RoW (September) | -7.6 | -14.2 | -8.4 | | | -8.4 | -10.1 | -14.2 | -7.6 | 6.6 |
| World (September) | -7.6 | -14.1 | -7.6 | | | -7.6 | -9.8 | -14.1 | -7.6 | 6.6 |
| | | | | | | | | | | |
| **2020 projection of fossil emissions** | | | | | | | | | | |
| | UEA | Priestley | Carbon Monitor | GCB | | Median | Average | Min | Max | Range |
| China | -3.1 | -9.4 | -0.3 | 0.4 | | -1.7 | -3.1 | -9.4 | 0.4 | 9.8 |
| USA | -10.5 | -16.3 | -13.7 | -10.6 | | -12.2 | -12.8 | -16.3 | -10.5 | 5.8 |
| EU27 | -9.6 | -12.9 | -7.1 | -17.0 | | -11.3 | -11.7 | -17.0 | -7.1 | 9.9 |
| India | -9.7 | -19.2 | -8.5 | -8.1 | | -9.1 | -11.4 | -19.2 | -8.1 | 11.1 |
| Rest of the World | -7.1 | -13.0 | -7.7 | -6.4 | | -7.4 | -8.6 | -13.0 | -6.4 | 6.5 |
| World | -6.9 | -13.0 | -6.5 | -5.8 | | -6.7 | -8.0 | -13.0 | -5.8 | 7.2 |

| Table A9. Funder and grant number (where relevant) | Author Initials |
|---|---|
| Australia, Integrated Marine Observing System (IMOS) | BT |
| Australian Government as part of the Antarctic Science Collaboration Initiative program | AL |
| Australian Government National Environment Science Program (NESP) | JGC, VH |
| Belgium Research Foundation – Flanders (FWO) (grant number UA C130206-18) | TG |
| BNP Paribas Foundation through Climate & Biodiversity initiative, philanthropic grant for developments of the Global Carbon Atlas | PC |
| China, National Natural Science Foundation (grant no. 41975155) | XY |
| China, National Natural Science Foundation (grant no. 71874097 and 41921005) and Beijing Natural Science Foundation (JQ19032) | ZL |
| EC Copernicus Atmosphere Monitoring Service implemented by ECMWF on behalf of the European Commission | FC |
| EC Copernicus Marine Environment Monitoring Service implemented by Mercator Ocean | MG |
| EC H2020 (4C; grant no 821003) | PF, RMA, SS, GPP, MOS, JIK, SL, NG, PL, TI |
| EC H2020 (CHE; grant no 776186) | LF |
| EC H2020 (CRESCENDO: grant no. 641816) | RS, EJ, AJPS, TI |
| EC H2020 (CONSTRAIN: grant no 820829) | RS, PMF |
| EC H2020 European Research Council (ERC) Synergy grant (IMBALANCE-P; grant no. ERC-2013-SyG-610028) | TG |
| EC H2020 (QUINCY; grant no 647204) | SZ |
| EC H2020 project (VERIFY: grant no. 776810) | CLQ, GPP, JIK, RMA, MWJ, PC, NV |
| European Space Agency Climate Change Initiative ESA-CCI RECCAP2 project 655 (ESRIN/4000123002/18/I-NB) | PF, PC, SS, MOS |
| French Institut National des Sciences de l'Univers (INSU) and Institut Pau- Emile Victor (IPEV), Sorbonne Universités (OSU Ecce-Terra), TAAF (Terres Australes et Antarctique Françaises), Museum National d'Histoire Naturelle (MNHN) | NM |
| French Institut de Recherche pour le Développement (IRD) | NL, NM |
| German Integrated Carbon Observation System (ICOS), Federal Ministry for Education and Research (BMBF); BONUS INTEGRAL (BONUS Blue Ocean and Federal Ministry of Education and Research Grant No. 03F0773A) | HCB |
| German Helmholtz Association in its ATMO programme and the state Baden-Württemberg, Germany, through bwHPC | AA |
| German Helmholtz Young Investigator Group Marine Carbon and Ecosystem Feedbacks in the Earth System (MarESys), grant number VH-NG-1301 | JH |
| German Research Foundation's Emmy Noether Programme (grant no. PO1751/1-1) | JP |
| German Stifterverband für die Deutsche Wissenschaft e.V. in collaboration with Volkswagen AG | SB |
| Icelandic Ministry for the Environment and Natural Resources | ABC |
| Japan Global Environmental Research Coordination System, Ministry of the Environment (grant number E1751) | SN, TO |
| Japan Environment Research and Technology Development Fund of the Ministry of the Environment (JPMEERF20142001 and JPMEERF20172001) | YN, NC |

| | |
|---|---|
| Japan Meteorological Agency (JMA) | KK |
| Kuehne + Nagel | TT |
| Monaco Fondation Prince Albert II de Monaco (www.fpa2.org) | NM, TT |
| Monaco, Yacht Club de Monaco | TT |
| Norwegian Research Council (grant no. 270061) | JS |
| Norwegian ICOS Norway and OTC Research Infrastructure Project, Research Council of Norway (grant number 245927) | MB, IS, AO |
| Swiss National Science Foundation (grant no. 200020_172476) | SL |
| UK Natural Environment Research Council (SONATA: grant no. NE/P021417/1) | DRW |
| UK Natural Environment Research Council (NE/R015953/1; NE/N018095/1) | VK |
| UK Natural Environmental Research Council (NE/R016518/1) | PIP |
| UK Newton Fund, Met Office Climate Science for Service Partnership Brazil (CSSP Brazil) | AW, ER |
| UK Royal Society: The European Space Agency OCEANFLUX projects | AJW |
| USA Department of Agriculture, National Institute of Food and Agriculture (grants no. 2015-67003-23489 and 2015-67003-23485) | DLL |
| USA Department of Commerce, NOAA/OAR's Global Ocean Monitoring and Observation Program | RW, AS, SA, DP, NRB, DRM |
| USA Department of Commerce, NOAA/OAR's Ocean Acidification Program | RW, SA, AJS, DP |
| USA Department of Energy, Office of Science and BER prg. (grant no. DE-SC000 0016323) | AKJ |
| USA Department of Energy, SciDac award number is DESC0012972; IDS grant award number is 80NSSC17K0348 | LC, GH |
| USA NASA Interdisciplinary Research in Earth Science Program. | BP |
| US National Science Foundation (grant number 1903722) | HT |
| USA Princeton University Environmental Institute and the NASA OCO2 science team, grant number 80NSSC18K0893. | LR |
| ORNL is managed by UT-Battelle, LLC, for the US DOE under contract DE-AC05-00OR22725. | APW |
| **Computing resources** | |
| Norway UNINETT Sigma2, National Infrastructure for High Performance Computing and Data Storage in Norway (NN2980K/NS2980K) | JS |
| The supercomputer systems of NIES (SX-Aurora) and MRI (FUJITSU Server PRIMERGY CX2550M5) | YN |
| MIROC4-ACTM inversion is run from JAMSTEC Super Computer system in coordination with Prabir Patra | NC |
| Japan National Institute for Environmental Studies computational resources | EK |
| TGCC under allocation 2019-A0070102201 made by GENCI | FC |
| UEA High Performance Computing Cluster, UK | DRW, CLQ |
| Supercomputing time was provided by the Météo-France/DSI supercomputing center. | RS, EJ |
| CarbonTracker Europe was supported by the Netherlands Organization for Scientific Research (NWO; grant no. SH-312, 17616) | WP |
| Deutsches Klimarechenzentrum (allocation bm0891) | JEMSN, JP |
| The Leibniz Supercomputing Centre provided computing time on its Linux-Cluster | KH |
| PRACE for awarding access to JOLIOT CURIE at GENCI@CEA, France | LB |

| The CESM project is supported primarily by the National Science Foundation (NSF). This material is based upon work supported by the National Center for Atmospheric Research, which is a major facility sponsored by the NSF under Cooperative Agreement No. 1852977. Computing and data storage resources, including the Cheyenne supercomputer (doi:10.5065/D6RX99HX), were provided by the Computational and Information Systems Laboratory (CISL) at NCAR. We thank all the scientists, software engineers, and administrators who contributed to the development of CESM2. | |
| --- | --- |
| | DLL |

**Figures and Captions**

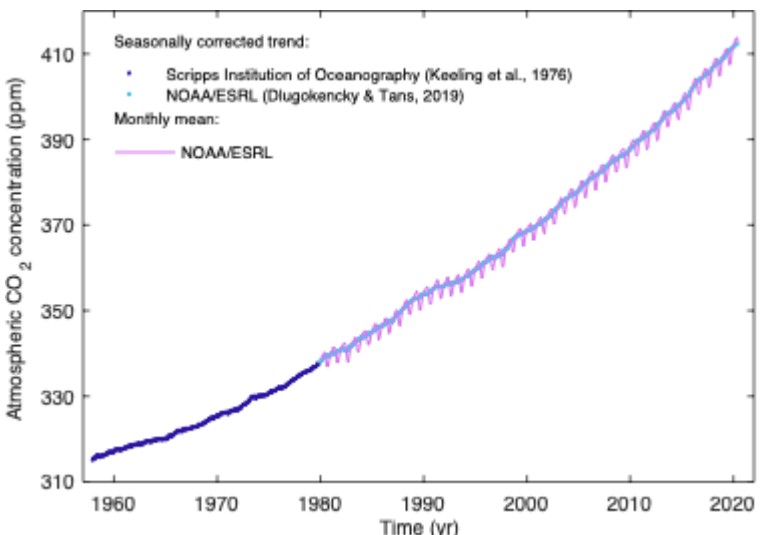

**Figure 1.** Surface average atmospheric $CO_2$ concentration (ppm). The 1980-2019 monthly data are from NOAA/ESRL (Dlugokencky and Tans, 2020) and are based on an average of direct atmospheric $CO_2$ measurements from multiple stations in the marine boundary layer (Masarie and Tans, 1995). The 1958-1979 monthly data are from the Scripps Institution of Oceanography, based on an average of direct atmospheric $CO_2$ measurements from the Mauna Loa and South Pole stations (Keeling et al., 1976). To take into account the difference of mean $CO_2$ and seasonality between the NOAA/ESRL and the Scripps station networks used here, the Scripps surface average (from two stations) was de-seasonalised and harmonised to match the NOAA/ESRL surface average (from multiple stations) by adding the mean difference of 0.542 ppm, calculated here from overlapping data during 1980-2012.

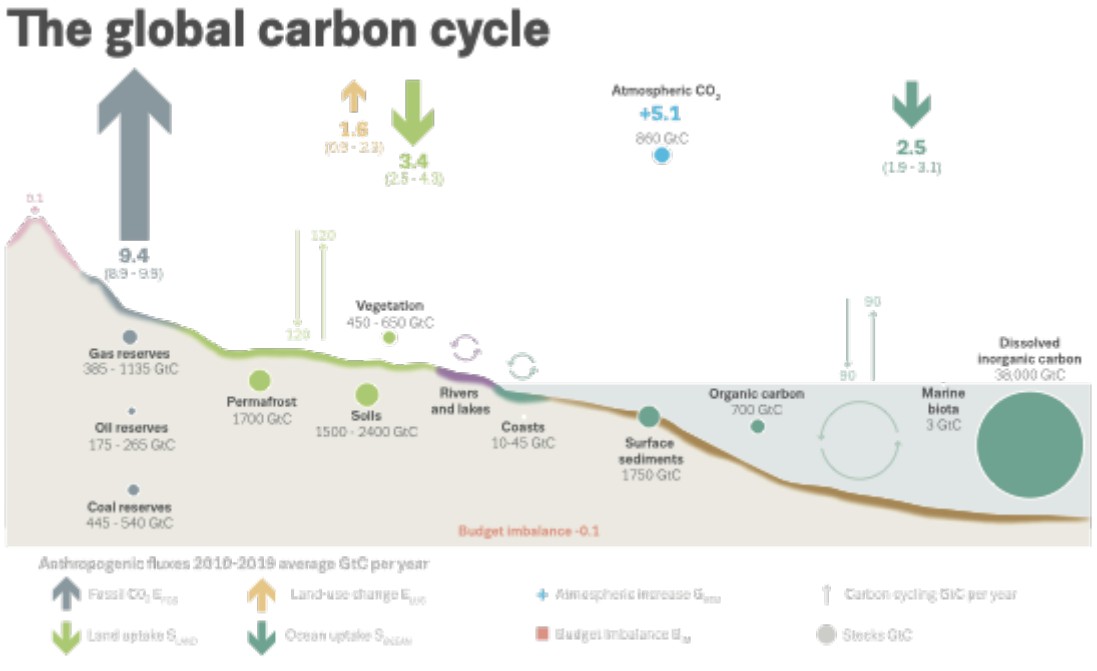

**Figure 2.** Schematic representation of the overall perturbation of the global carbon cycle caused by anthropogenic activities, averaged globally for the decade 2010-2019. See legends for the corresponding arrows and units. The uncertainty in the atmospheric $CO_2$ growth rate is very small ($\pm0.02$ Gt C yr$^{-1}$) and is neglected for the figure. The anthropogenic perturbation occurs on top of an active carbon cycle, with fluxes and stocks represented in the background and taken from Ciais et al. (2013) for all numbers, with the ocean gross fluxes updated to 90 GtC yr$^{-1}$ to account for the increase in atmospheric $CO_2$ since publication, and except for the carbon stocks in coasts which is from a literature review of coastal marine sediments (Price and Warren, 2016).

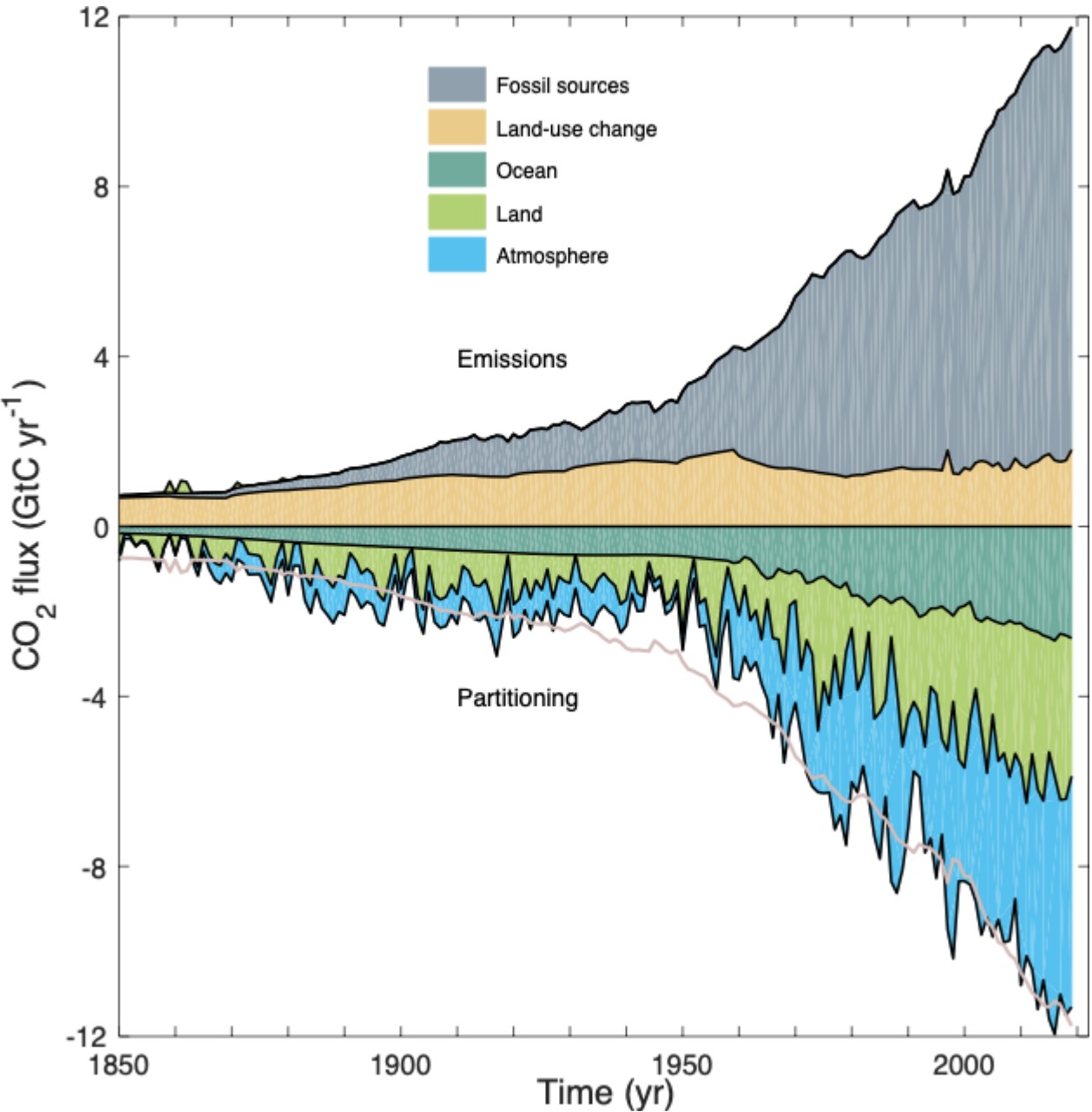

**Figure 3.** Combined components of the global carbon budget illustrated in Fig. 2 as a function of time, for fossil $CO_2$ emissions ($E_{FOS}$, including a small sink from cement carbonation; grey) and emissions from land-use change ($E_{LUC}$; brown), as well as their partitioning among the atmosphere ($G_{ATM}$; blue), ocean ($S_{OCEAN}$; turquoise), and land ($S_{LAND}$; green). The partitioning is based on nearly independent estimates from observations (for $G_{ATM}$) and from process model ensembles constrained by data (for $S_{OCEAN}$ and $S_{LAND}$), and does not exactly add up to the sum of the emissions, resulting in a budget imbalance which is represented by the difference between the bottom pink line (reflecting total emissions) and the sum of the ocean, land and atmosphere. All time series are in GtC yr$^{-1}$. $G_{ATM}$ and $S_{OCEAN}$ prior to 1959 are based on different methods. $E_{FOS}$ are

primarily from (Gilfillan et al. 2020), with uncertainty of about ±5% (±1σ); $E_{LUC}$ are from two bookkeeping models (Table 2) with uncertainties of about ±50%; $G_{ATM}$ prior to 1959 is from Joos and Spahni (2008) with uncertainties equivalent to about ±0.1-0.15 GtC yr$^{-1}$, and from Dlugokencky and Tans (2020) from 1959 with uncertainties of about ±0.2 GtC yr$^{-1}$; $S_{OCEAN}$ prior to 1959 is averaged from Khatiwala et al. (2013) and DeVries (2014) with uncertainty of about ±30%, and from a multi-model mean (Table 4) from 1959 with uncertainties of about ±0.5 GtC yr$^{-1}$; $S_{LAND}$ is a multi-model mean (Table 4) with uncertainties of about ±0.9 GtC yr$^{-1}$. See the text for more details of each component and their uncertainties.

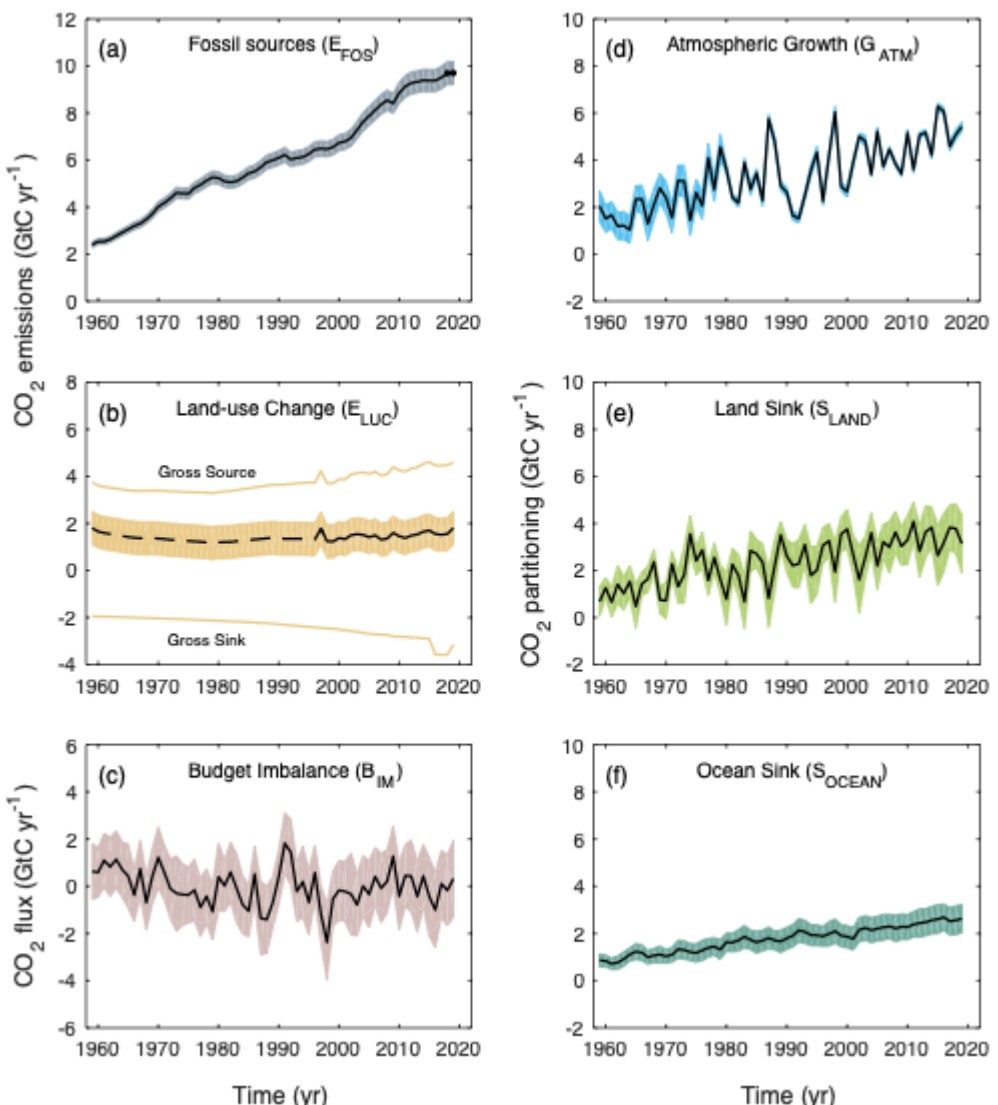

**Figure 4.** Components of the global carbon budget and their uncertainties as a function of time, presented individually for **(a)** fossil $CO_2$ emissions ($E_{FOS}$), **(b)** emissions from land-use change ($E_{LUC}$), **(c)** the budget imbalance that is not accounted for by the other terms, **(d)** growth rate in atmospheric $CO_2$ concentration ($G_{ATM}$), and **(e)** the land $CO_2$ sink ($S_{LAND}$, positive indicates a flux from the atmosphere to the land), **(f)** the ocean $CO_2$ sink ($S_{OCEAN}$, positive indicates a flux from the atmosphere to the ocean). All time series are in GtC yr$^{-1}$ with the uncertainty bounds representing $\pm 1\sigma$ in shaded colour. Data sources are as in Fig. 3. The black dots in **(a)** show values for 2018-2019 that originate from a different data set to the remainder of the data (see text). The dashed line in **(b)** identifies the pre-satellite period before the inclusion of emissions from peatland burning.

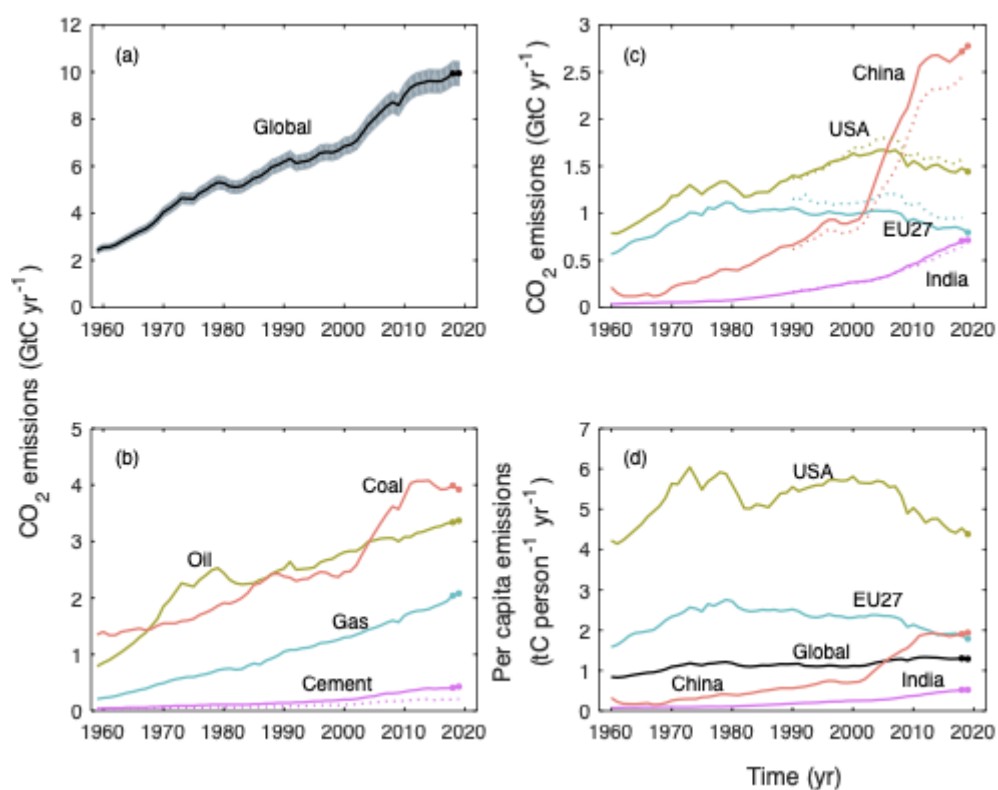

**Figure 5.** Fossil CO$_2$ emissions for **(a)** the globe, including an uncertainty of ± 5% (grey shading), and the emissions extrapolated using BP energy statistics (black dots), **(b)** global emissions by fuel type, including coal (salmon), oil (olive), gas (turquoise), cement (purple), and cement carbonation (dotted purple), and excluding gas flaring which is small (0.6% in 2013), **(c)** territorial (solid lines) and consumption (dashed lines) emissions for the top three country emitters (USA - olive; China - salmon; India - purple) and for the European Union (EU; turquoise for the 27 member states of the EU as of 2020), and **(d)** per-capita emissions for the top three country emitters and the EU (all colours as in panel **(c)**) and the world (black). In **(b-c)**, the dots show the data that were extrapolated from BP energy statistics for 2018-2019. All time series are in GtC yr$^{-1}$ except the per-capita emissions **(d)**, which are in tonnes of carbon per person per year (tC person$^{-1}$ yr$^{-1}$). Territorial emissions are primarily from Gilfillan et al. (2020) except national data for the USA and EU27 (the 27 member states of the EU) for 1990-2018, which are reported by the countries to the UNFCCC as detailed in the text; consumption-based emissions are updated from Peters et al. (2011a). See Section 2.1.1 for details of the calculations and data sources.

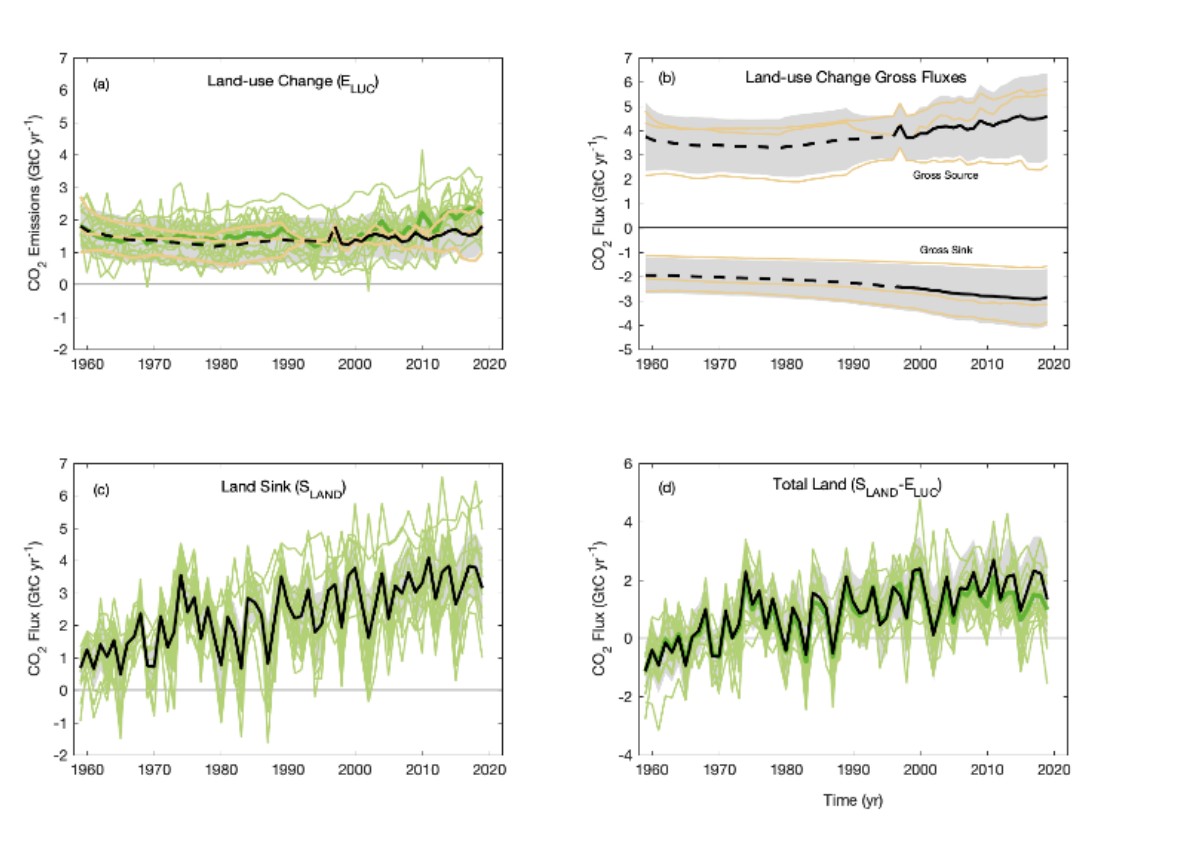

**Figure 6.** $CO_2$ exchanges between the atmosphere and the terrestrial biosphere as used in the global carbon budget (black with ±1σ uncertainty in grey shading), for **(a)** $CO_2$ emissions from land-use change ($E_{LUC}$). Estimates from the three bookkeeping models (brown lines) and the DGVM models (green) are shown individually, as is the multi-model mean of DGVM models (dark green). The dashed line identifies the pre-satellite period before the inclusion of peatland burning. (b) CO2 gross sinks (from regrowth after agricultural abandonment and wood harvesting) and gross sources (decaying material left dead on site and from products after clearing of natural vegetation for agricultural purposes, wood harvesting, and, for BLUE, degradation from primary to secondary land through usage of natural vegetation as rangeland, and emissions from peat drainage and peat burning). The sum of the gross sinks and sources is ELUC. Estimates from the three bookkeeping models (brown lines) are shown individually. **(c)** Land $CO_2$ sink ($S_{LAND}$) with individual DGVMs (green). **(d)** Total land $CO_2$ fluxes (**c minus a**) with individual DGVMs (green) and their multi-model mean (dark green).

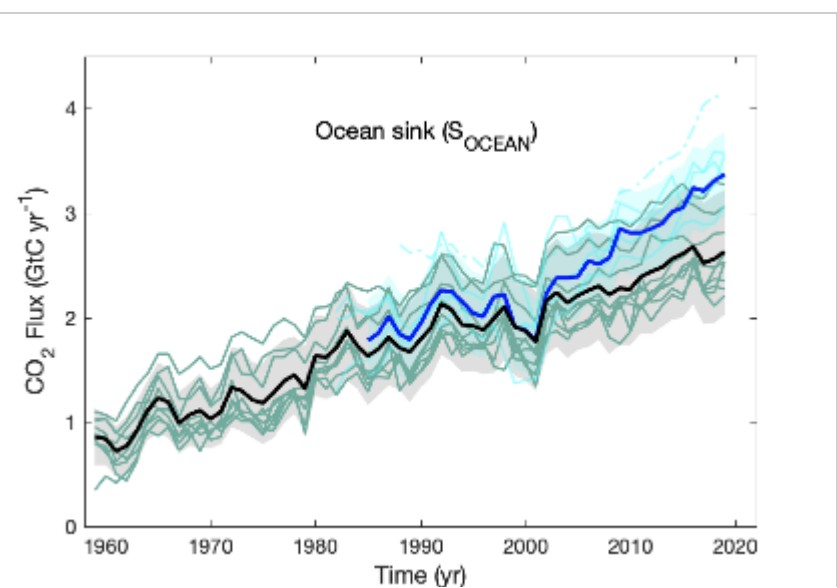

**Figure 7.** Comparison of the anthropogenic atmosphere-ocean $CO_2$ flux showing the budget values of $S_{OCEAN}$ (black; with $\pm1\sigma$ uncertainty in grey shading), individual ocean models (teal), and the ocean $pCO_2$-based flux products (ensemble mean in dark blue; with $\pm1\sigma$ uncertainty in light blue shading see Table 4, individual products in cyan, Watson et al as dashed-dotted line not used for ensemble mean). The $pCO_2$-based flux products were adjusted for the pre-industrial ocean source of $CO_2$ from river input to the ocean, which is not present in the ocean models, by adding a sink of 0.61 GtC yr$^{-1}$ to make them comparable to $S_{OCEAN}$ (see Section 2.7.3).

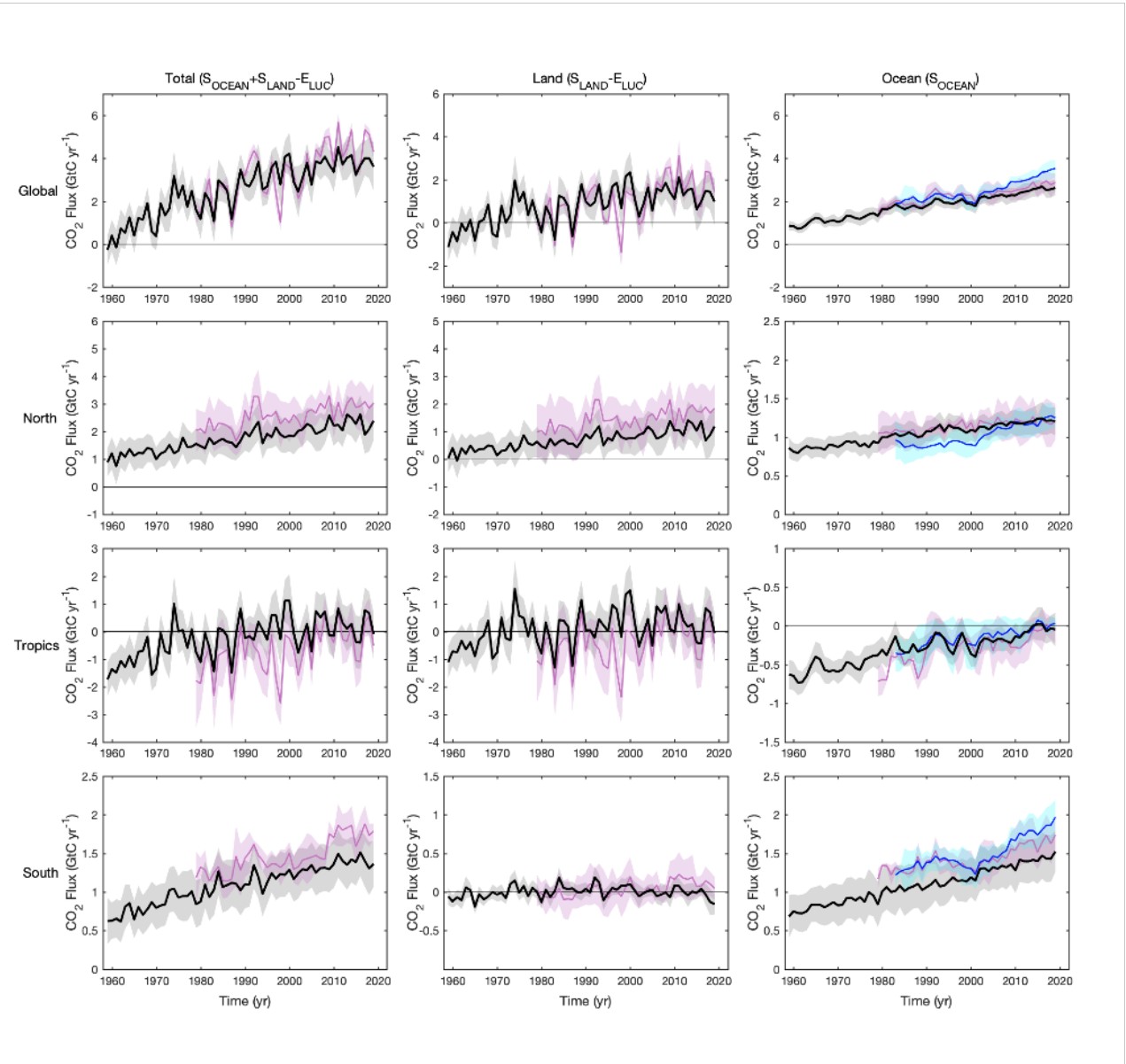

**Figure 8.** $CO_2$ fluxes between the atmosphere and the surface, $S_{OCEAN}$ and ($S_{LAND} - E_{LUC}$) by latitude bands for the (top) globe, (2nd row) north (north of 30°N), (3rd row) tropics (30°S-30°N), and (bottom) south (south of 30°S), and over (left) total ($S_{OCEAN} + S_{LAND} - E_{LUC}$), (middle) land only ($S_{LAND} - E_{LUC}$) and (right) ocean only ($S_{OCEAN}$). Positive values indicate a flux from the atmosphere to the land and/or ocean. Mean estimates from the combination of the process models for the land and oceans are shown (black line) with ±1σ of the model ensemble (grey shading). For total uncertainty, the land and ocean uncertainties are summed in quadrature. Mean estimates from the atmospheric inversions are shown (pink lines) with their ±1σ spread (pink shading). Mean estimates from the $pCO_2$-based flux products are shown for the ocean domain (dark blue lines)

with their $\pm 1\sigma$ spread (light blue shading). The global $S_{OCEAN}$ (upper right) and the sum of $S_{OCEAN}$ in all three regions represents the anthropogenic atmosphere-to-ocean flux based on the assumption that the preindustrial ocean sink was 0 GtC yr$^{-1}$ when riverine fluxes are not considered. This assumption does not hold on the regional level, where preindustrial fluxes can be significantly different from zero. Hence, the regional panels for $S_{OCEAN}$ represent a combination of natural and anthropogenic fluxes. Bias-correction and area-weighting were only applied to global $S_{OCEAN}$, hence the sum of the regions is slightly different from the global estimate (<0.08 GtC yr$^{-1}$).

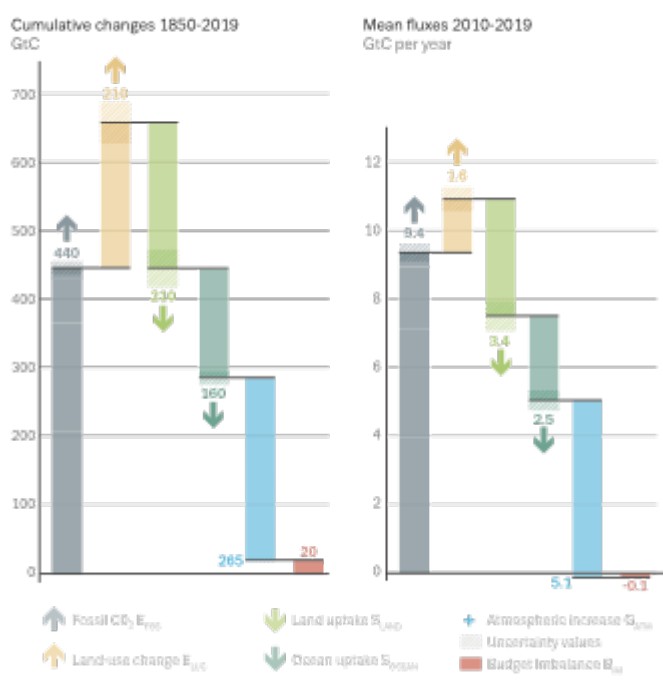

**Figure 9.** Cumulative changes during 1850-2019 and mean fluxes during 2010-2019 for the anthropogenic perturbation as defined in the legend.

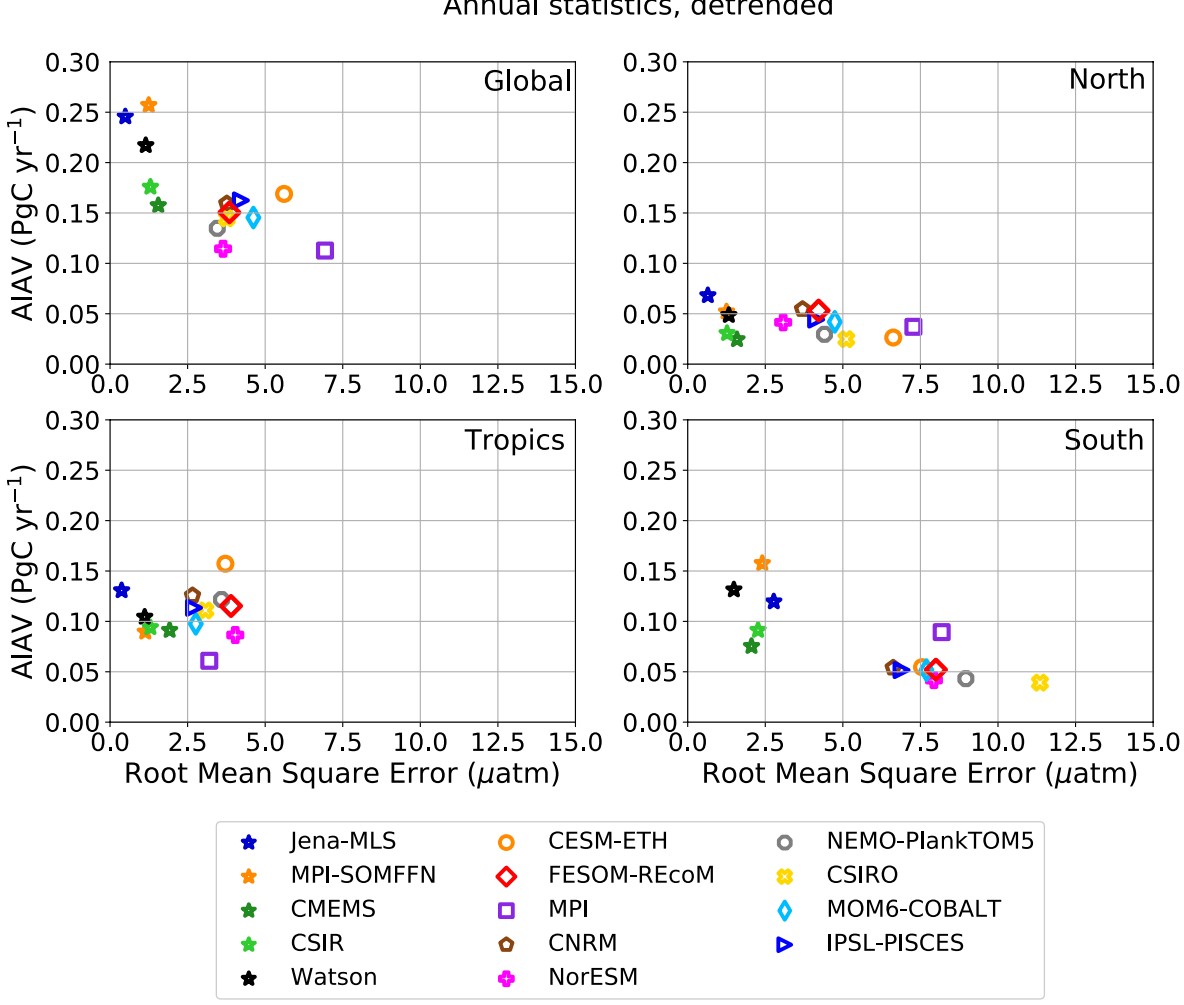

**Figure B1.** Evaluation of the GOBMs and flux products using the root mean squared error (RMSE) for the period 1985 to 2019, between the individual surface ocean pCO₂ estimates and the SOCAT v2020 database. The y-axis shows the amplitude of the interannual variability (A-IAV, taken as the standard deviation of a detrended time-series calculated as a 12-months running mean over the monthly flux time-series, Rödenbeck et al., 2015). Results are presented for the globe, north (>30°N), tropics (30°S-30°N), and south (<30°S) for the GOBMs (see legend circles) and for the pCO₂-based flux products (star symbols). The five pCO₂-based flux products use the SOCAT database and therefore are not fully independent from the data (see section 2.4.1).

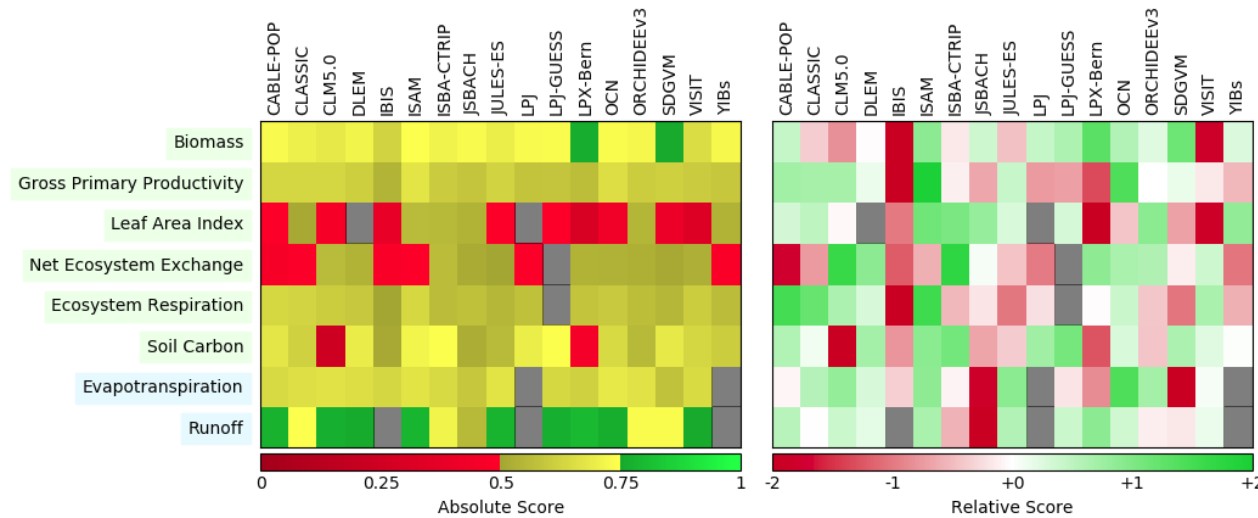

**Figure B2.** Evaluation of the DGVM using the International Land Model Benchmarking system (ILAMB; Collier et al., 2018) (left) absolute skill scores and (right) skill scores relative to other models. The benchmarking is done with observations for vegetation biomass (Saatchi et al., 2011; and GlobalCarbon unpublished data; Avitabile et al., 2016), GPP (Jung et al., 2010; Lasslop et al., 2010), leaf area index (De Kauwe et al., 2011; Myneni et al., 1997), net ecosystem exchange (Jung et al., 2010;Lasslop et al., 2010), ecosystem respiration (Jung et al., 2010;Lasslop et al., 2010), soil carbon (Hugelius et al., 2013;Todd-Brown et al., 2013), evapotranspiration (De Kauwe et al., 2011), and runoff (Dai and Trenberth, 2002). For each model-observation comparison a series of error metrics are calculated, scores are then calculated as an exponential function of each error metric, finally for each variable the multiple scores from different metrics and observational data sets are combined to give the overall variable scores shown in the left panel. Overall variable scores increase from 0 to 1 with improvements in model performance. The set of error metrics vary with data set and can include metrics based on the period mean, bias, root mean squared error, spatial distribution, interannual variability and seasonal cycle. The relative skill score shown in the right panel is a Z-score, which indicates in units of standard deviation the model scores relative to the multi-model mean score for a given variable. Grey boxes represent missing model data.

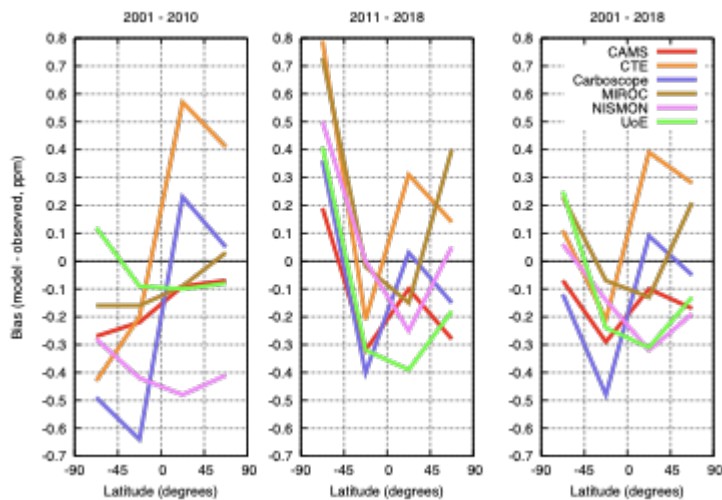

**Figure B3.** Evaluation of the atmospheric inversion products. The mean of the model minus observations is shown for four latitude bands in three periods: (left) 2001-2010, (centre) 2011-2018, (right) 2001-2018. The four models are compared to independent CO₂ measurements made onboard aircraft over many places of the world between 2 and 7 km above sea level. Aircraft measurements archived in the Cooperative Global Atmospheric Data Integration Project (CGADIP, 2020) from sites, campaigns or programs that cover at least 9 months between 2001 and 2018 and that have not been assimilated, have been used to compute the biases of the differences in four 45∘ latitude bins. Land and ocean data are used without distinction.

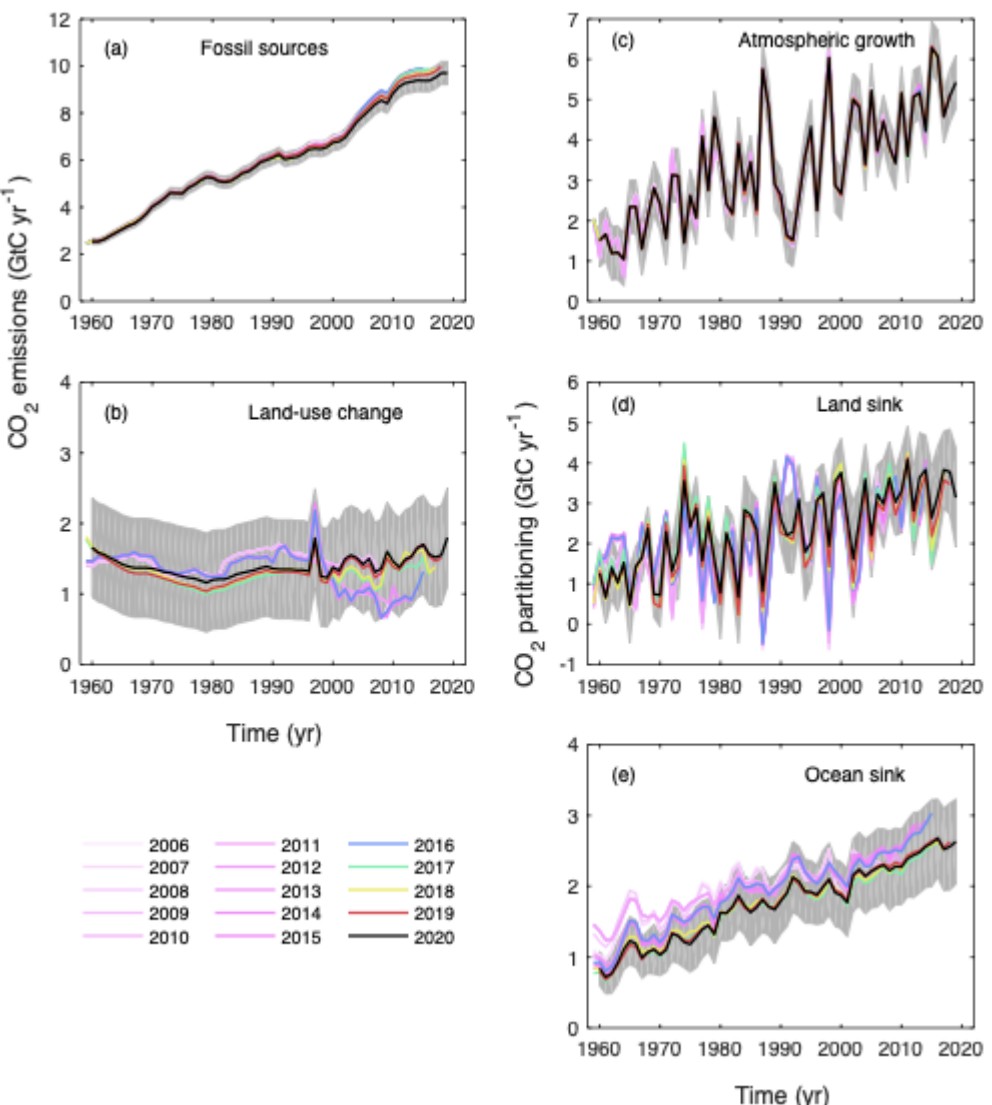

**Figure B4.** Comparison of global carbon budget components released annually by GCP since 2006. $CO_2$ emissions from **(a)** fossil $CO_2$ emissions ($E_{FOS}$), and **(b)** land-use change ($E_{LUC}$), as well as their partitioning among **(c)** the atmosphere ($G_{ATM}$), **(d)** the land ($S_{LAND}$), and **(e)** the ocean ($S_{OCEAN}$). See legend for the corresponding years, and Tables 3 and A7 for references. The budget year corresponds to the year when the budget was first released. All values are in GtC yr$^{-1}$. Grey shading shows the uncertainty bounds representing ±1σ of the current global carbon budget.

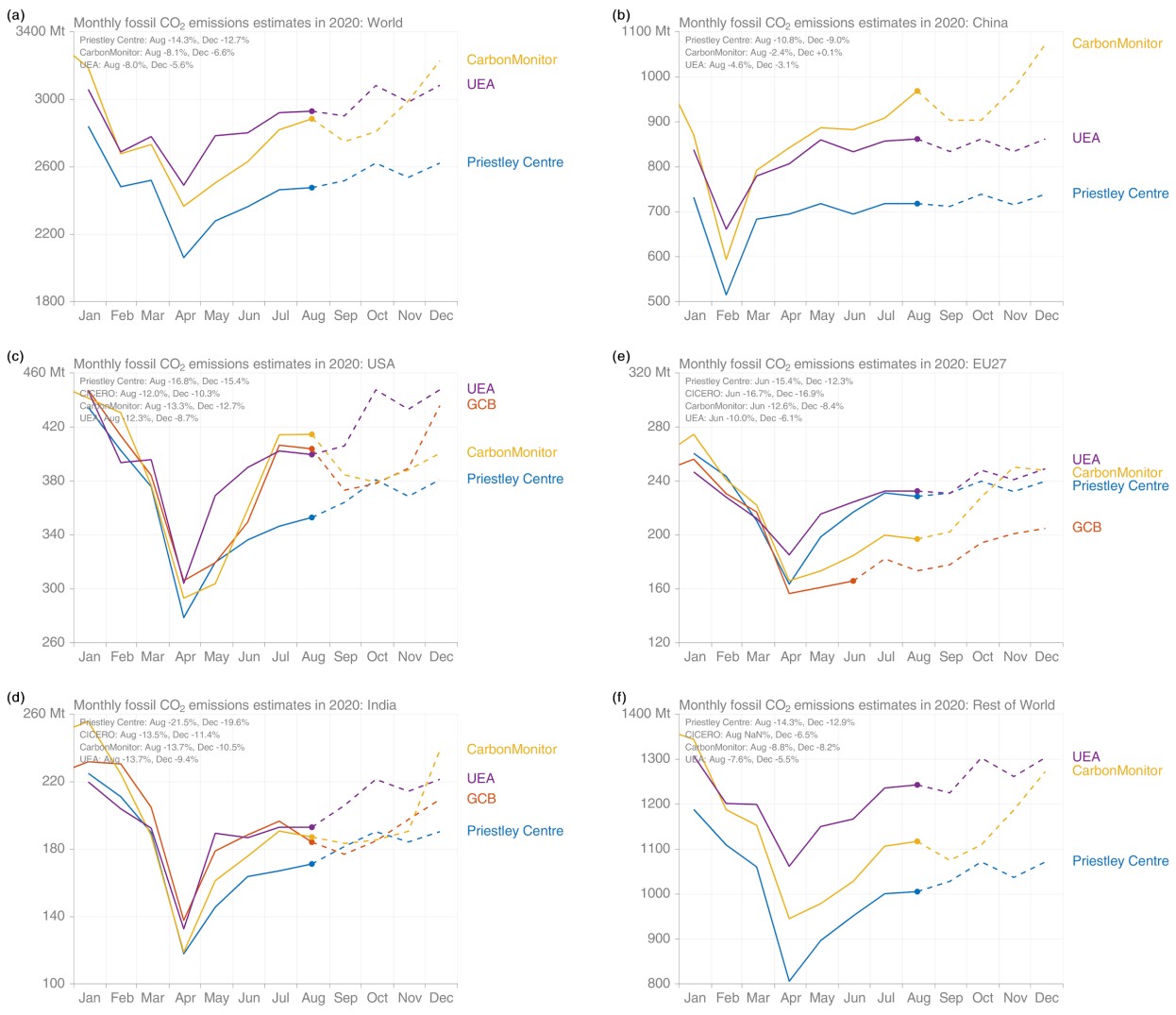

**Figure B5.** Monthly 2020 fossil CO$_2$ emission based on year-to-date data (solid lines) and projections (dashed lines) following four available approaches for (a) total world, (b) China, (c) USA, (d) European Union, (e) India, and (f) the rest of the world. Methods of the four approaches are described in Section 2.1.5 and Appendix C.

**Appendix C.** Supplementary Information

**Details of the Global Carbon Budget projection method**

**China**: The method for the projection uses: (1) the sum of monthly domestic production of raw coal, crude oil, natural gas and cement from the National Bureau of Statistics (NBS, 2020a), (2) monthly net imports of coal, coke, crude oil, refined petroleum products and natural gas from the General Administration of Customs of the People's Republic of China (2019); and (3) annual energy consumption data by fuel type and annual production data for cement from the NBS, using data for 2000-2018 (NBS, 2019), with the growth rates for 2019 taken from official preliminary statistics for 2019 (NBS, 2020a, 2020b). We estimate the full-year growth rate for 2020 using a Bayesian regression for the ratio between the annual energy consumption data (3 above) from 2014 through 2019, and monthly production plus net imports through August of each year (1+2 above). The uncertainty range uses the standard deviations of the resulting posteriors. Sources of uncertainty and deviations between the monthly and annual growth rates include lack of monthly data on stock changes and energy density, variance in the trend during the last three months of the year, and partially unexplained discrepancies between supply-side and consumption data even in the final annual data. The YTD estimate is made in the same way, but instead of regressing the ratio between historical monthly data for August and full-year annual data, monthly data for December is used instead, to produce regression results that capture the systematic differences between the monthly supply and annual consumption data, without the additional effect of projecting forward from August to the end of the year.

Note that in recent years, the absolute value of the annual growth rate for coal energy consumption, and hence total $CO_2$ emissions, has been consistently lower (closer to zero) than the growth or decline suggested by the monthly, tonnage-based production and import data, and this is reflected in the projection. This pattern is only partially explained by stock changes and changes in energy content, and it is therefore not possible to be certain that it will continue in any given year. For 2020 in particular, COVID-19-related lockdown and reopening in China, similar but delayed restrictions in major export markets, as well as unusual amounts of flooding and extreme weather during the summer months imply that seasonal patterns and correlations between supply, stock changes and consumption are likely to be quite different this year than in the previous years that the regression is based on. This adds a major but unquantified amount of uncertainty to the estimate.

**USA**: We use emissions estimated by the U.S. Energy Information Administration (EIA) in their Short-Term Energy Outlook (STEO) for emissions from fossil fuels to get both YTD and a full year projection (EIA, 2020). The STEO also includes a near-term forecast based on an energy forecasting model which is updated monthly (last update with preliminary data through August 2020), and takes into account expected temperatures , household expenditures by fuel type, energy markets, policies, and other effects. We combine this with our estimate of emissions from cement production using the monthly U.S. cement data from USGS for January-June 2020, assuming changes in cement production over the first part of the year apply throughout the year.

**India**: We use monthly emissions estimates for India updated from Andrew (2020) through August. These estimates are derived from many official monthly energy and other activity data sources to produce direct estimates of national $CO_2$ emissions, without the use of proxies. For purposes of comparison with other methods, we use a simple approach to extrapolating their observations by assuming the remaining months of the year change by the same relative amount compared to 2019 in the final month of observations.

**EU**: We use (1) monthly coal delivery data from Eurostat for January through June 2020 (Eurostat, 2020); (2) monthly oil and gas demand data for January through June from the Joint Organisations Data Initiative (JODI, 2020), with adjustments for deliveries to petrochemical industries using data from Eurostat (2020); and (3) cement production is assumed stable. For purposes of comparison with other methods, we use a simple approach to extrapolating their observations by assuming the remaining months of the year change by the same relative amount compared to 2019 in the final month of observations.

**Rest of the world**: This method only provides a full year projection. We use the close relationship between the growth in GDP and the growth in emissions (Raupach et al., 2007) to project emissions for the current year. This is based on a simplified Kaya Identity, whereby $E_{FOS}$ (GtC yr$^{-1}$) is decomposed by the product of GDP (USD yr$^{-1}$) and the fossil fuel carbon intensity of the economy ($I_{FOS}$; GtC USD$^{-1}$) as follows:

$$E_{FOS} = GDP \times I_{FOS} \tag{3}$$

Taking a time derivative of Equation (3) and rearranging gives:

$$\frac{1}{E_{FOS}} \frac{dE_{FOS}}{dt} = \frac{1}{GDP} \frac{dGDP}{dt} + \frac{1}{I_{FOS}} \frac{dI_{FOS}}{dt} \tag{4}$$

where the left-hand term is the relative growth rate of $E_{FOS}$, and the right-hand terms are the relative growth rates of GDP and $I_{FOS}$, respectively, which can simply be added linearly to give the overall growth rate.

The $I_{FOS}$ is based on GDP in constant PPP (Purchasing Power Parity) from the International Energy Agency (IEA) up to 2017 (IEA/OECD, 2019) and extended using the International Monetary Fund (IMF) growth rates through 2019 (IMF, 2020). Interannual variability in $I_{FOS}$ is the largest source of uncertainty in the GDP-based emissions projections. We thus use the standard deviation of the annual $I_{FOS}$ for the period 2009-2019 as a measure of uncertainty, reflecting a ±1σ as in the rest of the carbon budget.

**World**: This method only provides a full year projection. The global total is the sum of each of the countries and regions, but this year we additionally apply a GDP approach to the world to provide an additional consistency check (see Rest of World Description).