# Peer review of "Global Carbon Budget 2020"

_Earth System Science Data, 2020_

## Editor Comment (EC1) · David Carlson (Editor) · 2 Oct 2020

ESSD remains emphatically a free and open journal, consistent with Copernicus open-access mandates. On very rare occasions, due to extreme external interest, we conduct a closed discussion to prevent premature release of vital information. In this case, submission and evaluation materials, including all reviews and revisions, will become fully and openly accessible once we publish this paper.

---

## Referee Report (RR1)

**Review of the 2019 Global Carbon Budget by Han Dolman and Roxana Petrescu**

The authors are once again to be complimented on their work, the sheer breadth and number of data sources used is simple outstanding. As always, this will be a very useful resource for scientists (less so for policymakers, given the content, depth of treatment and length).

**Major comments**

We like the standardization of the inversions with a single a priori dataset for Fossil fuel emissions. This was much needed. The level of disagreement in the latitudinal between the inversions remains however large. The current version of the paper (as previously) however takes a rather lackluster attitude to this: unfortunately, not resolved. Given the emphasis on using inversions in the future using satellite data, it would not harm to include a few lines of thinking about this issue in the discussion.

That brings us to the discussion itself, which is very similar, almost the same as last year. It may be an idea to use the discussion to highlight the impact of changes made in this year's budget (the description of which is now hidden in the methods and results) to make it more relevant to this year's budget rather than sticking to the not so exiting repetition of last year general statements.

**Minor issues**

L 303. "developing countries like China". Better to change into something like countries with strongly developing economies like China.

L 443. you could add in brackets also the names of the external datasets for peat burning..

L 497. Only 2 Gt C for peat fires over almost 60 years seems minute amount. We remember estimates of peat fires after el-Nino's of almost similar order of magnitude.

L 532-534 If BLUE uses LUH2 and H&N2017 FRA/FAO and we have OSCAR using both approaches, averaging the three model results assuming that they are independent is not necessary the best approach,

Line 702: please reference only the IPCC report you use

L 905-908. These lines could be deleted. It just tells us what you do not do. Maybe better to integrate with the next bit to keep the focus.

L 1007. You go to some length estimating the loss of additional sink capacity, but you do not use it in the budget (L1203). Can you explain?

L 1047 land use change fluxes

line 1052 Should be rewritten a bit to make it more understandable. Suggestion:
"Emissions from newly added gross sources are on average 2-3 times larger than previous net emissions (X Tg C in GCB 2018), and increased from an average of 3.5 ± 1.2 GtC yr-1 for the decade of the 1960s to an average of 4.4 ± 1.6 GtC yr-1 during 2010-2019…"?

L 1196 H&N to add 2017

L 1597. True, but the tropical areas are probably even more undersampled.

L 1541. This is not new. Undersampling of the Southern Ocean has been an issue for many years. Can you rephrase to make it sound not as you discover something novel?

- in Table 5 and 6 we would add an extra column with numbers for the projections 2020, then next year you can compare the 2020 budget to this years' projected one

- Figure 6 caption we think is wrong, explanation for a,b,c doesn't match the sub-plots and d is missing (c should be d=c-a, b is not explained)

- Figure 9: if you show cumulative changes for 1850-2018 why not show also changes for last decade instead of mean flux?

---

## Referee Report (RR2)

Review 2020-286

Overall: an amazing urgent essential compilation presented with an excellent comprehensive manuscript. 'Chapeau' to authors.

Excellent product for ESSD as well. Please regard comments and suggestions that follow as small improvements to overall readability.

Line 85: "they  became the dominant source of anthropogenic emissions"

Line 114: "(including coasts and territorial seas) ??

Lines 116 to 119: "The global emissions and their partitioning among the atmosphere, ocean and land are in reality in balance, however due to imperfect spatial and/or temporal data coverage, errors in each estimate, and smaller terms not included in our budget estimate (discussed in Section 2.7), their sum does not necessarily add up to zero." Revise to:

Global emissions and their partitioning among the atmosphere, ocean and land are in reality in balance. Due to some combination of imperfect spatial and/or temporal data coverage, errors in each estimate, and smaller terms not included in our budget estimate (discussed in Section 2.7), their sum does not necessarily add up to zero."

Line 129: I do not know prevailing lingo, but "territorial" and "consumption-based" are not comparable. "Consumption-based estimates still rely on territorial boundaries? Better to write "production-based vs consumption-based"? Throughout the text you refer to countries or nations but rarely to territories? Or you refer to "national territories"? Reader does not encounter careful definition of "territorial emission inventories" until line 315. Some clean up and consistency needed here?

Line 167 'most recent' rather than "last"?

Line 201: "estimates of EFOS rely primarily" should be $E_{FOS}$?

Line 278 to 291, cement carbonation: good up-to-date discussion but Guo et al. is not cited in references?

Line 381: "in place for six weeks before they ease" By the time of publication we ay know that easing has not worked and that lockdowns - based on erratic or absent national policies - have resumed in many locations. Economic impacts uncertain everywhere; one does not want to see these authors or this product 'chase' political changes. Better to state impacts so far as known and documented but to stay away from Covid-19 projections?

Line 400, 401: a separate crowd-sourced tracking of aviation data currently in ESSD discussion (https://doi.org/10.5194/essd-2020-223, who knows how it got in ESSD and one scarcely knows if or how to credit it) shows a much steeper decline in aviation travel than mentioned here. Again, although they want to keep current and alert, we really do not want GCB authors trying to keep up with rapidly-moving hardly-certified external activity indicators? Better that they declare uncertainty - as they do elsewhere - rather than publish today what will change tomorrow? Alert but a bit more cautious; authors will know best approach.

Line 406: Do we need a reference to EDGAR here? (As you do later at line 900.) Or we assume readers find it in Liu et al. online? This entire section gets a bit speculative, a departure from past reliability of GCB?

Line 426 and several times following: If "Carbon Monitor" represents one of your reliable documented sources for 2020 emissions (e.g. appears frequently in text as well as in Table A8) we need a standardized reference?

Line 480 to 482 - meanwhile, FAO data and definitions also undergo update and - to a smaller degree - revision (e.g https://doi.org/10.5194/essd-2020-203); perhaps not yet valid for this edition of GCB but something to take account of in future versions?

Lines 493-494 - emissions from drained soils discussed in https://doi.org/10.5194/essd-2020-202?

Line 505: LUH2 never defined? I think you mean Hurtt et al. 2020 but not clear how one would access that reference? The landing page for the PCMDI DOI only lists a 2017 version plus the option for upates. No valid reviewed reference to LUH2?

Line 517: Likewise, FAO / FRA much used but never defined nor properly referenced.

Line 528: "anomalous fire season in Southeast Asia." Also (both 2019 and 2020) in Siberia, North America, Amazonia, etc. What seemed anomalous in the past now proves regular and expected albeit still unquantified? For both accounting through 2019 and projection for 2020, the fire term grows increasingly unknown and uncertain? How does growing uncertainty in fire emissions contribute to overall $E_{LUC}$ uncertainty?

Line 576: no definition of CRU nor of JRA although you use those acronyms frequently throughout this section.

Line 621, 622: "scale almost linearly with GFED over large areas (van der Werf et al., 2017)" out of date or no longer valid?

Line 632: authors have no doubt done expert assessment, but "pantropical fire emissions" in 2020 only two-thirds of 2019 seems counter to most reports? Give your readers some basis for confidence in this statement?

Line 723: SOCAT, should have been defined a few lines earlier?

Line 867: You have not explained $xCO_2$. For a broad range of readers, you will need to carefully explain all terms.

Line 880: Something weird here?

Line 1258 - New / updated information on SO sinks emerging recently and continually? Hard to keep a reliable annual budget going against the 'noise' of new ocean data, but possible changes in SO estimates might need a mention here? Or, wait until next version?

Line 1326 - Here a reader encounters "column $CO_2$ products" rather than (as earlier) $xCO_2$. Some clarification needed?

Lines 1338 to 1340, discussion of uncertainties in NH land sink: substantial redundancy here?

Line 1473, Section 3.4.2: Numeric errors here, corrected already by authors according to text supplied by editor? Someone needs to read the final (proof) version carefully to confirm final numbers.

Line 1521: "suggests we do not yet have a complete understanding of the underlying carbon cycle processes." The manuscript carries a necessary overall uncertainty: how much uncertainty of each component arises from reporting deficiencies and how much from missing processes? Here the authors seem to point to missing processes but much reporting earlier in the manuscript focussed on reporting (or, modeling) uncertainties. No hard line between weak reports and missing processes, but can authors give a clearer sense of where the problem lies? This sentence confuses rather than clarifies?  Discussion that follows in this paragraph perpetuates this duality: some improvements might come from "improving the underlying data and statistics", from "scrutiny of carbon variability in light of other Earth system data", and from "higher resolution and process knowledge at the regional level". If these expert authors truly do not know the best route toward improvement (reducing BIM), then say so explicitly. In which case the introductory sentence at line 1521, with its apparent focus on processes rather than underlying data, remains misleading at best? This reader very much appreciates subsequent discussion of uncertainties in southern ocean, in NH LUC, etc, as well as good estimates of how long (decade or decades) one would need to detect a change in a statistically-robust manner.  Lines 1581 and 1582 belong in the abstract as well, to alert readers who only browse?

---

## Editor Decision (ED1)

Verbatim text from two reviews compiled here.

**Review #3**, Tom Oda. Page and line numbers refer to track changes version; reviewer assumed line number =1 at the top of each page.

Dear the authors of the manuscript,

It is my pleasure to have an opportunity to review this manuscript. I do my review for ESSD manuscripts following the guideline summarized by Carlson and Oda (2018) ESSD. The 2018 guideline should tell you where I/my comments came from. Given the nature of ESSD, most of my comments go to the method section pointing out the information that I feel missing. Below is a summary of my comments. You will also see a list of my line-by-line comments. Hope my review is useful.

Sincerely,
Tomohiro Oda (toda@usra.edu)

1 Data description
While I imagine the amount of effort dedicated to compile this budget/manuscript, I feel the level of the information is not meeting the ESSD's standard. First, please improve the data/model tables. Tables should include name, version, data source (where we can get the data, w DOI), data citation (this is different than data source), etc of the items (e.g. data and models) used in this manuscript. For the data that are not yet published, the authors should provide enough information to allow readers to understand the data. This is the basis requirement. ESSD will be hosting this manuscript/paper not for carbon cycle scientists, but also for people in the broader Earth science community. Perhaps also for someone who randomly pick up this manuscript w/o any strong Earth Science background. ESSD papers need to be useful for such readers and the described data needs to remain available (see more in the 2018 guideline).

2 Summary/Rationale/Justification of the approaches
This is partially done in some sections, but we would like to see the summary of what are covered and what are not. I see the authors did a wonderful job to collect the all of the results and synthetize them. However, probably the authors would agree with me that there should be something not included in this manuscript. I use as atmospheric inversions. It is totally fair to use a subset of inversion models available in the community. I would like to suggest the authors to just list model inclusion criteria and expand discussions including what are not covered in this manuscript. For example, some highly relevant papers such as Crowell et al (2019) (OCO-2 inversion comparison) and Schuh et al. (2019) must be worth discussing in this manuscript. I thought the authors did a great job in the DVGM section. It was concise and short (given the amount of underlying info). Yet it has the basic info that I wanted to see. For some sections, I also thought it would be great to have a short review of what types of the approaches are taken to examine a flux component and tell us if you have them all in this report or not. That would help us reader to recognize the position of this synthesis report in the big picture of the carbon cycle science.

3 Uncertainty analysis
The authors did great job in listing the potential sources of uncertainties. I would like to suggest the authors to discuss the sources in more quantitative way. I can imagine some of the uncertainty sources are tough to assess. However, at least, the authors should give a try to convince us that the uncertainties discussed should be small enough and the results remain robust.

4 Summary of advance in our understanding of the global carbon cycle
The authors kind of did this, but I thought it would be great to have a summary section (or as a part of the abstract/conclusion) of the advance in our understanding of the global carbon cycle (excluding the methodological differences) beyond the budget breakdown (I understand this is the main thing, though).  For example, the authors mentioned the confidence levels for the budget components.  Are the levels are changing?  Since probably the authors are placed in the best position to summarize the current understanding of the carbon cycle science, I would like the authors to highlight a bit more of the state-of-the-art science results collected.  I believe it would be a benefit for the general scientist readers.  Also, it would make this ESSD manuscript accessible to the random people who just googled "global carbon budget" and found this manuscript.

P5, L5: Describe -> Describe and synthesize?
P5, L6: five major components, such as fossil fuel emissions (FOS), Land-use Change (LUC), G_atm, S_ocean and S_land, This is pretty obvious for us carbon cycle scientist, but maybe not for the ones who just randomly picked up this manuscript w/o any carbon cycle background.
P5, L24: COVID-19, spell out.

P6, L5: and gas flares?

P7, L20: introduce ppm

P8, L1: any other references? Maybe RECCAP reports/papers?
P8, L1: The IPCC methodology, which IPCC methodology do you refer to?
P8, L21: Citation for AR5
P8, L14: our estimates, which are mostly annual basis? (w some exceptions)

P9, L1: detailed description of the data sets, the data table does not seem to meet ESSD's standard.  For example, data description, data source(s) and data citation(s) are only insufficiently provided.
P9, P7: citation for AR6
P9, P17: I believe this should be the best synthesis effort. However, it is still true that this was done by a subset of scientists in the research community.  I feel this manuscript should mention the criteria for data and model collection.
P9, L20: ESSD requests to provide data sources in order to ESSD data users/ to examine original data.  In principle (ideally), the data sets used in an ESSD paper needs to be available with a DOI (see Carlson and Oda, 2018).  Many of the model output do not seem to be accessible for ESSD data users.

P10, L7: The estimates of EFOS -> The estimates of E_FOS in this study, there are variants of E_FOS emission calculations.  I assume this sentence refers to the reference approach (if we define it using the IPCC terminology)
P10, L10: peat fuel?
P10, L21: CDIAC, spell out
P10, L13: UNFCCC (spell out) national inventor reports, did you take estimate based on the reference approach (or sectoral approach)?
P14, L14: Most accurate -> the best estimates?  Inventories can't assure their accuracy by themselves.  Also, UNFCCC inventories from countries could be challenging to use together as they are not compiled in a globally systematic way like EDGAR does.  Please not following the IPCC guideline does not assure the accuracy.
P10, L17: BP, yes, but the BP stats does not offer data transparency and traceability that we would like to see in ESSD papers.

P10, L24: UN -> UNSD?

P10, L27: Do those values remain the same as defined in Marland and Rotty (1984)?   How much differences from other reference approach estimates (UNFCCC if used) should we expect to see due the values specifically used in the CDIAC estimates?

P10, L30: WWII, years?

P11, L3: It sounds like these are ones calculated by the sectoral approach.  How exactly did the authors reallocate IPCC-defined sectoral emissions to fuel-based emissions?

P11, L8: CDIAC, is this originally due to the data collection by UNSD (or submission by India)? I believe India is not the only country that reports data using their own fiscal year period.  Why India?

P11, L20: which values exactly did you use for projecting 2017 emissions?  There are several values you could use to do the projection.

P12, P10: how about oversea territories for other counties, such as UK?  How did you spatially allocate those emissions?

P12, P18: Could you provide information more quantitatively?   How large the impact would be?

P12, L21: Why can you mitigate the issue by adding international bunker emissions?  Even adding the international bunker emissions to the sum of country totals, you would still have ~3% or so emission discrepancy (e.g. Oda et al. 2018 ESSD).

P12, L24: Can you provide more quantitative information (e.g. sink size, sink rate, lifetime..) in order to let us know how significant this could be in the global carbon budget?

P13, L7: How did you assess the uncertainty?  Is the method independent from Andres et al. (2014)?  Note the uncertainty range for the global total (6-10%, 90% confidence interval) mentioned Andres et al. (2012) is based on the assessment by Marland and Rotty (1984).  So you would not bump up the uncertainty for the 2018 and 2019 while those were projected.  Any comments?

P13, L18: Also see Andres et al. (2014) for regional uncertainty estimates

P13, L20: medium confidence, what does this mean?

P13, Did Ciais et al. (2013) confirm consistency between E_fos and direct observations?  This seems to sound stretch.

P14, L3: Are those export and import terms consistent with ones used in the E_fos calculation?

P14, L21: Does GTAP produce fuel-based estimates rather than sectoral-based estimates?

P14, L14: Is it fair if we say the uncertainty assessment might not fully capture a systematic part of the uncertainty as well as the model structural errors?

P15, L15: three separate <global> studies?

P15, L19: Provide some mode details of projection models in order to allow readers to assess the results.

P15, P22: proxy data, such as?

P15, P24: What exactly?  This is important as you are comparing different projections in the single figure.

P15, L25: absolute <daily> emission changes?

P16, L24: calibrated to the traffic data in Paris?

P17, L26: How do you assess the uncertainties associated with individual projection models? We do not see the uncertainty estimates and can't tell how we should interpret the agreement and disagreement among the estimates.

P18, P2: Are these consistent with the UNFCCC inventories? At least, some compatibility with the UNFCCC inventories? How can we assess these estimates in relation to the UNFCCC reported values?

P19, L4: Short model/data description and model output data source (DOI) seem to be missing in this manuscript.

P23, L1: Inclusion criteria for other components (data and models) should be listed, too.
P23, L25: Given the challenges mentioned, how would you assess the uncertainty associated with projected ELUC?

P25, L23: Ok, so this is relative confidence levels used in this study. Then, what would be the confidence level for E_Fos?

P26, L14: Brief text regarding the inclusion criteria for the GOBMs? We want to hear the justification for the use of the model ensemble for this carbon budet assessment.

P29, L4: Is 101% suggesting an ocean area definition difference among models?
P32, L2: Introduce xCO2
P32, L8: Inclusion criteria (like you did for the DGMV section)?
P32, L23: While focusing on large scale analysis, it is surprising that the authors did not discuss the use of satellite data using a detailed report by Crowell et al. (2018) ACP. I'd expect more discussion regarding the use of satellite data here (e.g. in-situ vs. satellites, GOSAT/OCO-2, land/ocean glint…). Probably the authors would agree with me that this subset of inversion model calculations might have failed to capture something. Also, I feel Table A4 is not providing enough details while we all agree that the results do have sensitivity to the settings.
P32, L28: transport, Schuh et al. (2019) GBC

P33, L3: Need more details since this is not publicly available yet. Grid resolution (0.1deg?)?, temporal resolution (monthly? hourly?)? Data source?
P33, L10: not GCB, but Crowell et al. (2018) used the same prior info.

P34, 2.7 Would you be able to provide more quantitative assessment for these missing components? Convince us that the GCB assessment is still robust regardless of those missing components.
P34, L25: See Nassar et al. (2010) GMD and Wang et al. (2010) ERL

P35, L7: What about gas flare emissions? Not as Ch4, but as CO2?

P37, 3: Would you be able to summarize new findings/revisions/changes from what previous publication reported (improvements of the GCB product/report/summary)?

P46, P12: Are Crowell et al. (2018) and Schuh et al. (2019) irrelevant here??

P50, P10: This is a very interesting statement. For E_eos, the use of fuel consumed is a very straightforward way to get CO2 emitted (fuel burned). It still does not assure the accuracy by itself. However, from methodological perspective, it seems to be fair to give more trust to monthly fuel stats than proxy data-based approaches given many assumptions made. In fact, emission seasonality is often constructed using fuel stats (e.g. Andres et al. 2012 Tellus). Also, as the name suggests, proxy data are "proxy" for CO2 emissions. Proxy data would beat fuel consumption while we are not entirely sure how the proxy data get collected and how the indices are developed. We do have a history of the use of monthly stats for estimating

monthly emissions, but we've not even examined the single use of proxy data for estimating emissions.

P50, L17: Here is a preprint by Zeng et al. that shows some model simulations using the Carbon Monitor emissions.

P50, L1: I thought it would be a good idea to start with the same emission value at the beginning of 2020. YTD emissions include the differences at the beginning of 2020. The projected emissions further include differences from the YTD emissions. At least, the authors could remove one of the sources of the differences to make the interpretation of the results a little bit easier (while it is not perfect by no means). I believe the authors are not very confident with the accuracy of the absolute emission values, but the relative changes.

P51, L20: But you could assign higher confidence to Chinese YTD emissions from the carbon monitor given the amount of the data collected/used?

P53, L19: And also future perspectives?

P56, L18: First, the authors should provide names, versions, data sources (where we can get them), and data citation (w DOI) of the items used in this manuscript as a list. This is the basic requirement for ESSD papers. I do see some tables, but those are missing key information mentioned above. Note data source and data citation are not the same thing. Many of the citations listed are data/model citations. Second, data policy. I am not sure this is in line with ESSD's philosophy. Why not? Please refer to Carlson and Oda (2018).

References: Please check the links if they are still active. Especially ones w/o a DOI.

Table 2 & 4 : Data sources are missing.

Table 9: In addition to Korsbakken et al. (2016), Guan et al. (2012) Nature CC, gita-ton gap. That has demonstrated potential systematic errors in emission estimates due to poor energy stats data.

Appendix A. These tables need to be improved. Can't read some of the contents at all.

Appendix B. Maybe better to change the X axis range to 0-15 (and 0-0.3 for Y axis) in order to show the points better? Points are heavily overlapping.

Figure B3. How can we interpret this evaluation?

Figure B5. As mentioned earlier, maybe it would be better to start from the same Jan emission. This manuscript does not provide enough text to explain why those lines started from different estimates. Readers need to go back to the papers published. ESSD wants papers to carry enough info to sufficiently understand the contents.

**Review #4** (anonymous):

~~

Review of GCB2020 (Friedlingstein et al. 2020)

Abstract line 65-66 — not clear why 6% is the median of [6,6,7,13] … add decimal point to numbers?

Lines 103-105 — glad to see this, but perhaps you could be a bit more clear that the term "global carbon budget" is explicitly different and should not be confused with "remaining carbon budget" and it variants. Could add a citation here to [Matthews et al (2020) Opportunities and challenges in using remaining carbon budgets to guide climate policy, Nature Geoscience, in press.] who also articulate this distinction.

Lines 339-341 — unclear to me how uncertainty in consumption-based emissions would not be larger than territorial, since you are adding additional uncertain components like trade flows and sectoral emissions factors.

Lines 1047-1052 — this sentence is extremely long, and somewhat grammatically problematic. Also, I don't see mention of how gross fluxes were calculated in the methods section.

Lines 1368-1373 — I have noticed a general trend the past few years of overestimating the projected growth rate in fossil fuel emissions (e.g. last year's projection of 0.6% has been downgraded to 0.1% ... the projection for 2018 in LeQuere2018 was 2.7% and subsequently revised to 2.1% in F19 … 2017 was projected at 2.0% then subsequently updated to 1.6% …). Not sure if this is significant or not, but the relative overestimate in the projected value has been pretty consistent for the past few years, so maybe speaks to some bias in the projection methodology that is worth considering.

Lines 1454-1455 — why report the median here (and in the abstract) rather than the mean? The relative independence of the different estimate is also not terribly clear — on first glance, the UEA and GCB methods seem quite similar (also given the overlap in authorship), whereas the Priestly Centre estimate is more independent in terms of methodology. Given the various results presented here (including also the IEA estimate), a mean decrease of 8% seems equally plausible as a median of 6%.

---

## Author Response (AR2)

Response to reviewers

Reviewer 1 (Han Dolman and Roxana Petrescu)

**General Comments:**
The authors are once again to be complimented on their work, the sheer breadth and number of data sources used is simple outstanding. As always, this will be a very useful resource for scientists (less so for policymakers, given the content, depth of treatment and length).

*Thank you*

We like the standardization of the inversions with a single a priori dataset for Fossil fuel emissions. This was much needed. The level of disagreement in the latitudinal between the inversions remains however large. The current version of the paper (as previously) however takes a rather lack luster attitude to this: unfortunately, not resolved. Given the emphasis on using inversions in the future using satellite data, it would not harm to include a few lines of thinking about this issue in the discussion.

*We agree that the latitudinal flux distribution remains an open issue year after year. However, we would like to stress that this is not simply a question resulting from the inverse modeling, but rather an actual large open question to the whole carbon cycle community: what is the balance of sources and sinks between the tropics and northern latitudes? The atmospheric data combined with inverse modeling seems to us the only true constraint, as DGVMs are not driven by observations, or process knowledge, that have any spatial correlation over such distances. Having noted this, the constraint from atmospheric observation remains weak mostly due to transport model differences, and sparsity of tropical observations (page 32, bottom). But new constraints from XCO2 products seem to also support a larger tropical land-flux to the atmosphere, and together with the expanded ensemble of systems available to us, this allowed us to this year make strong updates to the text about the latitudinal fluxes. We identify the two groups of inverse solutions on page 44, line 1272 and in Section 3.2.3.3 we expand on this dipole in a full paragraph. But we agree that some of the text following that remains a lackluster acknowledgement of known shortcomings in inverse models and a general call for more data.*

*So to accommodate the suggestion from the reviewer, we have rewritten the first paragraph of Section 3.2.3.3, now referring to the recently published assessment of the tropical-NH dipole in the paper of Gaubert et al., (2019), to which our community contributed. The main conclusion in that work is that we are seeing a convergence of modeling results towards a neutral tropical land flux, from a growing suite of models that compare well to new observational constraints from the HIAPER and HIPPO campaigns.*

*Modified text: "The expanded ensemble of atmospheric inversions (from N=3 to N=6) allows to have a more representative sample of model-model differences e.g. in latitudinal*

*transport and other inversion settings (Table A4). When assessed for their tropical/northern land+ocean fluxes we see a dipole arise, where three models estimate a Northern extratropical sink close to 2.5 GtC/year, and the other three a sink of close to 3.5 GtC/year. The inversions resulting in a large Northern sink estimate also a tropical source (0.5 PgC/yr), whereas the others retain a tropical sink (-0.5 PgC/yr). Such differences were described previously in the work of Schuh et al, (2019) comparing the TM5 and GEOS-Chem transport models, with their preferred tropical-vs-NH flux distribution tied to their vertical mixing characteristics, consistent with the analysis of Stephens et al (2007). Both groups of GCB models perform equally well on our evaluation metric of the misfit of optimized CO2 from inversions against independent aircraft data in Fig B3 though. In a recent update of the Stephens et al. publication, Gaubert et al., (2019) specifically assessed a suite of inverse models for their latitudinal flux distribution additionally using aircraft data from the HIPPO and HIAPER campaigns, as well as from the NOAA Global Greenhouse Gas Reference Network's aircraft program. They concluded that vertical transport model differences (including five of the six transport models used here) did not drive the northern hemisphere uptake across the suite, and that a convergence towards a more neutral tropical land flux (and by inference a sizable tropical vegetation sink) is evident. This convergence was illustrated by a much reduced standard deviation on the tropical-vs-NH land fluxes across models, which went from nearly 2.1 PgC/yr (1-σ) in the TransCom T3L2 project (N=12), to 1.21 PgC/yr in the RECCAP group of models (N=11) presented by Peylin et al (2013), to 0.41 PgC/yr in their most recent ensemble (N=10). This illustrates how new observations, also from the more recent ATOM campaigns, could potentially differentiate between the more and less realistic realisations of the land sink shown in Fig.8."*

That brings us to the discussion itself, which is very similar, almost the same as last year. It may be an idea to use the discussion to highlight the impact of changes made in this year's budget (the description of which is now hidden in the methods and results) to make it more relevant to this year's budget rather than sticking to the not so exiting repetition of last year general statements.

*Thank you for the suggestion. The discussion section has been updated, discussing each component of the budget separately.*

**Individual Comments:**
L 303. "developing countries like China". Better to change into something like countries with strongly developing economies like China.

*Done, thank you*

*L443. You could add in brackets also the names of the external datasets for peat burning..*
*For conciseness (this refers to multiple datasets, not just peat burning), we prefer the reference to Sec. 2.2.1, where the external datasets are referenced and explained.*

*L497. Only 2 GtC for peat fires over almost 60 years seems minute amount. We remember estimates of peat fires after el-Nino's of almost similar order of magnitude.*

*Peat fire emissions in equatorial Asia following the 1997 El Nino were 0.5 PgC; in all other years emissions were 0.2 PgC or less, see https://www.geo.vu.nl/~gwerf/GFED/GFED4/tables/GFED4.1s_C.txt. Given pre-1980 peat fire emissions are 0 (see two sentences before in the manuscript) this adds to 1.99 PgC. Since FAO/FRA is also underlying the LUH2 forcing (FRA wood harvesting, FAO agricultural areas used by HYDE) all land-use datasets are dependent and there are no alternative datasets available for the global coverage and long time period needed. However, the 3 models differ very much in structure, which is why we average the three models rather than group by land-use input. A full assessment of dependencies requires exchanging the input data in alll model, which is model-structurally not possible for all and beyond the scope of this study.*

L905-908. These lines could be deleted. It just tells us what you do not do. Maybe better to integrate with the next bit to keep the focus

*We agree and have merged the sentences. This passage now reads: 'A new feature in this edition of the global carbon budget is the use of a consistent prior emissions dataset for EFOS across almost all inversion models, which avoids the need to correct the estimated land sink (by up to 0.5 GtC in the Northern extratropics) for most models. Only the UoE inversion used an alternative dataset and required a post-processing correction (see Table A4)'*

L1007.You go to some length estimating the loss of additional sink capacity, but you do not use it in the budget (L1203). Can you explain?

*We clarified this paragraph: the LASC is not a missing process that should be included in the budget, but a flux that explains differences between DGVM- and bookkeeping-based ELUC estimates, where the first are only used for the spread, not in absolute terms.*

L 1047 land use change fluxes

*Done, thank you*

line 1052 Should be rewritten a bit to make it more understandable.Suggestion: "Emissions from newly added gross sources are on average 2-3 times larger than previous net emissions (X Tg C in GCB 2018), and increased from an average of 3.5 ± 1.2 GtC yr-1 for the decade of the 1960s to an average of 4.4 ± 1.6 GtC yr-1 during 2010-2019..."?

*Thanks for pointing out that our sentence could be misunderstood. It is \*not\* about newly added sources, but about the newly added of net fluxes into gross fluxes. We have rewritten the sentence.*

L1196 H&N to add2017

*Done, thank you*

L1597. True, but the tropical areas are probably even more undersampled.

*This is correct but we highlighted the Northern hemisphere here as it is the region where significant long standing discrepancies between bottom up and top-down estimates remain.*

L1541. This is not new. Under sampling of the Southern Ocean has been an issue for many years. Can you rephrase to make it sound not as you discover something novel?

*Agreed, but what is new is that these uncertainties are now better quantified, hence reference to Watson et al. 2020. We reformulated the sentence to: The **long-standing sparse data coverage of pCO2 observations in the Southern compared to the Northern Hemisphere (e.g. Takahashi et al., 2009) continues to exist** (Bakker et al., 2016), **and to lead to** substantially higher uncertainty in the SOCEAN estimate for the Southern Hemisphere (Watson et al., 2020).*

Table 5 and 6 we would add an extra column with numbers for the projections 2020, then next year you can compare the 2020 budget to this years'projected one

*We would prefer not doing this. 2020 estimates are projected for fossil fuels emissions and, with lower confidence for land use changes emissions. We do not have any model based projections for the ocean and land sinks. As explained in the text, for the ocean, we assume the the ocean sink in 2020 would not be dramatically different from 2019. For land, we simply assume that it has to close the budget, given the ocean sink. We don't feel these estimates are robust enough to be highlighted in these tables.*

Figure 6 caption we think is wrong, explanation for a,b,c doesn't match the sub-plots and d is missing(c should be d=c-a, b is not explained)

*Thanks for highlighting this oversight. We have resolved the issue by accurately explaining the content of each panel.*

Figure 9: if you show cumulative changes for 1850-2018 why not show also changes for last decade instead of mean flux?

*Figure 9 can be seen as a synthesis figure that summarises the paper, showing the cumulative emissions since 1850 and the current emissions. What the reviewer proposes, showing the cumulative emissions for the last decade would essentially means multiplying the numbers in panel b by 10 (going from annual data in GtC/yr, to decadal sum in GtC). We are not sure that would be easier to understand for the reader.*

**Reviewer 2**

**General Comments:**

"Overall: an amazing urgent essential compilation presented with an excellent comprehensive manuscript. 'Chapeau' to authors. Excellent product for ESSD as well. Please regard comments and suggestions that follow as small improvements to overall readability."

Thank you

Individual Comments:
Line 85: "they only became the dominant source of anthropogenic emissions"

*Done, thank you*

Line 114: "(including coasts and territorial seas) ??

*Done, thank you*

Lines 116 to 119: "The global emissions and their partitioning among the atmosphere, ocean and land are in reality in balance, however due to imperfect spatial and/or temporal data coverage, errors in each estimate, and smaller terms not included in our budget estimate (discussed in Section 2.7), their sum does not necessarily add up to zero." Revise to: Global emissions and their partitioning among the atmosphere, ocean and land are in reality in balance. Due to some combination of imperfect spatial and/or temporal data coverage, errors in each estimate, and smaller terms not included in our budget estimate (discussed in Section 2.7), their sum does not necessarily add up to zero."

*Done, thank you*

Line 129: I do not know prevailing lingo, but "territorial" and "consumption-based" are not comparable. "Consumption-based estimates still rely on territorial boundaries? Better to write "production-based vs consumption-based"? Throughout the text you refer to countries or nations but rarely to territories? Or you refer to "national territories"? Reader does not encounter careful definition of "territorial emission inventories" until line 315. Some clean up and consistency needed here?

*There are two standard methods for emissions accounting. Territorial is used for inventories reported to the UNFCCC, while an economic boundary is used in emissions accounts. In the consumption emissions literature, the latter is called a production-based emissions account, contrasting with a (non-standard) consumption-based emissions account. In this study we present emissions on a territorial basis, consistent with UNFCCC reporting, as well as consumption-based emissions estimates. The territorial emissions estimates we present do not follow the boundary definition of production-based emissions accounting, which includes national economic activities occurring overseas and excludes activities of non-nationals within the territories.*

Line 167 'most recent' rather than "last"?

*Done, thank you*

Line 201: "estimates of EFOS rely primarily" should be EFOS?

*Done, thank you*

Line 278 to 291, cement carbonation: good up-to-date discussion but Guo et al. is not cited in references?

*The reference has now been added.*

Line 381: "in place for six weeks before they ease" By the time of publication we ay know that easing has not worked and that lockdowns - based on erratic or absent national policies - have resumed in many locations. Economic impacts uncertain everywhere; one does not want to see these authors or this product 'chase' political changes. Better to state impacts so far as known and documented but to stay away from Covid-19 projections?

*This text is the description of the Forster et al method, not an assumption we make. In any case, the text is updated as the dataset was updated. More broadly: Each year the GCB makes a projection. This year, because of COVID, we have been extra cautious and therefore use multiple estimates, and we believe, have an appropriate amount of caution. In any case, by the time of publication, we have a few more datapoints, so the projection component becomes less important.*

Line 400, 401: a separate crowd-sourced tracking of aviation data currently in ESSD discussion (https://doi.org/10.5194/essd-2020-223, who knows how it got in ESSD and one scarcely knows if or how to credit it) shows a much steeper decline in aviation travel than mentioned here. Again, although they want to keep current and alert, we really do not want GCB authors trying to keep up with rapidly-moving hardly-certified external activity indicators? Better that they declare uncertainty - as they do elsewhere - rather than publish today what will change tomorrow? Alert but a bit more cautious; authors will know best approach.

*As mentioned above, the issue of projection becomes less important by the time of publication. But key point of review noted: be cautious with the projection.*

Line 406: Do we need a reference to EDGAR here? (As you do later at line 900.) Or we assume readers find it in Liu et al. online? This entire section gets a bit speculative, a departure from past reliability of GCB?

*We are describing the Liu et al. (2020) method here, which made use of EDGAR. The reference to EDGAR would be in the Liu et al paper.*

Line 426 and several times following: If "Carbon Monitor" represents one of your reliable documented sources for 2020 emissions (e.g. appears frequently in text as well as in Table A8) we need a standardized reference?

*The reference is given before: Liu et al 2020, but we added here for clarity.*

Line 480 to 482 - meanwhile, FAO data and definitions also undergo update and - to a smaller degree - revision (e.g https://doi.org/10.5194/essd-2020-203); perhaps not yet valid for this edition of GCB but something to take account of in future versions?

*An updated version of the Houghton&Nassikas (2017) dataset is being produced using the updated FAO/FRA estimates; however, this dataset is not yet published and we therefore use H&N2017 based on the older FAO/FRA. The next year's budget paper is expected to include this revised H&N estimate.*

Lines 493-494 - emissions from drained soils discussed in https://doi.org/10.5194/essd-2020-202?

*Yes, thank you. We replaced the webportal by the article reference.*

Line 505: LUH2 never defined? I think you mean Hurtt et al. 2020 but not clear how one would access that reference? The landing page for the PCMDI DOI only lists a 2017 version plus the option for upates. No valid reviewed reference to LUH2?

*LUH stands for harmonized land-use change (which the sentence states). LUH2-GCB2020 is an unpublished update of the Hurtt et al 2020 dataset, which is the one published on PCMDI. We have rephrased the sentence to be clearer.*

Line 517: Likewise, FAO / FRA much used but never defined nor properly referenced

*We have added the reference to Houghton and Nassikas (2017). Further above, where we had introduced h&N2017, we had referenced FAO/FRA appropriately.*

Line 528: "anomalous fire season in Southeast Asia." Also (both 2019 and 2020) in Siberia, North America, Amazonia, etc. What seemed anomalous in the past now proves regular and expected albeit still unquantified? For both accounting through 2019 and projection for 2020, the fire term grows increasingly unknown and uncertain? How does growing uncertainty in fire emissions contribute to overall ELUC uncertainty?

*We extended on the special situation in Indonesia in section 2.2.4 and 3.3.1 and reference these sections here now. We also discuss and quantify Amazon fire emissions in more detail in 2.2.4 and 3.3.1. Fires in Siberia, Australia, North America were not largely attributable to land use activity, different from the Amazon and Indonesian fires, and are thus not part of ELUC.*

Line 576: no definition of CRU nor of JRA although you use those acronyms frequently throughout this section.

*Done, thank you*

Line 621, 622: "scale almost linearly with GFED over large areas (van der Werf et al., 2017)" out of date or no longer valid?

*For the annual updates of GFED we cannot rely on emissions based on burned area data given the long latency and change in versions since the 2017 publications. Active fire detections and its FRP are available in near-real time and for the annual updates we have scaled GFED estimates with these proxies using the overlapping 2003-2016 period. For large regions and annual numbers the correlation coefficients near unity, except for the Boreal region which is not relevant for this assessment. This approach is also used in the Global Fire Assimilation System (GFAS, Kaiser et al., 2013) and a paper on this approach will be part of the next GFED version which will be published in 2021.*

Line 632: authors have no doubt done expert assessment, but "pantropical fire emissions" in 2020 only two-thirds of 2019 seems counter to most reports? Give your readers some basis for confidence in this statement?

*We have added a discussion of uncertainties and added a confidence statement. We also clarified that our estimates relate to the 2020 calendar year -- indeed, 2020 deforestation rates are reported throughout reports and media to be higher in the Amazon than in 2019, but these estimates relate to the PRODES calendar year, which goes from Aug of the previous year to July of the current year; in Gregorian-calendar-year terms, deforestation went down.*

Line 723: SOCAT, should have been defined a few lines earlier?

*Done, thank you*

Line 867: You have not explained xCO2. For a broad range of readers, you will need to carefully explain all terms.

*Done, thank you*

Line 880: Something weird here?

*Done, subsection removed*

Line 1258 - New / updated information on SO sinks emerging recently and continually? Hard to keep a reliable annual budget going against the 'noise' of new ocean data, but possible changes in SO estimates might need a mention here? Or, wait until next version?

*We assume that the reviewer points to the SOCCOM floats as new/updated information on SO sink information. This is a great and much needed project. It estimates pCO2 from pH sensor measurements and estimated alkalinity, and hence pCO2 is associated with a larger uncertainty than from shipboard measurements. We have discussed on whether to use this data as input to the data-products but have decided against it due to the larger uncertainty. The ocean community needs to resolve how these data can be combined with other measurement types (e.g. cross-over comparisons with ships, moorings etc) to reduce its uncertainty. The Bushinsky et al paper (resulting from SOCCOM) is cited here as possible explanation, but it's not the right place to comment on new approaches here where we compare the estimates that we DO use.*

Line 1326 - Here a reader encounters "column CO2 products" rather than (as earlier) xCO2. Some clarification needed?

*Done, clarified earlier now.*

Lines 1338 to 1340, discussion of uncertainties in NH land sink: substantial redundancy here?

*Thank you, we merged the two sentences to avoid redundancy*

Line 1473, Section 3.4.2: Numeric errors here, corrected already by authors according to text supplied by editor? Someone needs to read the final (proof) version carefully to confirm final numbers.

*Done, thank you*

Line 1521: "suggests we do not yet have a complete understanding of the underlying carbon cycle processes." The manuscript carries a necessary overall uncertainty: how much uncertainty of each component arises from reporting deficiencies and how much from missing processes? Here the authors seem to point to missing processes but much reporting earlier in the manuscript focussed on reporting (or, modeling) uncertainties. No hard line between weak reports and missing processes, but can authors give a clearer sense of where the problem lies? This sentence confuses rather than clarifies? Discussion that follows in this paragraph perpetuates this duality: some improvements might come from "improving the underlying data and statistics", from "scrutiny of carbon variability in light of other Earth system data", and from "higher resolution and process knowledge at the regional level". If these expert authors truly do not know the best route toward improvement (reducing BIM), then say so explicitly. In which case the introductory sentence at line 1521, with its apparent focus on processes rather than underlying data, remains misleading at best? This reader very much appreciates subsequent discussion of uncertainties in southern ocean, in NH LUC, etc, as well as good estimates of how long (decade or decades) one would need to detect a change in a statistically-robust manner. Lines 1581 and 1582 belong in the abstract as well, to alert readers who only browse?

*Reading that section again, we don't find it "seems point to missing processes". We are giving three ways to reduce unexplained variability, none is advocating for "including missing processes". May be the reviewer got confused by the terms "carbon cycle processes" in the first sentence. We now changed to "carbon cycle dynamics" to avoid giving the impression this is about (missing) processes.*

**Reviewer 3**

**General Comments:**

It is my pleasure to have an opportunity to review this manuscript. I do my review for ESSD manuscripts following the guideline summarized by Carlson and Oda (2018) ESSD. The 2018 guideline should tell you where I/my comments came from. Given the nature of ESSD, most of my comments go to the method section pointing out the information that I feel missing. Below is a summary of my comments. You will also see a list of my line-by-line comments. Hope my review is useful.

1 Data description. While I imagine the amount of effort dedicated to compile this budget/manuscript, I feel the level of the information is not meeting the ESSD's standard. First, please improve the data/model tables. Tables should include name, version, data source (where we can get the data, w DOI), data citation (this is different than data source), etc of the items (e.g. data and models) used in this manuscript. For the data that are not yet published, the authors should provide enough information to allow readers to understand the data. This is the basis requirement. ESSD will be hosting this manuscript/paper not for carbon cycle scientists, but also for people in the broader Earth science community. Perhaps also for someone who randomly pick up this manuscript w/o any strong Earth Science background. ESSD papers need to be useful for such readers and the described data needs to remain available (see more in the2018 guideline).

*All data presented in the paper are freely available and provided with the paper in the form of two spreadsheets with a DOI  https://doi.org/10.18160/gcp-2020. The first spreadsheet gives the annual estimates of fossil fuel emissions, land use emissions, atmospheric CO2 increase, ocean carbon sink and land carbon sink as presented in this paper. Fossil fuel is further broken down in fuel types. Individual models estimates for land use changes emissions, ocean sink and land sinks are futher provided in the spreadsheet. The second spreadsheet gives fossil fuel estimates broken down by countries.  As summarised in Section 6: The data presented here are made available in the belief that their wide dissemination will lead to greater understanding and new scientific insights of how the carbon cycle works, how humans are altering it, and how we can mitigate the resulting human-driven climate change. The free availability of these data does not constitute permission for publication of the data. For research projects, if the data are essential to the work, or if an important result or conclusion depends on the data, co-authorship may need to be considered for the relevant data providers. Full contact details and information on how to cite the data shown here are given at the top of each page in the accompanying database and summarised in Table 2.*

2 Summary/Rationale/Justification of the approaches. This is partially done in some sections, but we would like to see the summary of what are covered and what are not. I see the authors did a wonderful job to collect the all of the results and synthetize them. However, probably the authors would agree with me that there should be something not included in this manuscript. I use as atmospheric inversions. It is totally fair to use a subset of inversion models available in the community. I would like to suggest the authors to just

list model inclusion criteria and expand discussions including what are not covered in this manuscript. For example, some highly relevant papers such as Crowell et al(2019) (OCO-2 inversion comparison) and Schuh et al. (2019) must be worth discussing in this manuscript. I thought the authors did a great job in the DVGM section. It was concise and short (given the amount of underlying info). Yet it has the basic info that I wanted to see. For some sections, I also thought it would be great to have a short review of what types of the approaches are taken to examine a flux component and tell us if you have them all in this report or not. That would help us reader to recognize the position of this synthesis report in the big picture of the carbon cycle science.

*As suggested, for the models used, we have now include a sentence explaining the criteria for including models, in a similar way as what was already done for the DGVMs. As for a "short review of the approaches", each component has its approach desccribed in the methodology section.*

3 Uncertainty analysis. The authors did great job in listing the potential sources of uncertainties. I would like to suggest the authors to discuss the sources in more quantitative way. I can imagine some of the uncertainty sources are tough to assess. However, at least, the authors should give a try to convince us that the uncertainties discussed should be small enough and the results remain robust.

*All uncertainty are described and quantified in sections 2.1 to 2.5*

4 Summary of advance in our understanding of the global carbon cycle. The authors kind of did this, but I thought it would be great to have a summary section (or as apart of the abstract/conclusion) of the advance in our understanding of the global carbon cycle (excluding the methodological differences) beyond the budget breakdown (I understand this is the main thing, though). For example, the authors mentioned the confidence levels for the budget components. Are the levels are changing? Since probably the authors are placed in the best position to summarize the current understanding of the carbon cycle science, I would like the authors to highlight a bit more of the state-of-the-art science results collected. I believe it would be a benefit for the general scientist readers. Also, it would make this ESSD manuscript accessible to the random people who just googled "global carbon budget" and found this manuscript**.**

*We do attempt to summarise our understanding in the discussion section  of the manuscript. We are bnot sure what the reviewer is asking by "having a summary section of the advacnce in understanding ... beyond the budget breadown" ?  Also we are aware that this paper is an ESSD publication that describes, present and analyses dataset of the global carbon  cycle. A review the state-of-the-art of all carbon sciences is outside the scope of this publication.*

**Individual comments**

P5, L5 (L45): Describe -> Describe and synthesize?

*Done thank you*

P5, L6 (L46): five major components, such as fossil fuel emissions (FOS), Land-use Change (LUC),G_atm, S_ocean and S_land, This is pretty obvious for us carbon cycle scientist, but maybe not for the ones who just randomly picked up this manuscript w/o any carbon cycle background.

*The five components are described in the following sentence.*

P5, L24 (L64): COVID-19, spell out.

*We assume COVID-19 does not need to be spelled out (i.e. Corona Virus Disease 2019) in the manuscript. We defer to the editor/publisher to decide if this is needed.*

P6, L5: and gas flares?
`

*If this is L108, then emissions from gas flaring is already represented in the list as "fossil fuel combustion and oxidation from all energy and industrial processes", so there is no need to list gas flaring separately, since flaring is the combustion of natural gas.*

P7, L20 (L123): introduce ppm

*ppm is introduced on page 5: parts per million (ppm)*

P8, L1 (L134): any other references? Maybe RECCAP reports/papers?

*RECCAP does regional assessments, it does not assess the global CO2 budget, we clarified the sentence.*

P8, L1 (L134): The IPCC methodology, which IPCC methodology do you refer to?

*Sentence has been rewritten, with reference to the "IPCC methodology" removed.*

P8, L21 (L154): Citation for AR5

Done, citation added

P8, L14 (L147): our estimates, which are mostly annual basis? (w some exceptions)

*Estimates are always annual or cumulated over time (Table 8).*

P9, L1 (L164): detailed description of the data sets, the data table does not seem to meet ESSD's standard. For example, data description, data source(s) and data citation(s) are only insufficiently provided.

*See response to first general comment*

P9, P7 (L170): citation for AR6

*Reference added*

P9, P17 (L180): I believe this should be the best synthesis effort. However, it is still true that this was done by a subset of scientists in the research community. I feel this manuscript should mention the criteria for data and model collection.

*This is now also done in section 2 for the ocean and land datasets/models used in the paper. For ELUC, we are not aware of any other global annual estimate.*

P9, L20 (L183): ESSD requests to provide data sources in order to ESSD data users/ to examine original data. In principle (ideally), the data sets used in an ESSD paper needs to be available with a DOI (see Carlson and Oda, 2018). Many of the model output do not seem to be accessible for ESSD data users.

*As explained in the first general comment, the data presented in the paper are availble with a DOI: https://doi.org/10.18160/gcp-2020*

P10, L7 (L201): The estimates of EFOS -> The estimates of E_FOS in this study, there are variants ofE_FOS emission calculations. I assume this sentence refers to the reference approach (if wedefine it using the IPCC terminology)

*Added "in this study". This sentence does not refer to any specific approach. The estimates for fossil CO2 are mostly from fossil fuels, and therefore the bottom-up estimates presented here are derived (in various ways) from energy data.*

P10, L10: peat fuel?

*CDIAC's "solid fuels" category includes peat used as fuel. In the GCB we use the term "coal" instead of "solid fuels" because it is more readily understandable to a wider audience, so here we are explicit that peat used as fuel is also included, since this question has been raised in the past.*

P10, L21: CDIAC, spell out

*This is now spelt out on first use.*

P10, L13: UNFCCC (spell out) national inventor reports, did you take estimate based on the reference approach (or sectoral approach)?

*Changed "UNFCCC inventory reports" to "national greenhouse gas inventory reports submitted to the United Nations Framework Convention on Climate Change (UNFCCC) ". Estimates are from the sectoral approach. The reference approach is intended only as a simple check of the sectoral approach.*

P14, L14: Most accurate -> the best estimates? Inventories can't assure their accuracy by themselves. Also, UNFCCC inventories from countries could be challenging to use together as they are not compiled in a globally systematic way like EDGAR does. Please not following the IPCC guideline does not assure the accuracy.

*Since EDGAR follows IPCC guidelines, and only Tier 1 level, it is difficult to conclude that EDGAR might in any way be superior to the NGHGIs submitted to the UNFCCC. For almost all fossil CO2 emissions, EDGAR applies the same simple method for all countries, while Annex-1 NGHGIs generally use more sophisticated methods with significantly more country-level knowledge and checking. NGHGIs are reported in a highly consistent format (CRF) with clearly delineated system boundaries, making them ideal for combined use. Our assessment is that of all country-level estimates available for Annex-1 countries, the NGHGIs have the most expert knowledge (orders of magnitude more time investment than any other) and access to detailed underlying data that is not available to any other method. We therefore assess these as the most reliable.*

P10, L17 (L211): BP, yes, but the BP stats does not offer data transparency and traceability that wewould like to see in ESSD papers.

*One could debate the relative transparency of individual datasets, none are perfect. In the case of BP, it is the only dataset available for the time period it is used (2018 and 2019), has been found to be robust, and is widely used. Not using BP would mean dropping two additional years of the budget and not performing a projection.*

P10, L24: UN -> UNSD?

*Changed as suggested.*

P10, L27: Do those values remain the same as defined in Marland and Rotty (1984)? How much differences from other reference approach estimates (UNFCCC if used) should we expect to see due the values specifically used in the CDIAC estimates?

*CDIAC's methods and the parameters used have changed since the 1984 publication. The changes are summarised in Andrew 2020a, based on interviews with Gregg Marland and Dennis Gilfillan. We have added a reference to Andrew 2020a at the end of this sentence. Andrew 2020a compares datasets, but not specifically CDIAC with the UNFCCC reference approach or specific parameter choices (however, in line with Andrew 2020a energy use differs as do parameters). It is out of scope to do that analysis here.*

P10, L30: WWII, years?

*Changed "two years" to "1942-43".*

P11, L3 (L227): It sounds like these are ones calculated by the sectoral approach. How exactly did the authors reallocate IPCC-defined sectoral emissions to fuel-based emissions?

*We reallocate the detailed estimates from the national greenhouse gas inventories of Annex 1*

P11, L8: CDIAC, is this originally due to the data collection by UNSD (or submission by India)? I believe India is not the only country that reports data using their own fiscal year period. Why India?

*Yes, this is because of India's submission of data to UNSD. Changed "We therefore" to "Given that India is the world's third-largest emitter and that a new data source is available that resolves these issues, we".*

P11, L20: which values exactly did you use for projecting 2017 emissions? There are several values you could use to do the projection.

*We used energy consumption in energy terms, EJ. Added "(in EJ)" to the sentence.*

P12, P10 (L264): how about oversea territories for other counties, such as UK? How did you spatially allocate those emissions?

*This is a tiny share of emissions, that would require substantial text to explain. The reviewer should take contact for more information.*

P12, P18 (L272): Could you provide information more quantitatively? How large the impact would be?

*The next line states the magnitude: "The discrepancy has grown over time from around zero in 1990 to over 500 MtCO2 in recent years, consistent with the growth in non-oxidised carbon (IEA, 2019). "*

P12, L21: Why can you mitigate the issue by adding international bunker emissions? Even adding the international bunker emissions to the sum of country totals, you would still have~3% or so emission discrepancy (e.g. Oda et al. 2018 ESSD).

*It is not the addition of bunker fuels that mitigates this issue, it is the use of sum of country totals. Bunker fuels must be added to this to obtain a global total.*

P12, L24 (L279): Can you provide more quantitative information (e.g. sink size, sink rate, lifetime..) in order to let us know how significant this could be in the global carbon budget?

*This is the methods section, so we don't tend to have numbers here. The requested information is found in section 3.1.1.*

P13, L7 (L293): How did you assess the uncertainty? Is the method independent from Andres et al.(2014)? Note the uncertainty range for the global total (6-10%, 90% confidence interval)mentioned Andres et al. (2012) is based on the assessment by Marland and Rotty (1984). So you would not bump up the uncertainty for the 2018 and 2019 while those were projected. Any comments?

*The uncertainty is based on the literature cited, and we take a more conservative approach. We do not make the uncertainty the same as Andres 2014 to ensure that we capture other structural issues, and so on. For 2018 and 2019 we make no change. The 2020 uncertainty is larger, but only marginally, and this demonstrates why no change is made in 2018 and 2019. The 2018 and 2019 values are taken from growth rates, thus the uncertainty is anchored in the 2017 value (relatively the uncertainty in growth rates is smaller than absolute uncertainties). Likewise, we do not consider uncertainty changing over time (eg, uncertainty in 2010 is the same as 1910, and so on. The uncertainty is a holistic perspective, not a uncertainty for a specific year.*

P13, L18 (L303): Also see Andres et al. (2014) for regional uncertainty estimates

*We have now added this reference, which was already referred to elsewhere in the article and should also have been here.*

P13, L20 (L306): medium confidence, what does this mean?

*This is introduced earlier in the article: "We also use a qualitative assessment of confidence level to characterise the annual estimates from each term based on the type, amount, quality and consistency of the evidence as defined by the IPCC (Stocker et al., 2013)."*

P13 (L310), Did Ciais et al. (2013) confirm consistency between E_fos and direct observations? This seems to sound stretch.

*We meant: as assessed in IPCC, ie Ciais et al (2013). This has been clarified now.*

P14, L3: Are those export and import terms consistent with ones used in the E_fos calculation?

*No. Exported and imported emissions on a consumption basis are calculated using calculated embodied carbon in traded goods.*

P14, L21: Does GTAP produce fuel-based estimates rather than sectoral-based estimates?

*GTAP publishes estimates by both fuel and sector.*

P14, L14: Is it fair if we say the uncertainty assessment might not fully capture a systematic part of the uncertainty as well as the model structural errors?

*The available literature suggests that various new errors in the calculation of consumption-based estimates tend to cancel. This literature is summarised in Peters et al 2012, the cited paper.*

P15, L15 (L361): three separate <global> studies?

*Ignored. "global studies" doesn't sound right. But we changed the previous sentence to clarify we are considering studies of 2020 global CO2 emissions here.*

P15, L19 (L365): Provide some mode details of projection models in order to allow readers to assess the results.

*Additional information on the GCB method is provided in the Supplementary Information (Appendix C). More details about the other three methods are give in their respective publications.*

P15, P22 (L368): proxy data, such as?

*Done, thank you. Examples of proxies are now given.*

P15, P24 (L370): What exactly? This is important as you are comparing different projections in the single figure.

*Not sure what this comment referred to, sorry.*

P15, L25 (L371): absolute <daily> emission changes?

*Change made as suggested*

P16, L24 (L401): calibrated to the traffic data in Paris?

*Yes this is correct. We clarified now.*

P17, L26 (L433): How do you assess the uncertainties associated with individual projection models? We do not see the uncertainty estimates and can't tell how we should interpret the agreement and disagreement among the estimates.

*Only one study provided uncertainty estimates (Le Quere et al). In this version, we refer to the study estimates and also include the uncertainty from Le Quere et al.*

P18, P2 (L438): Are these consistent with the UNFCCC inventories? At least, some compatibility with the UNFCCC inventories? How can we assess these estimates in relation to the UNFCCC reported values?

*They are not, because the UNFCCC definition differs from the scientific one. We have added a discussion including two recent references that discuss this in detail.*

P19, L4 (L471): Short model/data description and model output data source (DOI) seem to be missing in this manuscript.

*The models are described in Table 4 and Table A1. Global average annual time series of models output are available in the GCB spreadsheet (https://doi.org/10.18160/gcp-2020)*

P23, L1 (L593): Inclusion criteria for other components (data and models) should be listed, too.

*This is now also done in section 2 for the ocean and land datasets/models used in the paper. For ELUC, we are not aware of any other global annual estimate.*

P23, L25 (L617): Given the challenges mentioned, how would you assess the uncertainty associated with projected ELUC?

*We have added to Sec. 2.2.4 a confidence statement and discussed important uncertainties.*
P25, L23 (L676): Ok, so this is relative confidence levels used in this study. Then, what would be the confidence level for E_Fos?

*This is given on line 306 of the MS (section 2.1.2): medium confidence.*

P26, L14: Brief text regarding the inclusion criteria for the GOBMs? We want to hear the justification for the use of the model ensemble for this carbon budet assessment.

*As explained in section 2.4, "the estimates of the global ocean CO2 sink SOCEAN are from an ensemble of global ocean biogeochemistry models (GOBMs, Table A2) that meet observational constraints over the 1990s", with section 2.4.1 giving more details on tehse observational constraints. In addition, we now addded the following text: The GOBMs constrain the air-sea CO2 flux by the transport of carbon into the ocean interior, which is also the controlling factor of ocean carbon uptake in the real world. They cover the full globe and all seasons and were recently evaluated against surface ocean pCO2 observations, suggesting they are suitable to estimate the annual ocean carbon sink (Hauck et al., 2020).*

P29, L4 (L779): Is 101% suggesting an ocean area definition difference among models?

*Indeed, each ocean model has it's own spatial grid definition and resolution, hence not all having the same ocean surface.*

P32, L2 (L867): Introduce xCO2

*Done*

P32, L8 (L873): Inclusion criteria (like you did for the DGMV section)?

*Done for other components as well now. Thank you for the suggestion.*

P32, L23 (L888): While focusing on large scale analysis, it is surprising that the authors did not discuss the use of satellite data using a detailed report by Crowell et al. (2018) ACP. I'd expect more discussion regarding the use of satellite data here (e.g. in-situ vs. satellites, GOSAT/OCO-2,land/ocean glint...).  Probably the authors would agree with me that this subset of inversion model calculations might have failed to capture something.  Also, I feel Table A4 is not providing enough details while we all agree that the results do have sensitivity to the settings.P32, L28: transport, Schuh et al. (2019) GBC

*We thank the reviewer for this suggestion, but the text on page 32 L888 does not focus on large-scale analysis. We simply state here that OCO-2 and GoSAT-based results were created with CAMS and with the UoE systems, and the reference to Palmer et al documents that second effort. The report by Crowell et al about OCO-2 inversions is interesting and nice, but considering that we basically did not use or discuss XCO2-based inversions in this paper, we do not see a direct need to repeat details about land/ocean glint data, or results obtained with such data on the 2015/2016 carbon cycle.*
*As for Table A4, we are not sure how to modify the current version without further guidance, as the remark that it provides too little details is not specific enough.*
*Thank you for the suggested reference to Schuh et al., (2019), we have included it in the revised version*

P33, L3 (L898): Need more details since this is not publicly available yet. Grid resolution (0.1deg?)?,temporal resolution (monthly? hourly?)? Data source?

*We thank the reviewer for the request. We have now added a link to the data repository, which includes the requested details, as well as a report with full documentation*

P33, L10 (L905): not GCB, but Crowell et al. (2018) used the same prior info.

*Indeed, that is why we explicitly identified two important studies that did \*not\* do this.*

P34, 2.7 Would you be able to provide more quantitative assessment for these missing components? Convince us that the GCB assessment is still robust regardless of those missing components.

*We already provided a quantitative estimate for CO and CH4 oxydation (section 2.7.1), land-ocean C flux (2.7.3) and loss of sink capacity (2.7.4).   We now also provided an estimate of CO2 emissions from fossil carbonates (2.7.2).*

P34, L25 (L951): See Nassar et al. (2010) GMD and Wang et al. (2010) ERL

*Ignored, sorry, the comment was unclear.*

P35, L7 (L962): What about gas flare emissions? Not as Ch4, but as CO2?

*The reviewer's question is whether flared natural gas, which is in our estimates, makes this statement incorrect. It does not, since the statement is that emission (i.e. release to atmosphere) of fossil CH4 is not included, not that emissions from combusted methane are not included. But in fact CDIAC data include vented methane from oil and gas extraction as if it were immediately oxidised, so our statement is indeed incorrect, but not for the reason the reviewer gives.*

P37, 3 (L1018): Would you be able to summarize new findings/revisions/changes from what previous publication reported (improvements of the GCB product/report/summary)?

*Changes in products/estimates relative to previous GCB release are summarised in Table 3.*

P46, P12 (L1297): Are Crowell et al. (2018) and Schuh et al. (2019) irrelevant here??

*The study by Schuh et al indeed is a very valuable addition to this discussion. We have modified the text to also address a comment from reviewer #1, including this paper. Thank you*

P50, P10 (L1411): This is a very interesting statement. For E_eos, the use of fuel consumed is a very straightforward way to get $CO_2$ emitted (fuel burned). It still does not assure the accuracy by itself. However, from methodological perspective, it seems to be fair to give more trust to monthly fuel stats than proxy data-based approaches given many assumptions made. In fact, emission seasonality is often constructed using fuel stats (e.g. Andres et al. 2012 Tellus).Also, as the name suggests, proxy data are "proxy" for $CO_2$ emissions. Proxy data would be fuel consumption while we are not entirely sure how the proxy data get collected and how the indices are developed. We do have a history of the use of monthly stats for estimating monthly emissions, but we've not even examined the single use of proxy data for estimating emissions.

*We take this as a comment, but the text was modified to make these points clearer. Note, that we can't assume monthly data is perfect (it is estimated and sometimes not even revised to be consistent with yearly data). It is simply too early to say how the proxy-based methods have performed, until we get full year and reconcilled data.*

P50, L17 (L1418): Here is a preprint by Zeng et al. that shows some model simulations using the Carbon Monitor emissions.

*Noted*

P50, L1 (L1402): I thought it would be a good idea to start with the same emission value at the beginning of 2020. YTD emissions include the differences at the beginning of 2020. The projected emissions further include differences from the YTD emissions. At least, the authors could remove one of the sources of the differences to make the interpretation of the results a little bit easier (while it is not perfect by no means). I believe the authors are not very confident with the accuracy of the absolute emission values, but the relative changes.

*We are essentially assessing the results of different studies, though each study updated there estimates for the budget. So we are a little limited in what we can do to harmonise, as each study took a slightly different philosophy. We have harmonised where it made sense (eg, UEA is harmonised to the GCB monthly data)*

P51, L20 (L1450): But you could assign higher confidence to Chinese YTD emissions from the carbon monitor given the amount of the data collected/used?

*More data does not have to mean better estimate. We have to wait until we have reliable full year data before assuming one approach is better than the other.*

P53, L19 (L1509): And also future perspectives?
*This is exactly what we mean by "remaining cumulative CO2 emissions consistent with an ambition to stay below a given temperature limit".*

P56, L18 (L1600): First, the authors should provide names, versions, data sources (where we can get them), and data citation (w DOI) of the items used in this manuscript as a list. This is the basic requirement for ESSD papers. I do see some tables, but those are missing key information mentioned above. Note data source and data citation are not the same thing. Many of the citations listed are data/model citations. Second, data policy. I am not sure this is in line with ESSD's philosophy. Why not? Please refer to Carlson and Oda (2018).

*As explained above, all data presented in this paper are available via*
*https://doi.org/10.18160/gcp-2020*

References: Please check the links if they are still active. Especially ones w/o a DOI.

*Done, thank you.*

Table 2 & 4 : Data sources are missing.

*As explained above, all data presented in this paper are available via*
*https://doi.org/10.18160/gcp-2020*

Table 9: In addition to Korsbakken et al. (2016), Guan et al. (2012) Nature CC, gita-ton gap. That has demonstrated potential systematic errors in emission estimates due to poor energy stats data.
*Done, thank you.*

Appendix A. These tables need to be improved. Can't read some of the contents at all.

*Thanks, this will be done with the publisher*

Appendix B. Maybe better to change the X axis range to 0-15 (and 0-0.3 for Y axis) in order tos how the points better? Points are heavily overlapping.

*Done, thank you.*

Figure B3. How can we interpret this evaluation?
*This is explained in sections 3.1.3 and 3.2.3.3*

Figure B5. As mentioned earlier, maybe it would be better to start from the same Jan emission. This manuscript does not provide enough text to explain why those lines started from different estimates. Readers need to go back to the papers published. ESSD wants papers to carry enough info to sufficiently understand the contents.

*The data we could harmonise over is full year 2019 data, or we assume one dataset is correct. The datasets differ in how they map variability to the previous year. Some work of constant 2019 emissions (Forster et al), others work off the absolute change (Liu et al). The methods also have different estimates for Jan, Feb, etc, so it is not so easy or obvious to harmonise in the way suggested. This is more a structural uncertainty in the way the studies are perfor*med

**Reviewer 4**

**Individual Comments:**

Abstract line 65-66 - not clear why 6% is the median of [6,6,7,13]...add decimal point to numbers?

*We updated with the latest estimates, and included the uncertainty range from one study. The wording is changed slightly to help clarify. We did not include decimal points, as we are trying to emphasis the uncertainty here.*

Lines 103-105 - glad to see this, but perhaps you could be a bit more clear that the term" global carbon budget" is explicitly different and should not be confused with "remaining carbon budget" and it variants. Could add a citation here to [Matthews et al (2020) Opportunities and challenges in using remaining carbon budgets to guide climate policy, Nature Geoscience, in press.] who also articulate this distinction.

*Thanks, we clarified in the revised version. We can not refer to Matthews et al 2020 as it is not published yet.*

Lines 339-341 - unclear to me how uncertainty in consumption-based emissions would not be larger than territorial, since you are adding additional uncertain components like trade flows and sectoral emissions factors.

*The available literature suggests that various new errors in the calculation of consumption-based estimates tend to cancel. This literature is summarised in Peters et al 2012, the cited paper.*

Lines 1047-1052 - this sentence is extremely long, and somewhat grammatically problematic. Also, I don't see mention of how gross fluxes were calculated in the methods section.

*We have made the description of what the gross fluxes comprise easier to read and moved it to the methods section, where we add that these fluxes are internally calculated by each bookkeeping model.*

Lines 1368-1373 - I have noticed a general trend the past few years of overestimating the projected growth rate in fossil fuel emissions (e.g. last year's projection of 0.6% has been downgraded to 0.1% ... the projection for 2018 in LeQuere2018 was 2.7% and subsequently revised to 2.1% in F19...2017 was projected at 2.0% then subsequently updated to 1.6%...). Not sure if this is significant or not, but the relative overestimate in the projected value has

been pretty consistent for the past few years, so maybe speaks to some bias in the projection methodology that is worth considering.

*Much of this relates to China. The China method now (noting 2020 is an outlier) corrects this bias. The bias tends to be that the real estimate tends to be closer to zero.*

Lines 1454-1455 - why report the median here (and in the abstract) rather than the mean? The relative independence of the different estimate is also not terribly clear on first glance, the UEA and GCB methods seem quite similar (also given the overlap in authorship), whereas the Priestly Centre estimate is more independent in terms of methodology. Given the various results presented here (including also the IEA estimate), a mean decrease of 8% seems equally plausible as a median of 6%."

*The estimates are quite independent. The GCB approach is the most independent, based on monthly data. The three others use similar proxy data. Forster et al uses some data from Le Quere et al, suggesting they may be the most dependent. One reason to take the median is to put less weight on Forster et al. As described in Forster et al, the method probably leads to an overestimate. The three studies with the similar estimates are perhaps the three most independent. The median also gives less weight to outliers.*